# Multivariate Time Series Anomaly Detection with Idempotent Reconstruction

**Xin Sun**
Zhejiang University
xinsun1@zju.edu.cn

**Heng Zhou**
Zhejiang University
hengzhou@zju.edu.cn

**Chao Li**$^*$
Zhejiang University
chaoli@zju.edu.cn

## Abstract

Reconstruction-based methods are competitive choices for multivariate time series anomaly detection (MTS AD). However, one challenge these methods may suffer is over generalization, where abnormal inputs are also well reconstructed. In addition, balancing robustness and sensitivity is also important for final performance, as robustness ensures accurate detection in potentially noisy data, while sensitivity enables early detection of subtle anomalies. To address these problems, inspired by idempotent generative network, we take the view from the manifold and propose a novel module named **I**dempotent **G**eneration for **A**nomaly **D**etection (IGAD) which can be flexibly combined with a reconstruction-based method without introducing additional trainable parameters. We modify the manifold to make sure that normal time points can be mapped onto it while tightening it to drop out abnormal time points simultaneously. Regarding the latest findings of AD metrics, we evaluated IGAD on various methods with four real-world datasets, and they achieve visible improvements in VUS-PR than their predecessors, demonstrating the effective potential of IGAD for further improvements in MTS AD tasks. Our instructions on integrating IGAD into customized models and example codes are available at `https://github.com/ProEcho1/Idempotent-Generation-for-Anomaly-Detection-IGAD`.

## 1 Introduction

Multivariate time series (MTS) are continuously collected from numerous sensors [3, 56], which is widely present in many critical scenarios, such as production data from multiple devices in modern factories and monitoring data from various sensors in smart grids [12, 16, 25]. The task of anomaly detection (AD) in multivariate time series lies in determining whether each time point within the series is normal or abnormal, helping to identify possible malfunctions and minimize losses [69, 28, 8], where each time point in the series can be viewed as a time instance.

Unsupervised anomaly detection methods often develop reconstruction-based models, such as [54, 62, 50]. All time points are label-absent and viewed as normal in the training data. These models tend to show an encoder-decoder architecture, producing small reconstruction errors for normal time points and larger reconstruction errors for abnormal ones after training the ability to reconstruct only on these normal data, as in the second case in Fig.1. However, one challenge these methods may suffer is *over generalization* [48], where abnormal time points can also be reconstructed so well that it becomes harder to distinguish them [6, 39, 17], as shown in the first case of Fig.1. We conclude the reasons for this issue as that this may happen when: (1) the built model incorrectly captures the intrinsic patterns of abnormal series in a contaminated dataset for training; or (2) the model has an excessive decoding power, even for abnormal series. We provide math analysis for these in **Appendix.A**. The phenomenon of contaminated datasets exists across multiple domains,

---

$^*$ The Corresponding Author.

39th Conference on Neural Information Processing Systems (NeurIPS 2025).

such as data mining [30, 68] and computer vision [40, 49]. Moreover, it is difficult and impractical to guarantee clean time series for training during data collection, or manually and accurately control the feature extraction capabilities of the encoders and the reconstruction capabilities of the decoders.

Meanwhile, an additional issue is ***the balance of robustness and sensitivity*** of these models. On the one hand, robustness to noisy data is a frequently disregarded factor in time series anomaly detection applications, which refers to the capacity to accurately identify normal and abnormal time points even in the presence of noise. This is essential in practical applications, as real-world data unavoidably contain noise that can arise from sensor errors, loss of data transmission, or other external factors [73].

If a model performs poorly on noisy data, it may fail to detect true anomalies or falsely identify normal data as anomalous. Such situations can have severe consequences in critical applications, including industrial monitoring and medical diagnostics. However, sensitivity is crucial for identifying subtle deviations from normal patterns, which can indicate early signs of potential anomalies. High sensitivity enables the model to detect these minor aberrations, ensuring timely interventions. A most excessive case can be illustrated in the third example of Fig.1, where abnormal time points and normal time points affected by noise are close and mixed to distinguish. Hence, it is important to balance robustness and sensitivity to avoid false positives due to noise while maintaining the reliability of anomaly detection.

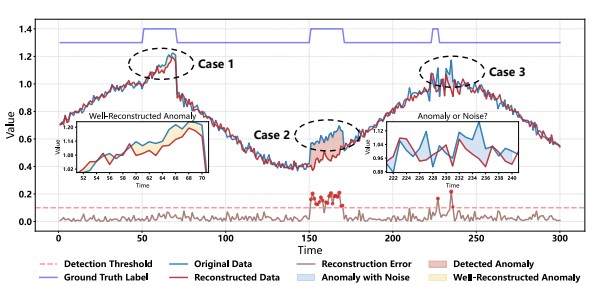

Figure 1: Three existing phenomena in the methods based on reconstruction. The issues of over generalization, correct detection, and the conflict of robustness and sensitivity are shown from left to right, marked from **Case 1** to **Case 3**.

To overcome these limitations, drawing inspiration from the idempotent generative network [47], we have adapted this method to the task of detecting anomalies in multivariate time series data. Our modification allows for the integration of this flexible module with existing reconstruction-based methods. First, we introduce the concept of idempotent generation. In general, we present $f(\cdot)$, $z$ as an encoder-decoder mapping network and a vector, respectively. An idempotent operator is one that, when applied multiple times in sequence, does not alter the result beyond the effect of the initial application, which can be denoted as $f(f(z)) = f(z)$. Second, we can regard the concept of a manifold as the relationships between these multidimensional time points and the geometric structures they form, which reflect the underlying patterns and dynamics of the time series data. Each time point can be viewed as an instance in the manifold. Here, we assume that a source distribution $\mathcal{P}_z$ and a target distribution $\mathcal{P}_x$ are in the same latent space. However, unlike Shocher et al. [47], we define $\mathcal{P}_x$ as a distribution consisting only of normal time instances. In this setting, when performing $f(x)$ for a given normal time instance $x \sim \mathcal{P}_x$, the ideal outcome is that $x$ remains unchanged. **Further, we represent our target manifold as the subset of all normal time instances $x$ that can be assigned to themselves after applying $f(\cdot)$ to minimize reconstruction errors.**

Given these, AD tasks can benefit from two aspects: (1) we use a modified $z$ under frequency domain features for a more controllable generation to address the challenge of balancing robustness and sensitivity, instead of directly sampling $z \sim \mathcal{N}(\mathbf{0}, \mathbf{I})$ from $\mathcal{P}_z$. For a time window consisting of normal time instances as input, it is practical to perform a Fast Fourier Transform (FFT) to extract information about frequency components and spectral features, and an Inverse Fast Fourier Transform (IFFT) with resampling is selected to generate noise $z$ that contains intrinsic patterns of normal series. Then $f(f(z)) = f(z)$ is performed to map these noise-affected instances to the target manifold. Through artificially introduced random factors and the idempotent constraint, our goal is to balance robustness and sensitivity to capture inherent data patterns in normal time series; (2) we further introduce $f(f(z)) \neq f(z)$ to tighten the target manifold to address the problem of over generalization. This adversarial strategy can be employed to prevent the manifold from incorrect expansion, excluding potential abnormal instances in the manifold. We will give more detailed explanations in Sect.3.

The major contributions of our work can be three-folded: (1) We explore ***over generalization*** and ***the balance between robustness and sensitivity*** from a manifold perspective, establishing the links of the manifold with these limitations; (2) To overcome these mentioned issues, we propose a novel module

named **I**dempotent **G**eneration for **A**nomaly **D**etection (IGAD), which can be flexibly integrated with reconstruction-based methods without introducing parameters that need to be trained; (3) Based on our experimental results in VUS-PR, noise-affected verification, and the distributions of anomaly scores, we demonstrate the effectiveness of IGAD in further improving the performance of different models in multivariate time series anomaly detection tasks.

## 2 Related Work

### 2.1 Multivariate Time Series Anomaly Detection

Traditional machine learning methods, such as LOF [7], OC-SVM [44], SVDD [51], and Isolation Forest [31] are widely used in anomaly detection. More advances include MPPCACD [65] and DAGMM [75], which integrate density estimation with deep representation learning. Clustering-based approaches, such as Deep SVDD [43], optimize a hypersphere to enclose normal samples, while extensions like Fuzzy C-Means [29] offer alternatives. In addition, deep models for sequential data [43, 9, 46] have been proposed. Contrastive learning-based methods, such as CARLA [11], DCdetector [66], TS-TCC [15] and CoST [57], are also well-designed AD models or powerful representation learning models that can be used for downstream tasks like AD.

Reconstruction-based methods trained in a self-supervised manner to regenerate inputs with high accuracy have made notable progress. Early methods include LSTM-based encoder-decoder models [35, 36], with OmniAnomaly [50] a further development. DAGMM [75], MSCRED [70], MTAD-GAT [72], USAD [4], CAE-M [71], FGANomaly [13], Anomaly Transformer [62], M2N2 [26], and XGBoost-based for in-core neutronare detectors [63] are also proposed as competitive coordinates. Most recently, spatial association-aware SARAD [10] has been proposed with joint time-spatial features. CATCH [60] performs MTS AD with frequency patching. Meanwhile, advanced time series foundation models such as OFA [74], TimesNet [58], FITS [64], Peri-midFormer [59], and ModernTCN [34] have shown their powerful abilities in various time series tasks, including reconstruction-based anomaly detection. However, one challenge with which these methods may struggle is over generalization [48], where abnormal inputs are too well reconstructed [6, 39, 17].

### 2.2 Idempotent Generative Network

The first work to raise the concept of idempotent generative network can be found in Shocher et al. [47], where they use this architecture to generate novel image samples belonging to a specific domain. Reconstruction objective, idempotent objective and tightness objective are included to ensure potential instances lying on the target manifold, mapping a random vector to the manifold and tightening the manifold to avoid unnecessary expansion. Recently, TrajCLIP [67] has employed this concept to predict the trajectory of pedestrians. In addition, it is also used to perform conditional generation [42] and test-time training [14]. However, the applicability of this theory to address limitations in the time series domain has not been fully explored.

## 3 Method

### 3.1 Overview

In this paper, we propose IGAD, a novel and flexible module based on idempotent reconstruction, to further improve existing reconstruction-based anomaly detection methods. IGAD introduces two key objectives: the idempotent objective, ensuring that both $x$ and $f(z)$ lie in the target manifold for improved robustness and sensitivity, and the tightness objective, which tightens the target manifold to mitigate the issues of over generalization. The overall architecture is shown in Fig.2.

### 3.2 Problem Setting

We define an MTS dataset as $\mathbf{D} = \{x_1, x_2, x_3, ..., x_n\}$ and each time point $x_i \in \mathbb{R}^k$, where $n$ represents the total number of time instances and $k$ represents the number of variables in MTS. In practice, we define a window size $w$, selecting every $w$ time points as a time interval, that is, $x^i \in \mathbb{R}^{w \times k}$ for $i = 1$ to $m$, with $m$ the number of time intervals. Furthermore, we denote each

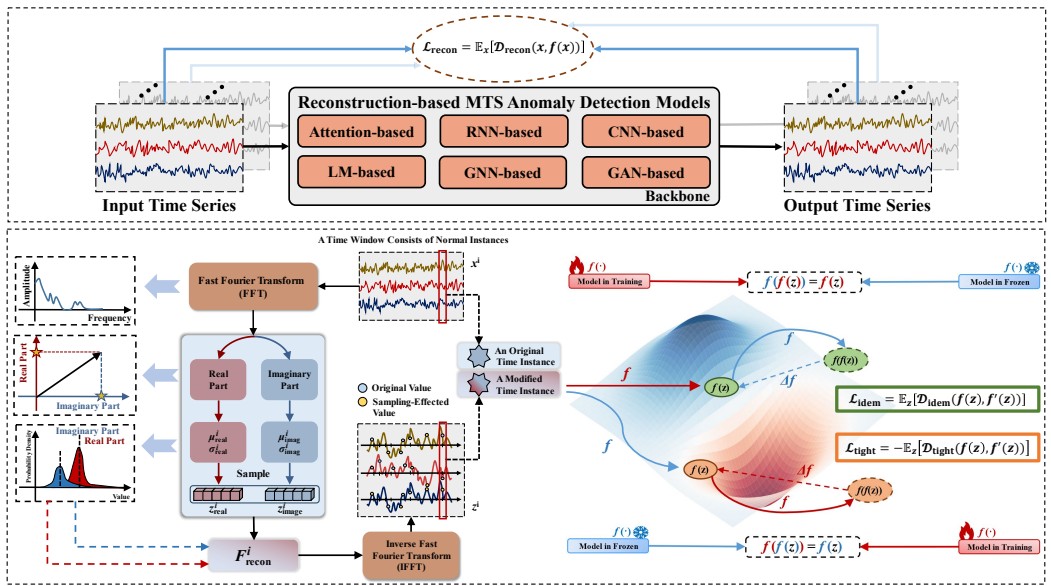

Figure 2: The overall architecture of reconstruction-based methods integrated with our proposed module IGAD. The upper part is the conventional reconstruction branch, which can be used to calculate $\mathcal{L}_{\text{recon}}$ in Eq.(1). The lower part shows the workflow of IGAD, where **the red arrow** $\rightarrow$ and **the blue arrow** $\rightarrow$ in the right manifold stand for the training and frozen model, i.e., $f(\cdot)$ and $f'(\cdot)$, respectively. After acquiring an augmented $z^i$ which consists of normal instances, it is passed through $f(f'(z^i))$ and $f'(f(z^i))$ to calculate $\mathcal{L}_{\text{idem}}$ and $\mathcal{L}_{\text{tight}}$, shown as Eq.(7) and Eq.(8).

existing reconstruction-based method as an idempotent generation function $f(\cdot)$ that takes a time interval $x^i$ or a vector $z$ as input and then produces the corresponding output $f(x^i)$ or $f(z)$. Here, we should mention that reconstructing each normal time instance well is equivalent to reconstructing each normal time interval well, since each time interval contains $w$ time instances.

## 3.3 Optimization Objective

### 3.3.1 Reconstruction Objective

As a crucial standard for reconstruction-based methods, they should well rebuild normal time intervals to produce minimal reconstruction loss. This objective can be achieved perfectly if $f(x^i)$ can be as close to itself as possible for each $x^i$. It is widely assumed that each $x^i$ in the training dataset is in a normal state. To evaluate the distance between each $x^i$ and its reconstructed sample $x^i_{\text{recon}}$ by $f(x^i)$, we define this objective as:

$$\mathcal{D}_{\text{recon}}(x, f(x)) = \frac{1}{m} \sum_{i=1}^{m} \|x^i - x^i_{\text{recon}}\|_2^2, \text{ with } \mathcal{L}_{\text{recon}}(x, f(x)) = \mathbb{E}_x[\mathcal{D}_{\text{recon}}(x, f(x))], \quad (1)$$

where $m$ is the total number of windows defined above. Based on this, we can present our target manifold, which contains only normal instances, more formally as:

$$\mathcal{M}_{\text{target}} = \{x^i : f(x^i) = x^i\} = \{x_j : \text{for each } x_j \in x^i\}. \quad (2)$$

This means that these models should reconstruct the samples well enough to make sure that each normal instance lies on the target manifold. However, they may exactly reconstruct abnormal instances, which affects the final detection accuracy. To further explore this, we introduce the following optimization objectives.

### 3.3.2 Idempotent Objective

The previous work [47] performs a mapping from a source distribution $\mathcal{P}_z$ (standard normal distribution) to $\mathcal{P}_x$ to generate novel image samples. However, in the field of MTS AD, we should inject more

crucial information into $\mathcal{P}_z$ for a more controllable and reliable generation. Moreover, in practical situations, the MTS data collection process may be influenced by disturbances and fluctuations from physical environments, which can be natural noise. If a model is able to extract valid features from noisy normal time series data, it indicates that the model has a higher sensitivity to the intrinsic normal patterns. Given these, we reform the role of this objective to extract more natural features from normal instances, further balancing robustness and sensitivity rather than generating novel samples. Concretely, we make an FFT for each original time interval $x^i$ in order to convert the time-domain signal to a frequency-domain signal, and the result usually takes a complex form. This process can be presented as:

$$\mathcal{F}\{x^{i,j}(t)\} = \mathbf{X}^{i,j}(F) = \mathrm{Re}(\mathbf{X}^{i,j}(F)) + \mathbf{i} \cdot \mathrm{Im}(\mathbf{X}^{i,j}(F)), \tag{3}$$

where $\mathcal{F}$ is the Fourier transform operator, and $x^{i,j}$ stands for the $j$-th time variable in the $i$-th time interval. $\mathbf{X}^{i,j}(F)$ is the transformed data with $F$ representing the frequency variable in the frequency domain. $\mathrm{Re}(\cdot)$ and $\mathrm{Im}(\cdot)$ are selected to obtain the real and imaginary part of $\mathbf{X}^{i,j}(F)$.

After acquiring these, we calculate the mean and standard deviation of $\mathrm{Re}(\mathbf{X}^{i,j}(F))$ and $\mathrm{Im}(\mathbf{X}^{i,j}(F))$ for the time interval $x^i$, denoted as $\mu_{\mathrm{real}}^i$, $\mu_{\mathrm{imag}}^i$, $\sigma_{\mathrm{real}}^i$ and $\sigma_{\mathrm{imag}}^i$, respectively. Now we can define two separate distributions on the real part and the imaginary part as $\mathcal{P}_{\mathrm{real}}^i = \mathcal{N}(\mu_{\mathrm{real}}^i, \sigma_{\mathrm{real}}^i)$ and $\mathcal{P}_{\mathrm{imag}}^i = \mathcal{N}(\mu_{\mathrm{imag}}^i, \sigma_{\mathrm{imag}}^i)$. Then, $z_{\mathrm{real}}^i$ and $z_{\mathrm{imag}}^i$ can be randomly sampled from $\mathcal{P}_{\mathrm{real}}^i$ and $\mathcal{P}_{\mathrm{imag}}^i$ to get the rebuilt frequency variable:

$$F_{\mathrm{recon}}^i = z_{\mathrm{real}}^i + \mathbf{i} \cdot z_{\mathrm{imag}}^i. \tag{4}$$

By performing inverse fast Fourier transform on $F_{\mathrm{recon}}^i$, we can get the final modified latent vector $z^i$, which is defined as:

$$z^i = \mathrm{Re}(\mathcal{F}^{-1}(F_{\mathrm{recon}}^i)), \tag{5}$$

where $z^i$ is the augmented vector for a time interval $x^i$. Instead of randomly sampling $z^i$, we inject the intrinsic features of normal instances into the final $z^i$ while introducing possible randomness of real applications to ensure that $\mathcal{M}_{\mathrm{target}}$ can contain enough normal instances. Then we utilize $f(f(z^i)) = f(z^i)$ to strengthen the reliability of $\mathcal{M}_{\mathrm{target}}$, reconstructing time intervals with normal features as well as possible. To avoid the instability during training caused by MSE Loss being overly sensitive to potential noise, the optimization objective can be formulated as:

$$\mathcal{D}_{\mathrm{idem}}(f(z), f(f(z))) = \frac{1}{m}\sum_{i=1}^{m} \left| f(z^i) - f(f(z^i)) \right|. \tag{6}$$

### 3.3.3 Tightness Objective

By introducing the reconstruction objective and the idempotent objective, we pay more attention to the abilities of reconstruction and recognition of normal data patterns. However, we must take one extreme case into consideration: If one model learns a mapping that can be formulated as $f(z^i) = z^i$, that is, their outputs are just their inputs, they will satisfy all the mentioned objectives perfectly while losing their meaning in MTS anomaly detection.

Here, we further address the issue of over generalization from the perspective of the defined $\mathcal{M}_{\mathrm{target}}$. Specifically, as shown in Fig.3, there are two different flow paths of gradients in models when minimizing the idempotent objective defined as (6). The first path is marked with **green line**, performing $f(f(z^i))$ to ensure that $z^i$ is better mapped to $\mathcal{M}_{\mathrm{target}}$. However, the second path marked with **red line** imports the potential risk of expanding the target manifold to include all visible samples. This phenomenon can explain the reason for over generalization: **accident expansion of manifold in a model.**

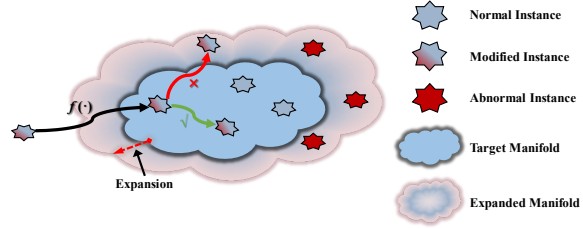

Figure 3: The expansion of manifold results in potential abnormal instances are included in the target manifold $\mathcal{M}_{\mathrm{target}}$.

It becomes necessary to encourage $\mathcal{M}_{\mathrm{target}}$ to be decorated adequately and reasonably to contain more normal instances, as well as avoid unnecessary expansion so that abnormal samples are not included in $\mathcal{M}_{\mathrm{target}}$. To address these, we divide our idempotent objective into two parts as shown in Fig.2. First, we exclusively optimize with respect to the first

instantiation of $f(\cdot)$, while freezing the second instantiation as a fixed copy of the current state of $f(\cdot)$, denoted as $f'(\cdot)$. Then the first part of the idempotent objective can be formulated as:

$$\mathcal{D}_{\text{idem}}(f(z), f'(f(z))) = \frac{1}{m} \sum_{i=1}^{m} \left| f(z^i) - f'(f(z^i)) \right|, \quad \mathcal{L}_{\text{idem}}(f(z), f'(z)) = \mathbb{E}_z[\mathcal{D}_{\text{idem}}(f(z), f'(f(z)))]. \tag{7}$$

Here, $f'(\cdot)$ can be viewed as a static mapping function, so $\mathcal{L}_{\text{idem}}$ is designed to prevent the potential expansion of the marked red path in Fig.3 by bringing $f(z^i)$ and $f'(f(z^i))$ closer. Meanwhile, to tighten $\mathcal{M}_{\text{target}}$ and exclude potential abnormal instances, we present the second part of the idempotent objective as a tightness objective. As we mentioned earlier, the red path in Fig.3 tends to expand $\mathcal{M}_{\text{target}}$. This inspires us that the **Inverse Effect** of this path can be selected to further tighten $\mathcal{M}_{\text{target}}$. Concretely, we only optimize the second instantiation of $f(\cdot)$ and treat the first instantiation as frozen, using the opposite of the distance between $f'(z^i)$ and $f(f'(z^i))$ to simulate this inverse effect:

$$\mathcal{D}_{\text{tight}}(f'(z), f(f'(z))) = \frac{1}{m} \sum_{i=1}^{m} \left| f'(z^i) - f(f'(z^i)) \right|, \quad \mathcal{L}_{\text{tight}}(f(z), f'(z)) = \mathbb{E}_z[-\mathcal{D}_{\text{tight}}(f'(z), f(f'(z)))]. \tag{8}$$

Due to the relationship between idempotent objective and tightness objective which can be shown as:

$$\mathcal{L}_{\text{idem}}(f(z), f'(z)) = -\mathcal{L}_{\text{tight}}(f'(z), f(z)), \tag{9}$$

an adversarial training strategy is introduced to make sure that $\mathcal{M}_{\text{target}}$ strongly focus on normal time instances, while tighten $\mathcal{M}_{\text{target}}$ so that potential abnormal instances are dropped out. However, there is a possible adverse effect in $\mathcal{L}_{\text{tight}}$ is that it may perform excessive modification to minimize its values, which increases the instability of the gradient during training. Hence, we smooth $\mathcal{L}_{\text{tight}}$ by:

$$\mathcal{L}_{\text{tight}}^* = \text{tahn}(\frac{\mathcal{L}_{\text{tight}}}{\alpha \mathcal{L}_{\text{recon}}})\alpha \mathcal{L}_{\text{recon}}, \tag{10}$$

where $\alpha$ is the control parameter. The meaning of this operation can be concluded that if a certain $x^i$ has larger reconstruction loss, it is mapped far from $\mathcal{M}_{\text{target}}$ and it is unnecessary to push the model so strong based on $z^i$ to effect the whole training process.

### 3.3.4 Final Objective

The final objective of a model can be divided into four parts: reconstruction loss ($\mathcal{L}_{\text{recon}}$), idempotent loss ($\mathcal{L}_{\text{idem}}$), tightness loss ($\mathcal{L}_{\text{tight}}^*$) and other auxiliary loss ($\mathcal{L}_{\text{aux}}$) in each original model. We provide the proofs for the coverage to normal instances in **Appendix.B.**, and the final objective is:

$$\mathcal{L} = \lambda_{\text{rec}}\mathcal{L}_{\text{recon}} + \lambda_{\text{idem}}\mathcal{L}_{\text{idem}} + \lambda_{\text{tight}}\mathcal{L}_{\text{tight}}^* + \lambda_{\text{aux}}\mathcal{L}_{\text{aux}} = \mathbb{E}_{x,z}[\lambda_{\text{rec}}\mathcal{D}_{\text{recon}} + \lambda_{\text{idem}}\mathcal{D}_{\text{idem}} - \lambda_{\text{tight}}\mathcal{D}_{\text{tight}} + \lambda_{\text{aux}}\mathcal{D}_{\text{aux}}]. \tag{11}$$

## 4 Experiment

### 4.1 Experiment Setting

#### 4.1.1 Dataset and Baseline

In our experiments, we selected four redesigned public datasets commonly used for MTS AD in [32], including SMD from [50], MSL from [24], PSM from [1] and SMAP from [24].

We have selected 15 reconstruction-based multivariate time series anomaly detection methods in our experiments, including: **(1) Specific models designed for MTS AD:** SARAD [10], Anomaly Transformer [62], FGANomaly [13], CAE-M [71], MTAD-GAT [72], MSCRED [70], OnimAnomaly [50] and DAGMM [75]; **(2) Time Series Foundation Models:** FITS [64], Peri-midFormer [59], ModernTCN [34], OFA [74] and TimesNet [58]. These basic models contain different intrinsic architectures, including GNN [61], Attention Mechanism [53, 27], GAN [18], GPT2 [41], and others shown in Fig.2. Some models unavailable are re-implemented from Tuli et al. [52]. More detailed introductions and descriptions of these models can be found in **Appendix.C**.

#### 4.1.2 Implemention Detail

The latest study on datasets and benchmarks [32] has designed well-organized datasets and searched for optimal hyperparameters in optimizer, learning rate, and weights of existing loss functions for

Table 1: VUS-PR on reconstruction-based models under different real-world datasets. All experimental results are repeated with five random seeds and reported as *Mean ± Standard Deviation*. The model-level improvement ratios $\Delta_{\text{model}}$ and the dataset-level improvement ratios $\Delta_{\text{data}}$ are calculated. Meanwhile, Wilcoxon signed rank tests are performed for anomaly scores, where we use ***, **, and * to denote statistical significance in 1%, 5% and in cases where $p$-values are greater than 0.05.

| Model | Venue | Dataset | | | | | | | |
|---|---|---|---|---|---|---|---|---|---|
| | | SMD | | | | MSL | | | |
| | | w / o IGAD | w / IGAD | $\Delta_{\text{model}}$ (%) | $p$-value | w / o IGAD | w / IGAD | $\Delta_{\text{model}}$ (%) | $p$-value |
| CATCH | ICLR, 2025 | $0.1904 \pm 0.0034$ | $0.1970 \pm 0.0022$ | +3.47 | *** | $0.0331 \pm 0.0010$ | $0.0334 \pm 0.0016$ | +0.91 | *** |
| M2N2 | AAAI, 2024 | $0.0211 \pm 0.0002$ | $0.0362 \pm 0.0066$ | +71.56 | *** | $0.2024 \pm 0.1532$ | $0.8250 \pm 0.2050$ | +307.61 | *** |
| FITS | ICLR, 2024 | $0.0409 \pm 0.0027$ | $0.0551 \pm 0.0027$ | +34.72 | *** | $0.0087 \pm 0.0017$ | $0.0107 \pm 0.0038$ | +22.99 | ** |
| ModernTCN | ICLR, 2024 | $0.1521 \pm 0.0019$ | $0.1954 \pm 0.0006$ | +28.47 | *** | $0.0486 \pm 0.0054$ | $0.0620 \pm 0.0419$ | +27.57 | *** |
| Peri-midFormer | NeurIPS, 2024 | $0.1545 \pm 0.0073$ | $0.1573 \pm 0.0078$ | +1.81 | *** | $0.0330 \pm 0.0017$ | $0.0356 \pm 0.0011$ | +7.88 | * |
| SARAD | NeurIPS, 2024 | $0.2266 \pm 0.0047$ | $0.2256 \pm 0.0041$ | -0.44 | *** | $0.0277 \pm 0.0134$ | $0.0504 \pm 0.0549$ | +81.95 | *** |
| TimesNet | ICLR, 2023 | $0.0741 \pm 0.0153$ | $0.0759 \pm 0.0265$ | +2.43 | *** | $0.0072 \pm 0.0028$ | $0.0073 \pm 0.0030$ | +1.39 | *** |
| OFA | NeurIPS, 2023 | $0.0559 \pm 0.0027$ | $0.1064 \pm 0.0183$ | +90.34 | *** | $0.0083 \pm 0.0050$ | $0.0105 \pm 0.0091$ | +26.51 | *** |
| A.T. | ICLR, 2022 | $0.0259 \pm 0.0043$ | $0.0421 \pm 0.0402$ | +62.55 | *** | $0.0063 \pm 0.0005$ | $0.0064 \pm 0.0009$ | +1.59 | *** |
| FGANomaly | TKDE, 2021 | $0.3611 \pm 0.0041$ | $0.3615 \pm 0.0117$ | +0.11 | *** | $0.0514 \pm 0.0128$ | $0.0578 \pm 0.0087$ | +12.45 | * |
| CAE-M | TKDE, 2021 | $0.0888 \pm 0.0909$ | $0.0863 \pm 0.0985$ | -2.82 | *** | $0.0043 \pm 0.0001$ | $0.0045 \pm 0.0001$ | +4.65 | *** |
| MTAD-GAT | ICDM, 2020 | $0.3764 \pm 0.0016$ | $0.4169 \pm 0.0250$ | +10.76 | *** | $0.1731 \pm 0.0022$ | $0.2504 \pm 0.1418$ | +44.66 | * |
| OmniAnomaly | KDD, 2019 | $0.2096 \pm 0.0022$ | $0.2139 \pm 0.0021$ | +2.05 | *** | $0.0052 \pm 0.0002$ | $0.0087 \pm 0.0028$ | +67.31 | *** |
| MSCRED | AAAI, 2019 | $0.3220 \pm 0.0277$ | $0.3245 \pm 0.0206$ | +0.78 | *** | $0.0111 \pm 0.0007$ | $0.0102 \pm 0.0008$ | -8.11 | * |
| DAGMM | ICLR, 2018 | $0.0322 \pm 0.0031$ | $0.1338 \pm 0.0476$ | +315.53 | *** | $0.0042 \pm 0.0009$ | $0.0068 \pm 0.0021$ | +61.90 | *** |
| $\Delta_{\text{data}}$ (%) | | Mean: 0.1554 | Mean: 0.1752 | +12.71 | | Mean: 0.0416 | Mean: 0.0920 | +120.89 | |

| Model | Venue | Dataset | | | | | | | |
|---|---|---|---|---|---|---|---|---|---|
| | | PSM | | | | SMAP | | | |
| | | w / o IGAD | w / IGAD | $\Delta_{\text{model}}$ (%) | $p$-value | w / o IGAD | w / IGAD | $\Delta_{\text{model}}$ (%) | $p$-value |
| CATCH | ICLR, 2025 | $0.1284 \pm 0.0031$ | $0.1326 \pm 0.0028$ | +3.27 | *** | $0.2882 \pm 0.0012$ | $0.2931 \pm 0.0009$ | +1.70 | *** |
| M2N2 | AAAI, 2024 | $0.2989 \pm 0.0055$ | $0.3010 \pm 0.0021$ | +0.70 | *** | $0.1934 \pm 0.0046$ | $0.1973 \pm 0.0425$ | +2.02 | *** |
| FITS | ICLR, 2024 | $0.1163 \pm 0.0003$ | $0.1173 \pm 0.0006$ | +0.86 | *** | $0.2704 \pm 0.0116$ | $0.2851 \pm 0.0128$ | +5.44 | *** |
| ModernTCN | ICLR, 2024 | $0.1383 \pm 0.0002$ | $0.1337 \pm 0.0002$ | -3.33 | *** | $0.4561 \pm 0.0094$ | $0.4144 \pm 0.0077$ | -9.14 | *** |
| Peri-midFormer | NeurIPS, 2024 | $0.1310 \pm 0.0005$ | $0.1311 \pm 0.0004$ | +0.08 | *** | $0.5064 \pm 0.0285$ | $0.5067 \pm 0.0254$ | +0.06 | *** |
| SARAD | NeurIPS, 2024 | $0.1499 \pm 0.0078$ | $0.1550 \pm 0.0049$ | +3.40 | *** | $0.8469 \pm 0.0189$ | $0.8477 \pm 0.0176$ | +0.09 | *** |
| TimesNet | ICLR, 2023 | $0.1174 \pm 0.0015$ | $0.1297 \pm 0.0055$ | +10.48 | *** | $0.2678 \pm 0.0758$ | $0.2734 \pm 0.0688$ | +2.09 | *** |
| OFA | NeurIPS, 2023 | $0.1261 \pm 0.0002$ | $0.1405 \pm 0.0082$ | +11.42 | *** | $0.2929 \pm 0.0255$ | $0.2969 \pm 0.0254$ | +1.37 | *** |
| A.T. | ICLR, 2022 | $0.1158 \pm 0.0083$ | $0.1517 \pm 0.0352$ | +31.00 | *** | $0.2397 \pm 0.0855$ | $0.2578 \pm 0.1130$ | +7.55 | *** |
| FGANomaly | TKDE, 2021 | $0.1970 \pm 0.0046$ | $0.1838 \pm 0.0054$ | -6.70 | *** | $0.9192 \pm 0.0274$ | $0.9835 \pm 0.0012$ | +7.00 | *** |
| CAE-M | TKDE, 2021 | $0.1503 \pm 0.0006$ | $0.1504 \pm 0.0129$ | +0.07 | *** | $0.0736 \pm 0.0001$ | $0.0736 \pm 0.0001$ | 0.00 | *** |
| MTAD-GAT | ICDM, 2020 | $0.1433 \pm 0.0025$ | $0.1785 \pm 0.0383$ | +24.56 | *** | $0.2320 \pm 0.0277$ | $0.5131 \pm 0.2348$ | +121.16 | *** |
| OmniAnomaly | KDD, 2019 | $0.1427 \pm 0.0008$ | $0.1431 \pm 0.0009$ | +0.28 | *** | $0.0780 \pm 0.0002$ | $0.9060 \pm 0.0272$ | +1061.54 | *** |
| MSCRED | AAAI, 2019 | $0.1902 \pm 0.0118$ | $0.1727 \pm 0.0171$ | -9.20 | *** | $0.0958 \pm 0.0009$ | $0.1274 \pm 0.0208$ | +32.99 | *** |
| DAGMM | ICLR, 2018 | $0.1672 \pm 0.0015$ | $0.1675 \pm 0.0025$ | +0.18 | *** | $0.0748 \pm 0.0013$ | $0.1083 \pm 0.0140$ | +44.79 | *** |
| $\Delta_{\text{data}}$ (%) | | Mean: 0.1542 | Mean: 0.1592 | +3.28 | | Mean: 0.3223 | Mean: 0.4056 | +25.83 | |

the majority models included in this study. For some of our selected base models, which are not temporarily imported, we set their hyperparameters in their original papers or repositories as optimal ones. Then these models are also integrated into this proposed pipeline to run in a universal data flow. Under these settings, $\lambda_{\text{rec}}$ and $\lambda_{\text{aux}}$ are fixed, then intervals [0.1, 1.0] with a step size of 0.1 for $\lambda_{\text{idem}}$ and $\lambda_{\text{tight}}$, as well as [1.1, 1.5] with a step size of 0.1 for $\alpha$ are used for a detailed grid search. Optuna [2] is selected for the search process. We program our codes with Python 3.8.13, PyTorch 1.13.0, CUDA 11.7 and Ubuntu 18.04 on a single NVIDIA RTX 3090 24GB GPU. All experiments are conducted under the same environments. Meanwhile, in our selected models, OFA [74] is an anomaly detection model based on GPT-2 [41], so it is trained after loading the pre-trained weights from huggingface. For the other models, they are trained from initialization according to the random seeds, following the standard training principle. More information on these can be found in **Appendix.D**.

### 4.1.3 Evaluation Metric

It has been highlighted that traditionally used MTS AD metrics such as F1, AUC-PR, AUC-ROC and Affiliation-F1 could show potential evaluation issues, while in comparison, VUS-PR emerges as the most robust, accurate and fair evaluation measure [32]. Given these, VUS-PR is selected as our key metric. For further detailed information, we also record the rest of these metrics, and more results such as hyperparameter analysis and visualization can be found in **Appendix.E**.

Following the work pipeline in Liu and Paparrizos [32], we use reconstruction errors for MTS AD. Concretely, given the original time series $X = \{x_0, x_1, x_2, ..., x_{n-1}\}$ and the reconstructed $X^{\text{recon}} = \{x_0^{\text{recon}}, x_1^{\text{recon}}, x_2^{\text{recon}}, ..., x_{n-1}^{\text{recon}}\}$, the reconstruction errors $e_t$ can be calculated as:

$$e_t = \|x_t - x_t^{\text{recon}}\|_2^2, \ t = 0, 1, \ldots, n-1, \tag{12}$$

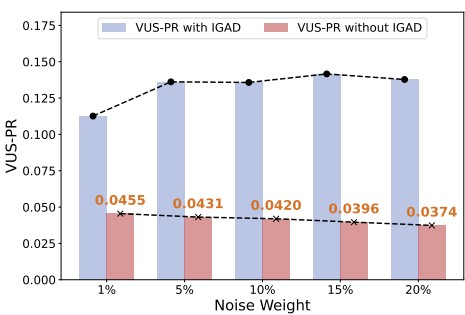
(a) VUS-PR under noise for SARAD on MSL.

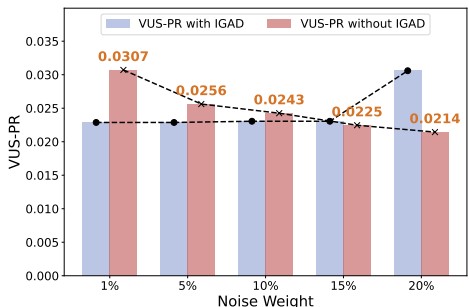
(b) VUS-PR under noise for A.T. on SMD.

Figure 4: Performance under different weighted noise. The weights of noise vary from 1% to 20%.

$$s_t = e_t + a_t, \ t = 0, 1, \ldots, n-1, \tag{13}$$

where $a_t$ denote other auxiliary detection scores defined in the original models. Then, the anomaly scores for each time point can be represented as $s_t^{\text{norm}}$ with normalized $s_t$ using MinMaxScaler. Finally, anomalies are detected using the threshold $\delta = \mu + 3\sigma$, where $\mu$ and $\sigma$ are the mean and standard deviation of $s_t^{\text{norm}}$, which can be shown as:

$$\text{Label}(s_t^{\text{norm}}) = \begin{cases} 1, & \text{if } s_t^{\text{norm}} > \delta \quad \text{(Abnormal)} \\ 0, & \text{if } s_t^{\text{norm}} \leq \delta \quad \text{(Normal)} \end{cases} \tag{14}$$

## 4.2 Result

### 4.2.1 Anomaly Detection w/ or w/o IGAD

To demonstrate the effectiveness of IGAD, we compare the performance of different methods on four real-world datasets. We calculate the improvement ratios from the perspectives of models ($\Delta_{\text{model}}$) and datasets ($\Delta_{\text{data}}$). The results in Tab.1 show that applying IGAD leads to noticeable improvements in performance. From the perspective of models, nearly 87% of the experiments (52 out of 60) show improvements, while more advanced performance is observed across the four real-world datasets at the dataset level. Surprisingly, it is found that CAE-M [71] achieves an average VUS-PR that is more than three times higher on dataset SMD after applying IGAD, and OmniAnomaly [50] even shows a ten-fold improvement on dataset SMAP. In addition, since varying degrees of enhancement are observed, Wilcoxon signed rank tests are conducted to verify statistical significance. More than 94% of the improved experiments (49 out of 52) show statistical significance. Meanwhile, we also observe limited drops in performance for some models when experiments are conducted on specific datasets, which leaves room for further investigation into the reasons behind these phenomena.

### 4.2.2 Balance of Robustness and Sensitivity

In this part, to verify the balance of robustness and sensitivity, we artificially introduce different weighted Gaussian noise into the original time series selected for testing. Specifically, we first sample a Gaussian noise from the standard normal distribution $\mathcal{N}(\mathbf{0}, \mathbf{I})$ with the same dimension as the testing data. Then the sampled noise is multiplied by a noise weight variation in [0.01, 0.05, 0.1, 0.15, 0.2] before being added to the time series for testing. In Fig.4, more cases are shown to further support these. In Fig.4(a), it can be observed that the model incorporated with IGAD not only shows higher VUS-PR than the one without IGAD, but also maintains stability and efficiency when meeting different weighted noise. In comparison, the model without IGAD shows a downward trend with higher noise weights. In Fig.4(b), the model with IGAD also maintains stability and outperforms when faced with more challenging tasks for weights of 15% and 20%.

### 4.2.3 Difference in Distributions of Abnormal Scores

According to the detection strategy shown as (14), an abnormal time instance at time $t$ should have a relatively higher anomaly score $s_t^{\text{norm}}$ to be detected as abnormal. Therefore, we further explore the

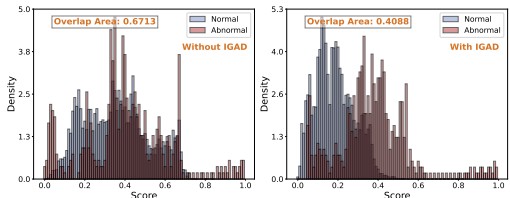
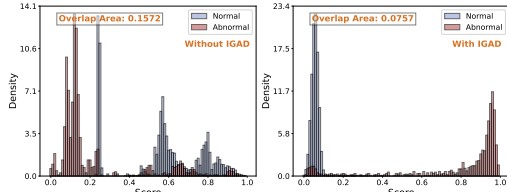

(a) Anomaly scores for CAE-M on SMD.  (b) Anomaly scores for OmniAnomaly on SMAP.

Figure 5: Anomaly score distributions for different models and datasets. In each sub-figure, the left one shows the distribution without IGAD, and the right one with IGAD.

Table 2: Results on different types of loss functions. † denotes the average value of all the mean values reported in Tab.1.

| Type | $\mathcal{L}_{\text{idem}}$ | $\mathcal{L}_{\text{tight}}$ | $\overline{\text{VUS-PR}}$ | $\Delta$ (%) |
|---|---|---|---|---|
| only $\mathcal{L}_1$ | $\mathcal{L}_1$ | $\mathcal{L}_1$ | **0.2080**† | N.A. |
| $\mathcal{L}_1 + \mathcal{L}_2$ | $\mathcal{L}_1$ | $\mathcal{L}_2$ | 0.1619 | -22.16 |
| $\mathcal{L}_2 + \mathcal{L}_1$ | $\mathcal{L}_2$ | $\mathcal{L}_1$ | 0.1857 | -10.72 |
| only $\mathcal{L}_2$ | $\mathcal{L}_2$ | $\mathcal{L}_2$ | 0.1684 | -19.04 |

$\mathcal{L}_1$ is the MAE Loss and $\mathcal{L}_2$ is the MSE Loss. We calculate $\overline{\text{VUS-PR}}$ as the average values of all experiments shown in Tab.1 (15 basic models and 4 datasets) for each combination of $\mathcal{L}_{\text{idem}}$ and $\mathcal{L}_{\text{tight}}$. This experimental strategy is also performed in Tab.3.

Table 3: Results on different operation subset. † denotes the average value of all the mean values reported in Tab.1.

| Component | | | $\overline{\text{VUS-PR}}$ | $\Delta$ (%) |
|---|---|---|---|---|
| $\mathcal{L}_{\text{idem}}$ | $\mathcal{L}_{\text{tight}}$ | $\mathcal{L}_{\text{tight}}^*$ | | |
| ✓ | ✓ | ✓ | **0.2080**† | N.A. |
| × | ✓ | ✓ | 0.1914 | -7.98 |
| ✓ | × | ✓ | Invalid | N.A. |
| ✓ | ✓ | × | 0.1822 | -12.40 |
| ✓ | × | × | 0.1704 | -18.08 |
| × | ✓ | × | 0.1803 | -13.32 |
| × | × | ✓ | Invalid | N.A. |
| × | × | × | 0.1684† | -19.04 |

distributions of the anomaly scores and calculate the areas of overlap in their corresponding density maps. In Fig.5, we show the results of two different models on two different datasets. The results demonstrate that after using IGAD, more distinguishable distributions of the anomaly scores can be observed for normal and abnormal time instances, as well as smaller overlap areas.

### 4.3 Ablation Study

#### 4.3.1 Different Loss Functions

Existing reconstruction-based MTS AD methods commonly use MSE Loss ($\mathcal{L}_2$) to measure the differences between the original and reconstructed time series due to its sensitivity to large errors, well-defined mathematical properties, and stability during optimization. However, we propose that MAE Loss ($\mathcal{L}_1$) is a more suitable choice for idempotent objective ($\mathcal{L}_{\text{idem}}$) and tightness objectives ($\mathcal{L}_{\text{tight}}$) because of its robustness to natural noise, which helps prevent over-tuning. We select different combinations of loss functions. The experiments summarized in Tab.2 reveal a consistent drop in performance when using MSE Loss, theoretically because its sensitivity to noise and randomness introduced by sampling artificially destabilize the training process.

#### 4.3.2 The Effectiveness of Each Objective

Here, we conduct an ablation study to assess the contribution of each component in our proposed module, specifically the effectiveness of the idempotent objective ($\mathcal{L}_{\text{idem}}$), tightness objective ($\mathcal{L}_{\text{tight}}$), and the smoothness operation of the tightness objective ($\mathcal{L}_{\text{tight}}^*$). The idempotent objective is designed to modify $\mathcal{M}_{\text{target}}$ to adequately capture normal instances. The tightness objective further refines $\mathcal{M}_{\text{target}}$ by tightening it as much as possible, excluding potential abnormal instances. Furthermore, ensuring the smoothness of the tightness objective helps stabilize the training process. As shown in Tab.3, the models that incorporate all three proposed components achieve the highest performance.

Table 4: VUS-PR on foundational models AE and VAE with different datasets.

| Model | Dataset | | | | | | | |
|---|---|---|---|---|---|---|---|---|
| | **SMD** | | | | **MSL** | | | |
| | w / o IGAD | w / IGAD | $\Delta_{model}$ (%) | $p$-value | w / o IGAD | w / IGAD | $\Delta_{model}$ (%) | $p$-value |
| AE | 0.3113±0.0029 | 0.3823±0.0053 | +22.81 | *** | 0.0083±0.0003 | 0.0370±0.0208 | +345.78 | *** |
| VAE | 0.3556±0.0041 | 0.3556±0.0037 | 0.00 | *** | 0.0086±0.0004 | 0.0087±0.0002 | +1.16 | *** |

| Model | Dataset | | | | | | | |
|---|---|---|---|---|---|---|---|---|
| | **PSM** | | | | **SMAP** | | | |
| | w / o IGAD | w / IGAD | $\Delta_{model}$ (%) | $p$-value | w / o IGAD | w / IGAD | $\Delta_{model}$ (%) | $p$-value |
| AE | 0.1437±0.0001 | 0.1640±0.0098 | +14.13 | *** | 0.0969±0.0007 | 0.9149±0.0784 | +844.17 | *** |
| VAE | 0.1464±0.0004 | 0.1465±0.0003 | +0.07 | *** | 0.0930±0.0014 | 0.0947±0.0014 | +1.83 | *** |

Figure 6: A comparison of different signal augmentation strategies. **Left:** Frequency Resampling Strategy in IGAD; **Middle:** Time-Domain PCA with $k = 5$; **Right:** Time-Domain PCA with $k = 10$.

## 4.4 The Effect of IGAD on Foundational Models

As indicated in [32], models with simpler architectures tend to yield better performance. Given this point, we perform further experiments to explore the effect of IGAD on foundational neural architectures, including AutoEncoders (AE) and Variational AutoEncoders (VAE), and the results are shown in Tab.4. The key findings here can be concluded as: (1) Before applying IGAD, VAE often shows higher levels than AE on the four selected datasets; (2) After applying IGAD, AE with IGAD shows higher improvements than VAE with IGAD, and even has better performance than VAE with IGAD; (3) AE and VAE achieve comparable results compared to certain complex models, which is consistent with the mentioned findings in [32].

## 4.5 Comparison between Frequency Resampling and Time-Domain PCA

FFT-based augmentation excels at creating diverse, globally consistent variations by perturbing the entire spectral profile, making it highly effective for dynamic time series. Conversely, time-domain PCA is confined to perturbing a few learned, high-variance linear patterns, which limits the diversity and fails to capture complex dynamics. Moreover, the fixed, data-independent basis of FFT makes it a lightweight, parameter-free module, avoiding the hyperparameter tuning for the number of principal components $k$ inherent to PCA. The results of comparison can be found in Fig.6.

## 5 Conclusion

This paper introduces IGAD, a novel module which can be easily integrated into reconstruction-based methods to enhance their effectiveness in detecting anomalies in MTS. Meanwhile, we conduct further experiments and explorations, addressing the issues of over generalization and overall performance balance of these models from the perspective of manifold. With defined optimization objectives, we aim to not only modify the target manifold, balancing the robustness and sensitivity to inherent normal patterns of time series, but also tighten the manifold to exclude potential abnormal time instances. The experimental results demonstrate the effectiveness of IGAD and show its great potential for further applications in MTS anomaly detection.

## Acknowledgements

This work was supported in part by Key R&D Program of Zhejiang Program No.2024C01065, in part by NSFC under grant No.U23A20326, and in part by State Key Laboratory of ICT Project No.ICT2025A09.

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

# Multivariate Time Series Anomaly Detection with Idempotent Reconstruction
## ————Appendix————

## Contents

# A Theoretical Analysis of *Over Generalization*

As we have discussed in Sect.1, we conclude the reasons for over generalization as that this problem may happen for two factors:

- A model incorrectly captures the intrinsic patterns of abnormal series in a contaminated dataset for training. This is **unavoidable** in real-world scenarios because large amounts of time series data are collected under the assumption that all time instances collected are normal points, since a system typically operates correctly under most conditions. However, it is inevitable that the system will have offsets or abnormal states over the long period of the data collection process. Meanwhile, since manually checking each time point for anomalies consumes a lot of human and time resources, these abnormal instances remain in the dataset for training without label scrutiny.
- A model has an excessive decoding power, even for abnormal series. This means that a model not only learns how to reconstruct normal time instances but also gains the ability to learn how to reconstruct abnormal instances.

In this part, we provide mathematical proofs and analysis for over generalization to explore how these two factors have an effect on the training process. Concretely, in Sect.A.1, we show the explanations in ideal and pure models, which have enough model capacity to learn everything. In addition, in Sect.A.2, we provide proofs and analysis in more general cases, and the effect of *regularization* is introduced, which aligns with the traditional principles of model design.

## A.1 Contaminated Data for Training and Over Expressive Models

Let $\mathcal{P}_x$ and $\mathcal{P}_a$ denote the distributions of normal and abnormal instances, respectively. When training data contains contaminated anomalies with rate $\eta \in [0, 1)$, the empirical distribution will be transformed to $\mathcal{P}_{\text{train}} = (1 - \eta)\mathcal{P}_x + \eta\mathcal{P}_a$. A reconstruction-based model $f : \mathcal{X} \to \mathcal{X}$ trained on the contaminated data aims to minimize the composite objective:

$$\mathcal{L}_{\text{recon}}(f(\cdot)) = (1 - \eta)\mathbb{E}_{x \sim \mathcal{P}_x}[\|x - f(x)\|_2^2] + \eta\mathbb{E}_{x \sim \mathcal{P}_a}[\|x - f(x)\|_2^2]. \tag{15}$$

Here, we consider a compact support $K \subset \mathcal{X}$ where $\mathbf{Supp}(\mathcal{P}_x) \cup \mathbf{Supp}(\mathcal{P}_a) \subseteq K$ and $\mathcal{P}_x, \mathcal{P}_a$ are absolutely continuous on $K$. By the universal approximation theorem [22], in a given assumption space $\mathcal{H}$, for any compact support $K \subset \mathcal{X}$ containing both normal and anomalous instances, there exists a neural network $f \in \mathcal{H}$, which can satisfy the following objective:

$$\sup_{x \in K} \|x - f(x)\|_2^2 \le \epsilon, \ \forall\epsilon > 0. \tag{16}$$

This implies that sufficiently expressive models can achieve arbitrarily small reconstruction errors on both distributions simultaneously. This theoretical capacity relies on the complexity of the unbounded model. Practical architectures with inductive biases, for example, the commonly selected bottleneck constraint, and other implicit regularization introduced during the design of the model alter the solution landscape.

## A.2 Optimization Dynamics with Regularization Constraint

For more general cases in practice, when $\eta > 0$, we consider an encoder-decoder-like mapping function $f_\theta(x) = g_\phi(h_\psi(x))$ with a bottleneck dimension sufficiently smaller than the data dimension, that is, $d_{\text{bottleneck}} \ll d_{\text{data}}$, which enforces information compression. This is a demonstrated strategy in this type of structures because it enforces the learning of compressed representations that retain the intrinsic structure of the data while discarding redundant information [21, 38], thus preventing trivial identity mappings [5] and promoting robust feature disentanglement [20]. In terms of Eq.(15), the optimization dynamics is governed by:

$$(1 - \eta)\nabla_\theta\mathbb{E}_x\left[\|x - f_\theta(x)\|_2^2\right] + \eta\nabla_\theta\mathbb{E}_{x'}\left[\|x' - f_\theta(x')\|_2^2\right] + \gamma\nabla_\theta\mathcal{R}(\theta) = 0, \tag{17}$$

where $\mathcal{R}(\theta)$ captures architectural constraints through implicit regularization and $\gamma$ quantifies the effective regularization strength from the bottleneck. To elucidate this equilibrium, more generally, we expanding the loss in the parameter space with $\mathcal{R}(\theta)$:

$$\mathcal{L}_{\text{recon}}(f_\theta) = \int_K \|x - f_\theta(x)\|_2^2\left[(1 - \eta)\mathcal{P}_x(x) + \eta\mathcal{P}_a(x)\right]dx + \gamma\mathcal{R}(\theta). \tag{18}$$

For reconstruction-based methods, $\mathcal{R}(\theta)$ can be introduced naturally to $\mathcal{R}(\theta) = \|f_\theta(x) - \mathbb{E}(x|\theta)\|_2^2$. Here, $\mathbb{E}[x|\theta] \triangleq \mathbb{E}_{z \sim p(z|\theta)}[g_\phi(z)]$ with $p(z|\theta)$ is the empirical distribution induced by the encoder with $z \triangleq h_\psi(x)$. This setting is justified by its alignment with the conservative estimation principle of the traditional design approach in the training process, that is, the model output is constrained to the typical pattern of the data itself, making it possible to reconstruct every time instance well. Crucially, since the training data are contaminated for $\eta > 0$, the empirical distribution $p(z|\theta)$ and $\mathbb{E}[x|\theta]$ are influenced by both normal and abnormal instances. To form the corresponding Euler-Lagrange equation, we take the functional derivative $\frac{\delta \mathcal{L}_{\text{recon}}(f_\theta)}{\delta f_\theta} = 0$ to get the expression:

$$[(1-\eta)\mathcal{P}_x(x) + \eta\mathcal{P}_a(x)](x - f_\theta(x)) = \gamma(f_\theta(x) - \mathbb{E}[x|\theta]), \tag{19}$$

We solve this elliptic equation under the strong bottleneck condition ($\gamma \gg \eta\mathcal{P}_a(x)$ and $\gamma \ll [(1-\eta)\mathcal{P}_x(x) + \eta\mathcal{P}_a(x)]$):

$$f_\theta^*(x) = \frac{[(1-\eta)\mathcal{P}_x(x) + \eta\mathcal{P}_a(x)]x + \gamma\mathbb{E}[x|\theta]}{(1-\eta)\mathcal{P}_x(x) + \eta\mathcal{P}_a(x) + \gamma} \tag{20}$$

$$= \mathbb{E}[x|\theta] + \frac{(1-\eta)\mathcal{P}_x(x) + \eta\mathcal{P}_a(x)}{(1-\eta)\mathcal{P}_x(x) + \eta\mathcal{P}_a(x) + \gamma}(x - \mathbb{E}[x|\theta]) \tag{21}$$

$$= \mathbb{E}[x|\theta] + \frac{1}{1 + \frac{\gamma}{(1-\eta)\mathcal{P}_x(x) + \eta\mathcal{P}_a(x)}}(x - \mathbb{E}[x|\theta]) \tag{22}$$

$$\approx \mathbb{E}[x|\theta] + \underbrace{\left(1 - \frac{\gamma}{(1-\eta)\mathcal{P}_x(x) + \eta\mathcal{P}_a(x)} + \mathcal{O}\left(\frac{\gamma^2}{[(1-\eta)\mathcal{P}_x(x) + \eta\mathcal{P}_a(x)]^2}\right)\right)}_{\text{Taylor Expansion}}(x - \mathbb{E}[x|\theta])$$

$$\tag{23}$$

$$= \mathbb{E}[x|\theta] + \left(1 - \frac{\gamma}{(1-\eta)\mathcal{P}_x(x) + \eta\mathcal{P}_a(x)}\right)(x - \mathbb{E}[x|\theta])$$
$$+ \mathcal{O}\left(\frac{\gamma^2}{[(1-\eta)\mathcal{P}_x(x) + \eta\mathcal{P}_a(x)]^2}\right) \cdot \frac{x - \mathbb{E}[x|\theta]}{\|x - \mathbb{E}[x|\theta]\|_2}. \tag{24}$$

### A.3   Mechanistic Interpretation of Over Generalization

The analytical decomposition from Eq.(20) to Eq.(24) reveals the fundamental mechanisms that govern the reconstruction behavior.

**Theorem 1** *When $\eta > 0$ and $\gamma > 0$, the optimal reconstruction function $f_\theta^*(x)$ satisfies the following expression, which can be acquired from Eq.(24):*

$$\|f_\theta^*(x) - x\|_2 \le \underbrace{\frac{\gamma}{A(x)}\|x - \mathbb{E}[x|\theta]\|_2}_{\text{Anomaly Suppression}} + \|\mathcal{O}\left(\frac{\gamma^2}{A(x)^2}\right) \cdot \underbrace{\zeta}_{\text{Unit Vector}}\|_2 \tag{25}$$

$$= \frac{\gamma}{A(x)}\|x - \mathbb{E}[x|\theta]\|_2 + \mathcal{O}\left(\frac{\gamma^2}{A(x)^2}\right) \tag{26}$$

*where $A(x) \triangleq (1-\eta)\mathcal{P}_x(x) + \eta\mathcal{P}_a(x)$ represents the local data density mixture at point $x$ and $\zeta \triangleq \frac{x - \mathbb{E}[x|\theta]}{\|x - \mathbb{E}[x|\theta]\|_2}$ is the term for direction correction.*

#### A.3.1   Density-Driven Error Scaling

The decomposition of the optimal reconstruction function $f_\theta^*(x)$ in Eq.(24) and Eq.(25) reveals a critical mechanism that governs over generalization. With the defined *mixed local density $A(x)$* and *regularization strength $\gamma$*, the primary error term $\frac{\gamma}{A(x)}$ exhibits an inverse proportionality to $A(x)$, leading to the following regimes:

- **High-Density Regions** ($\gamma \ll A(x)$):

$$\frac{\gamma}{A(x)} \to 0 \implies \|f_\theta^*(x) - x\|_2 \approx \mathcal{O}\left(\frac{\gamma^2}{A(x)^2}\right) \implies f_\theta^*(x) \approx x. \tag{27}$$

  Accurate reconstructions dominate as the density of normal instances suppresses abnormal residuals.

- **Low-Density Regions** ($\gamma \sim A(x)$):

$$\frac{\gamma}{A(x)} \approx 1 \implies \|f_\theta^*(x) - x\|_2 \approx \|\mathbb{E}[x|\theta] - x\|_2 + \mathcal{O}\left(\frac{\gamma^2}{A(x)^2}\right) \implies f_\theta^*(x) \approx \mathbb{E}[x|\theta].$$

(28)

The issue of over generalization may emerge as regularization forces reconstructions toward the latent manifold expectation.

### A.3.2 Latent Manifold Attraction

The defined $\mathbb{E}[x|\theta]$ encapsulates a dual mathematical role within our framework. **_Statistically_**, it is rigorously defined through the encoder-decoder architecture as $\mathbb{E}[x|\theta] \triangleq \mathbb{E}_{z \sim p(z|\theta)}[g_\phi(z)]$, where $p(z|\theta)$ represents the empirical latent distribution generated by the encoder $h_\psi(x)$. **_Geometrically_**, it also serves as the $L_2$ optimal projection anchor on the learned manifold $\mathcal{M}_{\text{target}}$, which satisfies the following objective:

$$\mathbb{E}[x|\theta] = \arg \min_{\tilde{x} \in \mathcal{M}_{\text{target}}} \mathbb{E}_{x \sim \mathcal{P}_x} \|\tilde{x} - x\|_2^2,$$

(29)

where $\mathcal{M}_{\text{target}} = \{g_\phi(z)\} = \{g_\phi(h_\psi(x))\}$ denotes the manifold induced by the decoder. This dual role establishes $\mathbb{E}[x|\theta]$ as an attractor, since both the statistical expectation of the decoder output and the geometric centroid minimize projection errors. These properties explain its ability to govern reconstruction behaviors while remaining sensitive to the underlying data density $\mathcal{P}_x(x)$, thus providing a unified perspective to analyze the expansion of the manifold under regularization constraints.

The learning process establishes a dynamic equilibrium between reconstruction fidelity and regularization forces, governed by the data density landscape. For normal samples $x \sim \mathcal{P}_x$, the model aims to preserve the accurate mapping:

$$f_\theta^*(x) \approx x \implies \|f_\theta(x) - x\|_2^2 \le \epsilon,$$

(30)

which preserves the geometric fidelity of normal instances on the manifold $\mathcal{M}_{\text{target}}$. Conversely, for anomalies $x_a \sim \mathcal{P}_a$, the regularization term enforces alignment with the latent manifold expectation:

$$f_\theta^*(x_a) \approx \mathbb{E}[x|\theta] \implies \|f_\theta(x_a) - \mathbb{E}[x|\theta]\|_2^2 \le \epsilon.$$

(31)

This competition induces a critical contamination threshold $\eta_{\text{crit}}$ defined by:

$$\eta_{\text{crit}} = \sup \left\{ \eta \in (0,1) \, \middle| \, \mathbb{E}_{x_a} \|x_a - f_\theta(x_a)\|_2^2 > \mathbb{E}_x \|x - f_\theta(x)\|_2^2 \right\}.$$

(32)

When $\eta > \eta_{\text{crit}}$, the expanded manifold $\mathcal{M}_{\text{target}}^* = \mathcal{M}_{\text{target}} \cup \{f_\theta(x_a)|x_a \sim \mathcal{P}_a\}$ exhibits dimensional inflation ($\dim(\mathcal{M}^*) > \dim(\mathcal{M}_{\text{target}})$), causing the anomaly-normal separability to collapse.

From this point, inspired by the manifold theorem, we propose IGAD, which can benefit both the balance of robustness and sensitivity and the tightness of the target manifold $\mathcal{M}_{\text{target}}$ to eliminate potential abnormal instances during training.

# B    Proof of the convergence for IGAD

**Theorem 2** *Under ideal conditions, IGAD can converge to the target distribution, which consists only of all normal time instances for a given dataset. For simplification, we select $x$ and $z$ for $x^i$ and $z^i$, respectively. We define the generated distribution, represented by $\mathcal{P}_\theta(y)$, as the PDF of $y$ when $y = f_\theta(z)$ and $z \sim \mathcal{P}_z$. Here, we only pay attention to the loss items relative, i.e. $\mathcal{L}_{\mathrm{recon}}$, $\mathcal{L}_{\mathrm{idem}}$ and $\mathcal{L}_{\mathrm{tight}}$. The final loss function can be divided into two parts:*

$$\mathcal{L}(\theta;\theta') = \underbrace{\lambda_{\mathrm{rec}}\mathcal{L}_{\mathrm{recon}}(\theta) + \lambda_{\mathrm{tight}}\mathcal{L}_{\mathrm{tight}}(\theta;\theta')}_{\mathcal{L}_{\mathrm{rt}}} + \lambda_{\mathrm{idem}}\mathcal{L}_{\mathrm{idem}}(\theta;\theta') \tag{33}$$

*We assume a large enough model capacity such that both terms can obtain a global minimum:*

$$\theta^* = \arg\min_\theta \mathcal{L}_{\mathrm{rt}}(\theta;\theta^*) = \arg\min_\theta \mathcal{L}_{\mathrm{idem}}(\theta;\theta^*) \tag{34}$$

*Then, $\exists \theta^* : \mathcal{P}_{\theta^*} = \mathcal{P}_x$ and for $\lambda_{\mathrm{idem}} = 1$, this is the only one possible $\mathcal{P}_{\theta^*}$.*

We first demonstrate the global minimum $\mathcal{L}_{\mathrm{rt}}$. After that, we further verify the global minimum for $\mathcal{L}_{\mathrm{idem}}$. For a given parameter $\theta$ in the parameter space $\Theta$ and an input $\mathcal{X}$, $\Phi_{\theta \in \Theta}(\mathcal{X})$ is used to calculate the differences between $f_\theta(\mathcal{X})$ and $\mathcal{X}$.

**Step 1: Global minimum of $\mathcal{L}_{\mathrm{rt}}$ given the current parameters $\theta^*$.**

$$\mathcal{L}_{\mathrm{rt}}(\theta;\theta^*) = \mathbb{E}_x\big[\mathcal{D}(f_\theta(x), x)\big] - \lambda_{\mathrm{tight}}\mathbb{E}_z\big[\mathcal{D}(f_\theta(f_{\theta^*}(z)), f_{\theta^*}(z))\big] \tag{35}$$

$$= \int \Phi_\theta(x)\mathcal{P}_x(x)dx - \lambda_{\mathrm{tight}} \int \Phi_\theta(f_{\theta^*}(z))\mathcal{P}_{\theta^*}(z)dz \tag{36}$$

Change variables: let $y := x$ for the left integral and $y := f_{\theta^*}(z)$ (a well-learned model should ensure that $x$ and $f_{\theta^*}(z)$ lie on the same $\mathcal{M}_{\mathrm{target}}$) for the right integral. Then Eq.(36) can be transformed into the following formular:

$$\mathcal{L}_{\mathrm{rt}}(\theta;\theta^*) = \int \Phi_\theta(y)\mathcal{P}_x(y)dy - \lambda_{\mathrm{tight}} \int \Phi_\theta(y)\mathcal{P}_{\theta^*}(y)dy \tag{37}$$

$$= \int \Phi_\theta(y) \underbrace{\Big(\boldsymbol{\mathcal{P}_x(y)} - \boldsymbol{\lambda}_{\mathrm{tight}}\boldsymbol{\mathcal{P}_{\theta^*}(y)}\Big)}_{\text{\color{blue}Regularization for Tightening the Manifold}} dy \tag{38}$$

Let $\mathbb{M} = \sup_{y_1,y_2}\mathcal{D}(y_1, y_2)$, where the supremum is taken over all possible pairs $y_1, y_2$. Since $\Phi_\theta$ is non-negative, the global minimum is achieved when:

$$\Phi_{\theta^*}(y) = \mathbb{M} \cdot \big[\mathbf{1}_{\{\mathcal{P}_x(y) < \lambda_{\mathrm{tight}}\mathcal{P}_{\theta^*}(y)\}}\big], \ \forall y \tag{39}$$

**Step 2: Global minimum of $\mathcal{L}_{\mathrm{idem}}$.**

$$\mathcal{L}_{\mathrm{idem}}(\theta, \theta^*) = \mathbb{E}_z\big[\mathcal{D}\left(f_{\theta^*}(f_\theta(z)), f_\theta(z)\right)\big] \tag{40}$$

$$= \mathbb{E}_z\big[\Phi_{\theta^*}(f_\theta(z))\big] \tag{41}$$

Substituting $\Phi_{\theta^*}$ from Eq.(39) and exchange the position of $\theta$ and $\theta^*$ because we check the minimum of the inner $f$ for $\mathcal{L}_{\mathrm{idem}}$, instead of the outer $f$ in $\mathcal{L}_{\mathrm{tight}}$:

$$\mathcal{L}_{\mathrm{idem}}(\theta;\theta^*) = \mathbb{M} \cdot \mathbb{E}_z\big[\mathbf{1}_{\{\mathcal{P}_x(y) < \lambda_{\mathrm{idem}}\mathcal{P}_\theta(y)\}}\big] \tag{42}$$

Taking $\arg\min_\theta$ of Eq.(42):

$$\theta^* = \mathbb{M} \cdot \arg\min_\theta \mathbb{E}_z\big[\mathbf{1}_{\{P_x(y) < \lambda_{\mathrm{idem}}P_\theta(y)\}}\big] \tag{43}$$

Given these operations, if $\mathcal{P}_{\theta^*} = \mathcal{P}_x$ and $\lambda_{\mathrm{idem}} \leq 1$, the loss value is 0. Specifically, for $\lambda_{\mathrm{idem}} = 1$, $\theta^* : \mathcal{P}_{\theta^*} = \mathcal{P}_x$ is the only minimizer because the total sum of the probability must be 1. In addition, any deviation where $P_\theta(y) < P_x(y)$ implies $\exists y$ with $P_\theta(y) > P_x(y)$, increasing the loss.

# C    Explanations for Basic Models

In our experiments, we select 15 basic reconstruction-based models with different structures to evaluate IGAD more comprehensively. We consider the selective strategy of these basic models from two perspectives. First, we select well-designed models specifically tailored for multivariate time series anomaly detection. In addition, we also take time series foundation models into consideration, which can be competitive candidates in various time series tasks, including reconstruction-based MTS AD. Here, we provide more detailed descriptions of the selected models for a better understanding of their structures.

## C.1    Models Designed for MTS AD

We include the following models designed for MTS AD in our experiments: CATCH [60], M2N2 [26], SARAD [10], Anomaly Transformer [62], FGANomaly [13], CAE-M [71], MTAD-GAT [72], MSCRED [70], OnimAnomaly [50] and DAGMM [75].

- CATCH [60]: CATCH framework introduces two key innovations for multivariate time series anomaly detection: (1) A frequency patching mechanism that partitions the frequency domain into fine-grained bands to better capture diverse subsequence anomalies, addressing the limitations of coarse-grained frequency analysis in existing methods; (2) A novel Channel Fusion Module with a dynamic correlation discovery mechanism that employs a bi-level optimization strategy to adaptively learn context-aware channel interactions, clustering relevant channels while mitigating noise from irrelevant ones through masked attention, effectively bridging the gap between channel-independent and channel-dependent approaches. The framework further enhances detection robustness through a dual-domain reconstruction objective based on time and frequency, and a novel point-aligned scoring mechanism that synergizes temporal and spectral anomalies, enabling superior performance in detecting both point and heterogeneous subsequence anomalies across varied real-world and synthetic scenarios.

- M2N2 [26]: M2N2 is a novel test-time adaptation framework for unsupervised time series anomaly detection to address the *new normal problem* caused by distribution shifts between training and test data. First, a trend estimation module using exponential moving averages to dynamically detrend input sequences, enabling adaptation to evolving data patterns while preserving underlying dynamics. Then, a self-supervised model update strategy that selectively updates parameters during inference using predicted normal instances, effectively learning new normal patterns while mitigating contamination from anomalies. The approach bridges test-time adaptation with time series anomaly detection through its dual mechanism of trend-aware normalization and confidence-based parameter adjustment, requiring neither access to training data nor additional supervision. By combining real-time trend adaptation with model fine-tuning on detrended sequences, the method demonstrates superior robustness to distribution shifts across diverse real-world benchmarks while maintaining computational efficiency suitable for streaming applications.

- SARAD [10]: SARAD is a novel approach for time series anomaly detection that integrates Spatial Association Reduction with data reconstruction via Transformer-based models. Its innovation lies in capturing both temporal and spatial dependencies within multivariate time series data, a challenge that previous methods addressed largely only from a temporal perspective. The key feature of SARAD is its dual focus on data reconstruction errors and progression reconstruction errors, where the latter focuses on spatial changes in anomaly propagation. SARAD leverages Multi-Head Self-Attention from Transformer layers to capture spatial relationships between features over time, and uses this information in conjunction with progression-based metrics to robustly detect anomalies. Unlike traditional models that may struggle with short-range anomalies or overlook spatially distributed anomalies, SARAD effectively identifies anomalies by observing how spatial associations evolve, even when the underlying data distribution is shifted.

- Anomaly Transformer [62]: Anomaly Transformer introduces a novel approach to time-series anomaly detection by leveraging a Transformer-based model and focusing on the concept of association discrepancy. This model incorporates a dual-branch mechanism within the Anomaly-Attention module, which enhances its ability to distinguish between nor-

mal and abnormal data points. The key innovation lies in the use of Association Discrepancy, measured through symmetrized Kullback–Leibler divergence, between the learned series association and a prior association. By employing a minimax strategy during optimization, the model minimizes the prior-association in the early phase while maximizing the association discrepancy in the later phase, ensuring a more robust distinction between normal and abnormal time points. This method improves the detection performance by forcing the model to focus more on non-adjacent time series data, thus enhancing its sensitivity to anomalies. The final anomaly score is a combination of reconstruction loss and association discrepancy, ensuring that both components contribute to detection, offering a more accurate and interpretable framework for time-series anomaly detection.

- FGANomaly [13]: The proposed model introduces a novel approach to anomaly detection by leveraging Generative Adversarial Networks in the context of multivariate time series data, with a particular focus on handling polluted or noisy training sets. The core innovation lies in the use of a GAN framework, where the generator learns to reconstruct normal time series, while the discriminator distinguishes between real and reconstructed data. Unlike traditional methods that may struggle with noisy or incomplete training data, this model introduces a specific mechanism to adapt the GAN training process to be robust to polluted data. It utilizes a data preprocessing strategy that filters out or reduces the impact of noisy segments, ensuring the model learns meaningful patterns from the time series. This unique approach enables the model to efficiently detect anomalies by leveraging the powerful generative capabilities of GANs while simultaneously addressing the challenges posed by noisy real-world time series data.

- CAE-M [71]: CAE-M addresses several challenges in multivariate time-series anomaly detection, particularly in the presence of noisy data. This proposed approach integrates a convolutional autoencoder for feature extraction, which captures spatial dependencies in multi-sensor time-series signals, with a memory network that combines both non-linear and linear prediction methods to capture temporal dependencies. The key innovation of CAE-M lies in the joint optimization of these components using a compound objective function, which simultaneously minimizes reconstruction error, prediction error, and a regularization term based on Maximum Mean Discrepancy (MMD). The MMD penalty is particularly crucial as it mitigates the influence of noisy data by encouraging the learned feature distribution to approximate that of a Gaussian distribution, thus reducing over-fitting. This architecture allows the model to effectively differentiate between normal and anomalous data even when the training set is polluted with noise.

- MTAD-GAT [72]: MTAD-GAT introduces a novel framework for anomaly detection in multivariate time series data by explicitly capturing the correlations between different features and timestamps. The unique structure of this model leverages two parallel Graph Attention Network (GAT) layers: one feature-oriented and one time-oriented. The feature-oriented GAT layer models the causal relationships between different time-series features, while the time-oriented GAT layer captures temporal dependencies within each time-series. This dual attention mechanism allows the model to dynamically learn both feature-wise and temporal dependencies. Furthermore, MTAD-GAT integrates both forecasting-based and reconstruction-based models, optimizing them through a joint objective function to enhance the representation of time-series data. The forecasting model focuses on single-timestamp predictions, while the reconstruction model learns a latent representation of the entire time-series, making the model robust against various anomaly types.

- MSCRED [70]: MSCRED introduces an effective approach for unsupervised anomaly detection and diagnosis in multivariate time series. The core innovation lies in its ability to jointly tackle three key tasks: anomaly detection, root cause identification, and anomaly severity interpretation. MSCRED achieves this by constructing multi-scale system signature matrices that represent the inter-correlations between time series at different temporal resolutions. These signature matrices are then processed through a fully convolutional encoder to capture spatial dependencies, while an attention-based Convolutional Long Short-Term Memory Network models the temporal dependencies across time steps. The decoder reconstructs these matrices, and the residuals are used to identify anomalies. This architecture is enhanced by its attention mechanism, which adaptively focuses on the most relevant historical time steps to improve anomaly detection.

- OnimAnomaly [50]: The proposed model introduces an innovative approach to anomaly detection by incorporating a Stochastic Recurrent Neural Network (SRNN) to model the temporal dependencies and capture the inherent uncertainty within multivariate time-series data. The key innovation of this model is the introduction of stochasticity in the recurrent network, where the model learns a distribution over the hidden states instead of a deterministic hidden representation. This probabilistic approach allows the SRNN to better handle incomplete data by explicitly modeling the uncertainty in the data generation process. The network is structured to combine both temporal and spatial dependencies by employing a combination of recurrent layers with stochastic units and a mixture of Gaussian distributions to represent uncertainty. Furthermore, the model includes a robust loss function that incorporates both reconstruction error and a regularization term based on the variance of the learned hidden states.

- DAGMM [75]: DAGMM introduces an architecture for unsupervised anomaly detection that combines the strengths of dimensionality reduction via a deep autoencoder with density estimation through a Gaussian Mixture Model (GMM). The key point of this approach is the joint optimization, where both the dimensionality reduction and the density estimation components are optimized simultaneously in an end-to-end manner, eliminating the need for pre-training and decoupled training. This architecture includes two main components: a compression network that reduces the dimensionality of input data and encodes it alongside the reconstruction error, and an estimation network that evaluates the likelihood of each data point within the GMM framework. This joint training, facilitated by the estimation network's regularization, allows the autoencoder to avoid suboptimal local minima and better capture the essential features of the data for anomaly detection.

## C.2 Time Series Foundation Model

Time series foundation models have shown their powerful potential for downstream time series tasks, including forecasting, imputation, classification, and also reconstruction-based anomaly detection. We select FITS [64], Peri-midFormer [59], ModernTCN [34], OFA [74], and TimesNet [58] in our experiments.

- FITS [64]: FITS introduces an innovative approach to time series analysis by operating within the complex frequency domain. It utilizes complex-valued linear interpolation to capture both amplitude and phase information, enabling the model to effectively learn amplitude scaling and phase shifting. This ability allows FITS to achieve state-of-the-art performance in tasks such as forecasting and anomaly detection. Despite its advanced capabilities, FITS maintains a remarkably compact architecture consisting of approximately 10,000 parameters, making it highly efficient. This compactness ensures that FITS is particularly well-suited for deployment on edge devices with limited computational resources, offering an excellent balance of performance and efficiency. By leveraging these innovative techniques, FITS demonstrates that high accuracy can be achieved in time series analysis without the need for large, resource-intensive models.

- Peri-midFormer [59]: Peri-midFormer presents an innovative transformer-based architecture that decomposes time series data into a periodic pyramid structure. This decomposition captures multi-periodic variations by representing the time series at multiple levels, each corresponding to different periodic components. The model employs self-attention mechanisms to effectively capture complex temporal relationships across these levels, enhancing its performance in tasks such as forecasting, imputation, classification, and anomaly detection.

- ModernTCN [34]: ModernTCN revitalizes convolutional approaches in time series analysis by introducing a pure convolutional structure that efficiently captures both cross-time and cross-variable dependencies. By incorporating large convolutional kernels and multiple convolutional layers, ModernTCN achieves substantial effective receptive fields, enabling it to model complex temporal patterns effectively. This design results in state-of-the-art performance across various time series tasks, including forecasting, imputation, classification, and anomaly detection.

- OFA [74]: One Fits All leverages pre-trained language models (LMs) to enhance time series analysis across multiple tasks. By fine-tuning these LMs on time series data, the model adapts the rich, generalized representations learned from large-scale textual data to the

specific characteristics of time series data. This approach demonstrates that pre-trained models from natural language or image domains can achieve comparable or even superior performance in time series tasks such as classification, forecasting, and anomaly detection, highlighting the versatility and power of pre-trained LMs in this context.

- TimesNet [58]: TimesNet introduces a task-general backbone for time series analysis by transforming 1D time series data into 2D tensors based on multiple periods. This transformation allows the model to capture both intraperiod and interperiod variations effectively. Utilizing a parameter-efficient inception block, TimesNet discovers multi-periodicity adaptively and extracts complex temporal variations from the transformed 2D tensors. This design enables TimesNet to achieve consistent state-of-the-art performance across five common time series analysis tasks, including short- and long-term forecasting, imputation, classification, and anomaly detection.

# D  Hyperparameter Setting

There are five crucial hyperparameters during our experiments, including $\lambda_{rec}$, $\lambda_{idem}$, $\lambda_{tight}$, $\lambda_{aux}$ and $\alpha$. **In the latest study conducted by Liu and Paparrizos [32], the pipeline has made sufficient explorations for the optimal $\lambda_{rec}$ and $\lambda_{aux}$, and these two hyperparameters are fixed when performing experiments.** Meanwhile, we also select certain reconstruction-based models that have not been temporarily imported into this pipeline. **For these models, we use the hyperparameters suggested in their original papers or repositories for $\lambda_{rec}$ and $\lambda_{aux}$.** For the hyperparameters introduced by IGAD, we perform a detailed grid search for $\lambda_{idem}$, $\lambda_{tight}$, and $\alpha$. The search intervals for each hyperparameter can be shown as:

- $\lambda_{idem}$: [0.1, 0.2, 0.3, 0.4, 0.5, 0.6, 0.7, 0.8, 0.9, 1.0],
- $\lambda_{tight}$: [0.1, 0.2, 0.3, 0.4, 0.5, 0.6, 0.7, 0.8, 0.9, 1.0],
- $\alpha$: [1.1, 1.2, 1.3, 1.4, 1.5].

Then, we summarize the optimal hyperparameters for each model and each dataset in Tab.5.

Table 5: Optimal hyperparameters after grid search in our experiments.

| Model | Venue | SMD | | | MSL | | | PSM | | | SMAP | | |
|---|---|---|---|---|---|---|---|---|---|---|---|---|---|
| | | $\lambda_{idem}$ | $\lambda_{tight}$ | $\alpha$ | $\lambda_{idem}$ | $\lambda_{tight}$ | $\alpha$ | $\lambda_{idem}$ | $\lambda_{tight}$ | $\alpha$ | $\lambda_{idem}$ | $\lambda_{tight}$ | $\alpha$ |
| CATCH | ICLR, 2025 | 0.5 | 0.9 | 1.1 | 0.1 | 0.2 | 1.5 | 0.3 | 0.5 | 1.4 | 1.0 | 0.5 | 1.5 |
| M2N2 | AAAI, 2024 | 0.5 | 0.4 | 1.2 | 0.1 | 0.5 | 1.4 | 0.1 | 0.3 | 1.5 | 0.1 | 1.0 | 1.3 |
| FITS | ICLR, 2024 | 0.5 | 0.1 | 1.1 | 1.0 | 1.0 | 1.5 | 0.3 | 1.0 | 1.2 | 0.2 | 0.3 | 1.5 |
| ModernTCN | ICLR, 2024 | 0.1 | 1.0 | 1.4 | 0.1 | 0.8 | 1.3 | 0.1 | 0.1 | 1.1 | 0.1 | 0.1 | 1.1 |
| Peri-midFormer | NeurIPS, 2024 | 0.1 | 1.0 | 1.5 | 0.5 | 0.4 | 1.4 | 0.8 | 0.9 | 1.4 | 0.1 | 0.1 | 1.3 |
| SARAD | NeurIPS, 2024 | 0.1 | 0.1 | 1.1 | 0.1 | 0.9 | 1.5 | 0.4 | 0.2 | 1.5 | 0.1 | 0.1 | 1.1 |
| TimesNet | ICLR, 2023 | 0.1 | 1.0 | 1.4 | 0.8 | 0.1 | 1.3 | 0.1 | 1.0 | 1.2 | 1.0 | 1.0 | 1.3 |
| OFA | NeurIPS, 2023 | 0.1 | 1.0 | 1.5 | 0.8 | 0.9 | 1.3 | 0.5 | 0.4 | 1.3 | 0.3 | 0.1 | 1.3 |
| A.T. | ICLR, 2022 | 0.9 | 1.0 | 1.4 | 1.0 | 0.2 | 1.1 | 0.8 | 0.7 | 1.4 | 0.7 | 0.8 | 1.3 |
| FGANomaly | TKDE, 2021 | 0.2 | 0.1 | 1.4 | 0.1 | 0.1 | 1.1 | 0.1 | 0.4 | 1.3 | 0.1 | 1.0 | 1.5 |
| CAE-M | TKDE, 2021 | 0.1 | 0.1 | 1.5 | 0.8 | 0.1 | 1.5 | 0.2 | 1.0 | 1.1 | 0.1 | 0.1 | 1.5 |
| MTAD-GAT | ICDM, 2021 | 0.8 | 0.8 | 1.1 | 0.2 | 0.3 | 1.3 | 0.1 | 0.6 | 1.4 | 0.9 | 1.0 | 1.4 |
| OmniAnomaly | KDD, 2019 | 0.2 | 0.3 | 1.5 | 0.1 | 0.8 | 1.3 | 0.9 | 0.7 | 1.5 | 0.2 | 0.5 | 1.5 |
| MSCRED | AAAI, 2019 | 0.1 | 0.5 | 1.4 | 0.4 | 0.4 | 1.5 | 0.1 | 0.1 | 1.1 | 0.3 | 0.9 | 1.5 |
| DAGMM | ICLR, 2018 | 0.2 | 1.0 | 1.1 | 0.8 | 0.4 | 1.4 | 0.2 | 0.1 | 1.1 | 0.1 | 0.9 | 1.5 |

# E   More Detailed Experimental Results

## E.1   Distinguishable Distributions of Anomaly Scores and Auxiliary Metrics

In this section, we will provide more experimental results for other detailed information. First, we show more cases where the application of IGAD effectively generates more distinguishable distributions of anomaly scores for normal and abnormal instances, shown as Fig.7(a) and Fig.7(b). Second, we list other evaluation metrics for classification tasks such as AUC-PR, AUC-ROC, VUS-ROC and different types of F1 from Tab.6 to Tab.13, which can still serve as auxiliary metrics although they exist potential evaluation shortcomings in the field of multivariate time series anomaly detection [32]. Improvements in these metrics can also be observed.

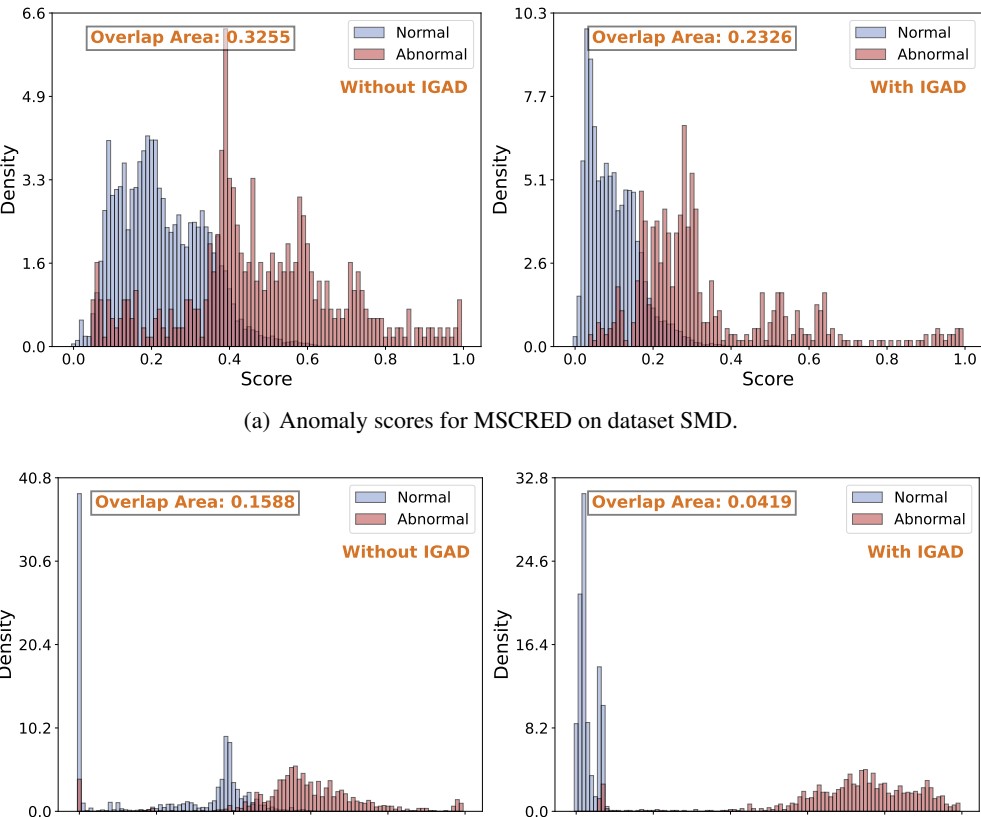

(a) Anomaly scores for MSCRED on dataset SMD.

(b) Anomaly scores for FGANomaly on dataset SMAP.

Figure 7: Anomaly score distributions for different models and datasets.

Table 6: More results on dataset SMD with IGAD.

| Model | AUC-PR | AUC-ROC | VUS-ROC | Standard-F1 |
|---|---|---|---|---|
| CATCH | $0.2341 \pm 0.0038$ | $0.8952 \pm 0.0019$ | $0.8343 \pm 0.0031$ | $0.1831 \pm 0.0041$ |
| M2N2 | $0.0407 \pm 0.0080$ | $0.6401 \pm 0.0772$ | $0.5537 \pm 0.0630$ | $0.0335 \pm 0.0308$ |
| FITS | $0.0564 \pm 0.0023$ | $0.7490 \pm 0.0122$ | $0.7300 \pm 0.0175$ | $0.0470 \pm 0.0200$ |
| ModernTCN | $0.2486 \pm 0.0011$ | $0.8966 \pm 0.0002$ | $0.8383 \pm 0.0004$ | $0.3063 \pm 0.0022$ |
| Peri-midFormer | $0.2026 \pm 0.0071$ | $0.8797 \pm 0.0037$ | $0.8209 \pm 0.0026$ | $0.1694 \pm 0.0055$ |
| SARAD | $0.2748 \pm 0.0096$ | $0.9092 \pm 0.0054$ | $0.8259 \pm 0.0089$ | $0.3155 \pm 0.0120$ |
| TimesNet | $0.0861 \pm 0.0347$ | $0.7553 \pm 0.0272$ | $0.7156 \pm 0.0298$ | $0.1303 \pm 0.0628$ |
| OFA | $0.1433 \pm 0.0254$ | $0.6935 \pm 0.0121$ | $0.6639 \pm 0.0101$ | $0.2220 \pm 0.0353$ |
| A.T. | $0.0679 \pm 0.0914$ | $0.5137 \pm 0.0763$ | $0.4960 \pm 0.0536$ | $0.0661 \pm 0.1288$ |
| FGANomaly | $0.4929 \pm 0.0116$ | $0.9324 \pm 0.0023$ | $0.8705 \pm 0.0065$ | $0.4943 \pm 0.0308$ |
| CAE-M | $0.1687 \pm 0.1294$ | $0.6635 \pm 0.1191$ | $0.5314 \pm 0.1537$ | $0.1326 \pm 0.1118$ |
| MTAD-GAT | $0.4562 \pm 0.0379$ | $0.8911 \pm 0.0239$ | $0.8886 \pm 0.0227$ | $0.5304 \pm 0.0193$ |
| OmniAnomaly | $0.2621 \pm 0.0026$ | $0.9052 \pm 0.0031$ | $0.8274 \pm 0.0024$ | $0.2830 \pm 0.0029$ |
| MSCRED | $0.4377 \pm 0.0217$ | $0.9216 \pm 0.0331$ | $0.8567 \pm 0.0329$ | $0.3939 \pm 0.0093$ |
| DAGMM | $0.2285 \pm 0.0684$ | $0.6875 \pm 0.0275$ | $0.4962 \pm 0.0401$ | $0.1449 \pm 0.1425$ |

| Model | PA-F1 | Event-based-F1 | R-based-F1 | Affiliation-F1 |
|---|---|---|---|---|
| CATCH | $0.4055 \pm 0.1009$ | $0.2638 \pm 0.0186$ | $0.0056 \pm 0.0000$ | $0.6418 \pm 0.0039$ |
| M2N2 | $0.2570 \pm 0.2359$ | $0.0666 \pm 0.0626$ | $0.0027 \pm 0.0024$ | $0.4552 \pm 0.1265$ |
| FITS | $0.5331 \pm 0.0347$ | $0.0938 \pm 0.0333$ | $0.0046 \pm 0.0000$ | $0.7750 \pm 0.0177$ |
| ModernTCN | $0.4886 \pm 0.0004$ | $0.3846 \pm 0.0015$ | $0.0056 \pm 0.0000$ | $0.6524 \pm 0.0001$ |
| Peri-midFormer | $0.4670 \pm 0.1321$ | $0.2529 \pm 0.0294$ | $0.0056 \pm 0.0000$ | $0.6198 \pm 0.0720$ |
| SARAD | $0.4912 \pm 0.0114$ | $0.3920 \pm 0.0141$ | $0.0056 \pm 0.0000$ | $0.7231 \pm 0.0034$ |
| TimesNet | $0.6006 \pm 0.0614$ | $0.2222 \pm 0.0866$ | $0.0047 \pm 0.0000$ | $0.8155 \pm 0.0287$ |
| OFA | $0.7442 \pm 0.0211$ | $0.3747 \pm 0.0296$ | $0.0050 \pm 0.0002$ | $0.8561 \pm 0.0285$ |
| A.T. | $0.3166 \pm 0.4273$ | $0.2140 \pm 0.3299$ | $0.0033 \pm 0.0031$ | $0.5968 \pm 0.3153$ |
| FGANomaly | $0.6531 \pm 0.0362$ | $0.6095 \pm 0.0258$ | $0.0056 \pm 0.0000$ | $0.7839 \pm 0.0392$ |
| CAE-M | $0.3069 \pm 0.1643$ | $0.2520 \pm 0.1600$ | $0.0313 \pm 0.0146$ | $0.2563 \pm 0.1742$ |
| MTAD-GAT | $0.8913 \pm 0.0249$ | $0.7379 \pm 0.0282$ | $0.0047 \pm 0.0003$ | $0.9257 \pm 0.0210$ |
| OmniAnomaly | $0.5340 \pm 0.0336$ | $0.3775 \pm 0.0115$ | $0.0056 \pm 0.0000$ | $0.7751 \pm 0.0282$ |
| MSCRED | $0.6960 \pm 0.0402$ | $0.6109 \pm 0.0312$ | $0.0056 \pm 0.0001$ | $0.7587 \pm 0.0329$ |
| DAGMM | $0.3319 \pm 0.2084$ | $0.3286 \pm 0.1672$ | $0.0106 \pm 0.0067$ | $0.3648 \pm 0.1950$ |

Table 7: More results on dataset SMD without IGAD.

| Model | AUC-PR | AUC-ROC | VUS-ROC | Standard-F1 |
|---|---|---|---|---|
| CATCH | $0.2272 \pm 0.0053$ | $0.8891 \pm 0.0014$ | $0.8222 \pm 0.0019$ | $0.1811 \pm 0.0031$ |
| M2N2 | $0.0234 \pm 0.0003$ | $0.4342 \pm 0.0034$ | $0.3623 \pm 0.0068$ | $0.0487 \pm 0.0000$ |
| FITS | $0.0419 \pm 0.0025$ | $0.6946 \pm 0.0192$ | $0.6769 \pm 0.0220$ | $0.0356 \pm 0.0076$ |
| ModernTCN | $0.1930 \pm 0.0022$ | $0.8792 \pm 0.0012$ | $0.8188 \pm 0.0020$ | $0.1676 \pm 0.0020$ |
| Peri-midFormer | $0.2004 \pm 0.0063$ | $0.8782 \pm 0.0036$ | $0.8198 \pm 0.0028$ | $0.1680 \pm 0.0042$ |
| SARAD | $0.2766 \pm 0.0128$ | $0.9103 \pm 0.0069$ | $0.8264 \pm 0.0113$ | $0.3134 \pm 0.0104$ |
| TimesNet | $0.0824 \pm 0.0188$ | $0.7583 \pm 0.0170$ | $0.7147 \pm 0.0216$ | $0.1294 \pm 0.0373$ |
| OFA | $0.0630 \pm 0.0030$ | $0.6887 \pm 0.0068$ | $0.6778 \pm 0.0072$ | $0.1178 \pm 0.0112$ |
| A.T. | $0.0266 \pm 0.0053$ | $0.5213 \pm 0.0293$ | $0.5175 \pm 0.0240$ | $0.0037 \pm 0.0082$ |
| FGANomaly | $0.4943 \pm 0.0063$ | $0.9320 \pm 0.0071$ | $0.8790 \pm 0.0143$ | $0.4663 \pm 0.0094$ |
| CAE-M | $0.1746 \pm 0.1266$ | $0.6676 \pm 0.1213$ | $0.5360 \pm 0.1568$ | $0.1268 \pm 0.1049$ |
| MTAD-GAT | $0.3949 \pm 0.0031$ | $0.8710 \pm 0.0043$ | $0.8700 \pm 0.0043$ | $0.5150 \pm 0.0048$ |
| OmniAnomaly | $0.2578 \pm 0.0011$ | $0.9003 \pm 0.0025$ | $0.8231 \pm 0.0020$ | $0.2798 \pm 0.0017$ |
| MSCRED | $0.4373 \pm 0.0293$ | $0.9228 \pm 0.0326$ | $0.8517 \pm 0.0271$ | $0.3878 \pm 0.0340$ |
| DAGMM | $0.0983 \pm 0.0067$ | $0.5554 \pm 0.0136$ | $0.4055 \pm 0.0109$ | $0.1041 \pm 0.0080$ |

| Model | PA-F1 | Event-based-F1 | R-based-F1 | Affiliation-F1 |
|---|---|---|---|---|
| CATCH | $0.3289 \pm 0.0441$ | $0.2380 \pm 0.0248$ | $0.0056 \pm 0.0000$ | $0.6416 \pm 0.0018$ |
| M2N2 | $0.3832 \pm 0.0000$ | $0.0851 \pm 0.0000$ | $0.0044 \pm 0.0000$ | $0.5238 \pm 0.0038$ |
| FITS | $0.4901 \pm 0.0580$ | $0.0768 \pm 0.0153$ | $0.0046 \pm 0.0000$ | $0.7382 \pm 0.0113$ |
| ModernTCN | $0.3416 \pm 0.0231$ | $0.2309 \pm 0.0115$ | $0.0056 \pm 0.0000$ | $0.6316 \pm 0.0002$ |
| Peri-midFormer | $0.5292 \pm 0.0042$ | $0.2522 \pm 0.0027$ | $0.0056 \pm 0.0000$ | $0.6411 \pm 0.0023$ |
| SARAD | $0.4905 \pm 0.0130$ | $0.3904 \pm 0.0142$ | $0.0056 \pm 0.0000$ | $0.7315 \pm 0.0249$ |
| TimesNet | $0.6230 \pm 0.0434$ | $0.2284 \pm 0.0559$ | $0.0047 \pm 0.0000$ | $0.8128 \pm 0.0259$ |
| OFA | $0.6813 \pm 0.0316$ | $0.2243 \pm 0.0241$ | $0.0045 \pm 0.0000$ | $0.8178 \pm 0.0129$ |
| A.T. | $0.1257 \pm 0.2811$ | $0.0182 \pm 0.0407$ | $0.0009 \pm 0.0020$ | $0.3300 \pm 0.2620$ |
| FGANomaly | $0.5990 \pm 0.0089$ | $0.5588 \pm 0.0201$ | $0.0056 \pm 0.0000$ | $0.7236 \pm 0.0351$ |
| CAE-M | $0.3072 \pm 0.1654$ | $0.2522 \pm 0.1607$ | $0.0311 \pm 0.0147$ | $0.2562 \pm 0.1743$ |
| MTAD-GAT | $0.8556 \pm 0.0044$ | $0.7075 \pm 0.0070$ | $0.0046 \pm 0.0000$ | $0.9264 \pm 0.0007$ |
| OmniAnomaly | $1.0000 \pm 0.0000$ | $1.0000 \pm 0.0000$ | $0.3893 \pm 0.0526$ | $0.9961 \pm 0.0011$ |
| MSCRED | $0.6846 \pm 0.0686$ | $0.5927 \pm 0.0695$ | $0.0056 \pm 0.0000$ | $0.7394 \pm 0.0799$ |
| DAGMM | $0.2317 \pm 0.0012$ | $0.1778 \pm 0.0014$ | $0.0406 \pm 0.0008$ | $0.1789 \pm 0.0003$ |

Table 8: More results on dataset MSL with IGAD.

| Model | AUC-PR | AUC-ROC | VUS-ROC | Standard-F1 |
|---|---|---|---|---|
| CATCH | $0.0286 \pm 0.0007$ | $0.7808 \pm 0.0045$ | $0.7808 \pm 0.0045$ | $0.0000 \pm 0.0000$ |
| M2N2 | $0.8250 \pm 0.2050$ | $0.9984 \pm 0.0030$ | $0.9984 \pm 0.0030$ | $0.0000 \pm 0.0000$ |
| FITS | $0.0103 \pm 0.0038$ | $0.5104 \pm 0.0508$ | $0.5105 \pm 0.0508$ | $0.0000 \pm 0.0000$ |
| ModernTCN | $0.0634 \pm 0.0424$ | $0.8082 \pm 0.0182$ | $0.8085 \pm 0.0181$ | $0.0718 \pm 0.0306$ |
| Peri-midFormer | $0.0346 \pm 0.0015$ | $0.8160 \pm 0.0252$ | $0.8160 \pm 0.0252$ | $0.0168 \pm 0.0233$ |
| SARAD | $0.0552 \pm 0.0635$ | $0.6906 \pm 0.0499$ | $0.6908 \pm 0.0497$ | $0.0785 \pm 0.1079$ |
| TimesNet | $0.0074 \pm 0.0030$ | $0.4316 \pm 0.0824$ | $0.4316 \pm 0.0826$ | $0.0000 \pm 0.0000$ |
| OFA | $0.0114 \pm 0.0112$ | $0.5077 \pm 0.0696$ | $0.5077 \pm 0.0696$ | $0.0200 \pm 0.0447$ |
| A.T. | $0.0063 \pm 0.0007$ | $0.5069 \pm 0.0154$ | $0.5068 \pm 0.0152$ | $0.0000 \pm 0.0000$ |
| FGANomaly | $0.0525 \pm 0.0103$ | $0.8315 \pm 0.0532$ | $0.8314 \pm 0.0531$ | $0.0986 \pm 0.0577$ |
| CAE-M | $0.0041 \pm 0.0001$ | $0.2066 \pm 0.0074$ | $0.2067 \pm 0.0072$ | $0.0000 \pm 0.0000$ |
| MTAD-GAT | $0.3401 \pm 0.1288$ | $0.7066 \pm 0.0655$ | $0.7063 \pm 0.0656$ | $0.3037 \pm 0.0928$ |
| OmniAnomaly | $0.0088 \pm 0.0029$ | $0.5024 \pm 0.0864$ | $0.5025 \pm 0.0862$ | $0.0000 \pm 0.0000$ |
| MSCRED | $0.0099 \pm 0.0011$ | $0.6883 \pm 0.0361$ | $0.6883 \pm 0.0363$ | $0.0000 \pm 0.0000$ |
| DAGMM | $0.0072 \pm 0.0029$ | $0.2631 \pm 0.0669$ | $0.2630 \pm 0.0666$ | $0.0000 \pm 0.0000$ |

| Model | PA-F1 | Event-based-F1 | R-based-F1 | Affiliation-F1 |
|---|---|---|---|---|
| CATCH | $0.0000 \pm 0.0000$ | $0.0000 \pm 0.0000$ | $0.0000 \pm 0.0000$ | $0.9332 \pm 0.0038$ |
| M2N2 | $0.0000 \pm 0.0000$ | $0.0000 \pm 0.0000$ | $0.0000 \pm 0.0000$ | $\text{Nan} \pm \text{NaN}$ |
| FITS | $0.0000 \pm 0.0000$ | $0.0000 \pm 0.0000$ | $0.0000 \pm 0.0000$ | $0.7619 \pm 0.1013$ |
| ModernTCN | $0.2940 \pm 0.0347$ | $0.0822 \pm 0.0336$ | $0.0210 \pm 0.0001$ | $0.9703 \pm 0.0013$ |
| Peri-midFormer | $0.1529 \pm 0.2106$ | $0.0214 \pm 0.0298$ | $0.0083 \pm 0.0114$ | $0.9682 \pm 0.0017$ |
| SARAD | $0.2991 \pm 0.4104$ | $0.1415 \pm 0.1965$ | $0.0083 \pm 0.0114$ | $0.9389 \pm 0.0360$ |
| TimesNet | $0.0000 \pm 0.0000$ | $0.0000 \pm 0.0000$ | $0.0000 \pm 0.0000$ | $0.8442 \pm 0.0523$ |
| OFA | $0.1467 \pm 0.3280$ | $0.0400 \pm 0.0894$ | $0.0032 \pm 0.0072$ | $0.8031 \pm 0.0757$ |
| A.T. | $0.0000 \pm 0.0000$ | $0.0000 \pm 0.0000$ | $0.0000 \pm 0.0000$ | $0.9499 \pm \text{NaN}$ |
| FGANomaly | $0.2739 \pm 0.1532$ | $0.1135 \pm 0.0660$ | $0.0177 \pm 0.0099$ | $0.9842 \pm 0.0018$ |
| CAE-M | $0.0000 \pm 0.0000$ | $0.0000 \pm 0.0000$ | $0.0000 \pm 0.0000$ | $\text{NaN} \pm \text{NaN}$ |
| MTAD-GAT | $1.0000 \pm 0.0000$ | $1.0000 \pm 0.0000$ | $0.0119 \pm 0.0019$ | $0.9990 \pm 0.0003$ |
| OmniAnomaly | $0.0000 \pm 0.0000$ | $0.0000 \pm 0.0000$ | $0.0000 \pm 0.0000$ | $0.6452 \pm 0.1220$ |
| MSCRED | $0.0000 \pm 0.0000$ | $0.0000 \pm 0.0000$ | $0.0000 \pm 0.0000$ | $0.3804 \pm \text{NaN}$ |
| DAGMM | $0.0000 \pm 0.0000$ | $0.0000 \pm 0.0000$ | $0.0000 \pm 0.0000$ | $0.4275 \pm 0.0554$ |

Table 9: More results on dataset MSL without IGAD.

| Model | AUC-PR | AUC-ROC | VUS-ROC | Standard-F1 |
|---|---|---|---|---|
| CATCH | $0.0286 \pm 0.0008$ | $0.7807 \pm 0.0052$ | $0.7806 \pm 0.0052$ | $0.0000 \pm 0.0000$ |
| M2N2 | $0.2024 \pm 0.1532$ | $0.9637 \pm 0.0233$ | $0.9637 \pm 0.0233$ | $0.0000 \pm 0.0000$ |
| FITS | $0.0419 \pm 0.0025$ | $0.6946 \pm 0.0192$ | $0.6769 \pm 0.0220$ | $0.0356 \pm 0.0076$ |
| ModernTCN | $0.1930 \pm 0.0022$ | $0.8792 \pm 0.0012$ | $0.8188 \pm 0.0020$ | $0.1676 \pm 0.0020$ |
| Peri-midFormer | $0.2004 \pm 0.0063$ | $0.8782 \pm 0.0036$ | $0.8198 \pm 0.0028$ | $0.1680 \pm 0.0042$ |
| SARAD | $0.2766 \pm 0.0128$ | $0.9103 \pm 0.0069$ | $0.8264 \pm 0.0113$ | $0.3134 \pm 0.0104$ |
| TimesNet | $0.0824 \pm 0.0188$ | $0.7583 \pm 0.0170$ | $0.7147 \pm 0.0216$ | $0.1294 \pm 0.0373$ |
| OFA | $0.0630 \pm 0.0030$ | $0.6887 \pm 0.0068$ | $0.6778 \pm 0.0072$ | $0.1178 \pm 0.0112$ |
| A.T. | $0.0266 \pm 0.0053$ | $0.5213 \pm 0.0293$ | $0.5175 \pm 0.0240$ | $0.0037 \pm 0.0082$ |
| FGANomaly | $0.4943 \pm 0.0063$ | $0.9320 \pm 0.0071$ | $0.8790 \pm 0.0143$ | $0.4663 \pm 0.0094$ |
| CAE-M | $0.1746 \pm 0.1266$ | $0.6676 \pm 0.1213$ | $0.5360 \pm 0.1568$ | $0.1268 \pm 0.1049$ |
| MTAD-GAT | $0.3949 \pm 0.0031$ | $0.8710 \pm 0.0043$ | $0.8700 \pm 0.0043$ | $0.5150 \pm 0.0048$ |
| OmniAnomaly | $0.2578 \pm 0.0011$ | $0.9003 \pm 0.0025$ | $0.8231 \pm 0.0020$ | $0.2798 \pm 0.0017$ |
| MSCRED | $0.4373 \pm 0.0293$ | $0.9228 \pm 0.0326$ | $0.8517 \pm 0.0271$ | $0.3878 \pm 0.0340$ |
| DAGMM | $0.0983 \pm 0.0067$ | $0.5554 \pm 0.0136$ | $0.4055 \pm 0.0109$ | $0.1041 \pm 0.0080$ |

| Model | PA-F1 | Event-based-F1 | R-based-F1 | Affiliation-F1 |
|---|---|---|---|---|
| CATCH | $0.0000 \pm 0.0000$ | $0.0000 \pm 0.0000$ | $0.0000 \pm 0.0000$ | $0.9332 \pm 0.0040$ |
| M2N2 | $0.0000 \pm 0.0000$ | $0.0000 \pm 0.0000$ | $0.0000 \pm 0.0000$ | $\mathrm{Nan} \pm \mathrm{NaN}$ |
| FITS | $0.4901 \pm 0.0580$ | $0.0768 \pm 0.0153$ | $0.0046 \pm 0.0000$ | $0.7382 \pm 0.0113$ |
| ModernTCN | $0.3416 \pm 0.0231$ | $0.2309 \pm 0.0115$ | $0.0056 \pm 0.0000$ | $0.6316 \pm 0.0002$ |
| Peri-midFormer | $0.5292 \pm 0.0042$ | $0.2522 \pm 0.0027$ | $0.0056 \pm 0.0000$ | $0.6411 \pm 0.0023$ |
| SARAD | $0.4905 \pm 0.0130$ | $0.3904 \pm 0.0142$ | $0.0056 \pm 0.0000$ | $0.7315 \pm 0.0249$ |
| TimesNet | $0.6230 \pm 0.0434$ | $0.2284 \pm 0.0559$ | $0.0047 \pm 0.0000$ | $0.8128 \pm 0.0259$ |
| OFA | $0.6813 \pm 0.0316$ | $0.2243 \pm 0.0241$ | $0.0045 \pm 0.0000$ | $0.8178 \pm 0.0129$ |
| A.T. | $0.1257 \pm 0.2811$ | $0.0182 \pm 0.0407$ | $0.0009 \pm 0.0020$ | $0.3300 \pm 0.2620$ |
| FGANomaly | $0.5990 \pm 0.0089$ | $0.5588 \pm 0.0201$ | $0.0056 \pm 0.0000$ | $0.7236 \pm 0.0351$ |
| CAE-M | $0.3072 \pm 0.1654$ | $0.2522 \pm 0.1607$ | $0.0311 \pm 0.0147$ | $0.2562 \pm 0.1743$ |
| MTAD-GAT | $0.8556 \pm 0.0044$ | $0.7075 \pm 0.0070$ | $0.0046 \pm 0.0000$ | $0.9264 \pm 0.0007$ |
| OmniAnomaly | $1.0000 \pm 0.0000$ | $1.0000 \pm 0.0000$ | $0.3893 \pm 0.0526$ | $0.9961 \pm 0.0011$ |
| MSCRED | $0.6846 \pm 0.0686$ | $0.5927 \pm 0.0695$ | $0.0056 \pm 0.0000$ | $0.7394 \pm 0.0799$ |
| DAGMM | $0.2317 \pm 0.0012$ | $0.1778 \pm 0.0014$ | $0.0406 \pm 0.0008$ | $0.1789 \pm 0.0003$ |

Table 10: More results on dataset PSM with IGAD.

| Model | AUC-PR | AUC-ROC | VUS-ROC | Standard-F1 |
|---|---|---|---|---|
| CATCH | $0.1380 \pm 0.0030$ | $0.5680 \pm 0.0102$ | $0.4974 \pm 0.0099$ | $0.0257 \pm 0.0027$ |
| M2N2 | $0.4124 \pm 0.0252$ | $0.8527 \pm 0.0024$ | $0.7793 \pm 0.0063$ | $0.0548 \pm 0.0272$ |
| FITS | $0.1203 \pm 0.0005$ | $0.5137 \pm 0.0020$ | $0.4559 \pm 0.0104$ | $0.0144 \pm 0.0006$ |
| ModernTCN | $0.1408 \pm 0.0002$ | $0.5659 \pm 0.0004$ | $0.4938 \pm 0.0005$ | $0.0291 \pm 0.0001$ |
| Peri-midFormer | $0.1378 \pm 0.0003$ | $0.5520 \pm 0.0009$ | $0.4909 \pm 0.0035$ | $0.0338 \pm 0.0005$ |
| SARAD | $0.1787 \pm 0.0095$ | $0.6627 \pm 0.0186$ | $0.4476 \pm 0.0155$ | $0.0310 \pm 0.0021$ |
| TimesNet | $0.1355 \pm 0.0076$ | $0.5516 \pm 0.0151$ | $0.4839 \pm 0.0141$ | $0.0259 \pm 0.0070$ |
| OFA | $0.1490 \pm 0.0108$ | $0.5700 \pm 0.0180$ | $0.5147 \pm 0.0208$ | $0.0145 \pm 0.0057$ |
| A.T. | $0.1809 \pm 0.0739$ | $0.6199 \pm 0.0833$ | $0.5028 \pm 0.0635$ | $0.1100 \pm 0.1421$ |
| FGANomaly | $0.2401 \pm 0.0235$ | $0.7214 \pm 0.0110$ | $0.5769 \pm 0.0106$ | $0.0067 \pm 0.0055$ |
| CAE-M | $0.1773 \pm 0.0376$ | $0.6427 \pm 0.0402$ | $0.4493 \pm 0.0245$ | $0.0275 \pm 0.0117$ |
| MTAD-GAT | $0.1904 \pm 0.0509$ | $0.6783 \pm 0.0665$ | $0.6099 \pm 0.0623$ | $0.0284 \pm 0.0017$ |
| OmniAnomaly | $0.1659 \pm 0.0006$ | $0.6233 \pm 0.0037$ | $0.4324 \pm 0.0073$ | $0.0257 \pm 0.0001$ |
| MSCRED | $0.1928 \pm 0.0246$ | $0.7115 \pm 0.0335$ | $0.5034 \pm 0.0550$ | $0.0220 \pm 0.0124$ |
| DAGMM | $0.2116 \pm 0.0047$ | $0.6913 \pm 0.0045$ | $0.4736 \pm 0.0070$ | $0.0375 \pm 0.0050$ |

| Model | PA-F1 | Event-based-F1 | R-based-F1 | Affiliation-F1 |
|---|---|---|---|---|
| CATCH | $0.3629 \pm 0.0010$ | $0.0990 \pm 0.0084$ | $0.0032 \pm 0.0000$ | $0.1300 \pm 0.0100$ |
| M2N2 | $0.3320 \pm 0.0000$ | $0.0274 \pm 0.0000$ | $0.0069 \pm 0.0010$ | $0.0215 \pm 0.0011$ |
| FITS | $0.7490 \pm 0.0155$ | $0.1197 \pm 0.0086$ | $0.0030 \pm 0.0000$ | $0.3014 \pm 0.0270$ |
| ModernTCN | $0.3624 \pm 0.0001$ | $0.1077 \pm 0.0002$ | $0.0032 \pm 0.0000$ | $0.1360 \pm 0.0000$ |
| Peri-midFormer | $0.3640 \pm 0.0001$ | $0.1300 \pm 0.0068$ | $0.0032 \pm 0.0000$ | $0.1686 \pm 0.0116$ |
| SARAD | $0.3313 \pm 0.0039$ | $0.0710 \pm 0.0018$ | $0.0032 \pm 0.0000$ | $0.1130 \pm 0.0015$ |
| TimesNet | $0.8278 \pm 0.0434$ | $0.1563 \pm 0.0212$ | $0.0031 \pm 0.0001$ | $0.3117 \pm 0.0466$ |
| OFA | $0.8248 \pm 0.0810$ | $0.1879 \pm 0.0471$ | $0.0031 \pm 0.0000$ | $0.3150 \pm 0.0950$ |
| A.T. | $0.3841 \pm 0.2627$ | $0.1285 \pm 0.1321$ | $0.0023 \pm 0.0013$ | $0.3753 \pm 0.3316$ |
| FGANomaly | $0.2710 \pm 0.1519$ | $0.0360 \pm 0.0244$ | $0.0031 \pm 0.0018$ | $0.0634 \pm 0.0258$ |
| CAE-M | $0.2783 \pm 0.1384$ | $0.0730 \pm 0.0308$ | $0.0032 \pm 0.0001$ | $0.1079 \pm 0.0242$ |
| MTAD-GAT | $0.7170 \pm 0.0350$ | $0.1424 \pm 0.0388$ | $0.0032 \pm 0.0000$ | $0.2223 \pm 0.0190$ |
| OmniAnomaly | $0.0355 \pm 0.0001$ | $0.0409 \pm 0.0002$ | $0.0031 \pm 0.0000$ | $0.0750 \pm 0.0000$ |
| MSCRED | $0.1744 \pm 0.1606$ | $0.0523 \pm 0.0242$ | $0.0032 \pm 0.0002$ | $0.0945 \pm 0.0226$ |
| DAGMM | $0.3400 \pm 0.0016$ | $0.0794 \pm 0.0026$ | $0.0032 \pm 0.0000$ | $0.1168 \pm 0.0027$ |

Table 11: More results on dataset PSM without IGAD.

| Model | AUC-PR | AUC-ROC | VUS-ROC | Standard-F1 |
|---|---|---|---|---|
| CATCH | $0.1323 \pm 0.0036$ | $0.5514 \pm 0.0096$ | $0.5044 \pm 0.0091$ | $0.0194 \pm 0.0116$ |
| M2N2 | $0.3915 \pm 0.0135$ | $0.8464 \pm 0.0033$ | $0.7856 \pm 0.0033$ | $0.0648 \pm 0.0000$ |
| FITS | $0.1180 \pm 0.0004$ | $0.5083 \pm 0.0009$ | $0.4732 \pm 0.0014$ | $0.0104 \pm 0.0003$ |
| ModernTCN | $0.1463 \pm 0.0003$ | $0.5742 \pm 0.0005$ | $0.5101 \pm 0.0019$ | $0.0316 \pm 0.0003$ |
| Peri-midFormer | $0.1378 \pm 0.0004$ | $0.5511 \pm 0.0014$ | $0.4910 \pm 0.0042$ | $0.0349 \pm 0.0005$ |
| SARAD | $0.1568 \pm 0.0135$ | $0.6524 \pm 0.0268$ | $0.4493 \pm 0.0097$ | $0.0290 \pm 0.0012$ |
| TimesNet | $0.1211 \pm 0.0022$ | $0.5136 \pm 0.0038$ | $0.4610 \pm 0.0048$ | $0.0173 \pm 0.0028$ |
| OFA | $0.1310 \pm 0.0004$ | $0.5390 \pm 0.0005$ | $0.4772 \pm 0.0015$ | $0.0110 \pm 0.0006$ |
| A.T. | $0.1162 \pm 0.0094$ | $0.5161 \pm 0.0360$ | $0.4875 \pm 0.0281$ | $0.0000 \pm 0.0000$ |
| FGANomaly | $0.2620 \pm 0.0119$ | $0.7480 \pm 0.0082$ | $0.5993 \pm 0.0080$ | $0.0031 \pm 0.0012$ |
| CAE-M | $0.1608 \pm 0.0009$ | $0.6548 \pm 0.0018$ | $0.4682 \pm 0.0025$ | $0.0248 \pm 0.0004$ |
| MTAD-GAT | $0.1495 \pm 0.0023$ | $0.6126 \pm 0.0056$ | $0.5373 \pm 0.0069$ | $0.0255 \pm 0.0005$ |
| OmniAnomaly | $0.1655 \pm 0.0010$ | $0.6218 \pm 0.0031$ | $0.4314 \pm 0.0083$ | $0.0257 \pm 0.0001$ |
| MSCRED | $0.2165 \pm 0.0100$ | $0.7431 \pm 0.0197$ | $0.5425 \pm 0.0456$ | $0.0220 \pm 0.0095$ |
| DAGMM | $0.2126 \pm 0.0025$ | $0.6912 \pm 0.0044$ | $0.4765 \pm 0.0040$ | $0.0350 \pm 0.0006$ |

| Model | PA-F1 | Event-based-F1 | R-based-F1 | Affiliation-F1 |
|---|---|---|---|---|
| CATCH | $0.7125 \pm 0.2074$ | $0.1237 \pm 0.0276$ | $0.0031 \pm 0.0001$ | $0.2804 \pm 0.1142$ |
| M2N2 | $0.3320 \pm 0.0000$ | $0.0274 \pm 0.0000$ | $0.0073 \pm 0.0000$ | $0.0219 \pm 0.0000$ |
| FITS | $0.8835 \pm 0.0004$ | $0.2184 \pm 0.0080$ | $0.0031 \pm 0.0000$ | $0.5232 \pm 0.0069$ |
| ModernTCN | $0.3639 \pm 0.0007$ | $0.1138 \pm 0.0073$ | $0.0032 \pm 0.0000$ | $0.1757 \pm 0.0161$ |
| Peri-midFormer | $0.3642 \pm 0.0002$ | $0.1397 \pm 0.0162$ | $0.0032 \pm 0.0000$ | $0.1911 \pm 0.0249$ |
| SARAD | $0.1074 \pm 0.1207$ | $0.0569 \pm 0.0049$ | $0.0032 \pm 0.0000$ | $0.1015 \pm 0.0059$ |
| TimesNet | $0.8550 \pm 0.0021$ | $0.1648 \pm 0.0074$ | $0.0031 \pm 0.0000$ | $0.4767 \pm 0.0394$ |
| OFA | $0.8683 \pm 0.0048$ | $0.1960 \pm 0.0082$ | $0.0031 \pm 0.0000$ | $0.5743 \pm 0.0147$ |
| A.T. | $0.0664 \pm 0.1484$ | $0.0050 \pm 0.0112$ | $0.0004 \pm 0.0009$ | $0.0792 \pm \text{NaN}$ |
| FGANomaly | $0.3214 \pm 0.0014$ | $0.0200 \pm 0.0029$ | $0.0028 \pm 0.0000$ | $0.0386 \pm 0.0008$ |
| CAE-M | $0.3148 \pm 0.0006$ | $0.0547 \pm 0.0001$ | $0.0035 \pm 0.0000$ | $0.0944 \pm 0.0000$ |
| MTAD-GAT | $0.3319 \pm 0.0016$ | $0.0874 \pm 0.0019$ | $0.0032 \pm 0.0000$ | $0.1823 \pm 0.0002$ |
| OmniAnomaly | $0.0355 \pm 0.0001$ | $0.0409 \pm 0.0002$ | $0.0031 \pm 0.0000$ | $0.0750 \pm 0.0000$ |
| MSCRED | $0.3461 \pm 0.0249$ | $0.0633 \pm 0.0130$ | $0.0032 \pm 0.0001$ | $0.1150 \pm 0.0204$ |
| DAGMM | $0.3397 \pm 0.0005$ | $0.0780 \pm 0.0002$ | $0.0032 \pm 0.0000$ | $0.1140 \pm 0.0002$ |

Table 12: More results on dataset SMAP with IGAD.

| Model | AUC-PR | AUC-ROC | VUS-ROC | Standard-F1 |
|---|---|---|---|---|
| CATCH | $0.2942 \pm 0.0008$ | $0.6088 \pm 0.0017$ | $0.6088 \pm 0.0017$ | $0.1329 \pm 0.0020$ |
| M2N2 | $0.1971 \pm 0.0427$ | $0.6754 \pm 0.0868$ | $0.6753 \pm 0.0869$ | $0.0000 \pm 0.0000$ |
| FITS | $0.2858 \pm 0.0126$ | $0.7845 \pm 0.0196$ | $0.7845 \pm 0.0197$ | $0.0194 \pm 0.0045$ |
| ModernTCN | $0.4165 \pm 0.0079$ | $0.8004 \pm 0.0053$ | $0.8004 \pm 0.0054$ | $0.2067 \pm 0.0025$ |
| Peri-midFormer | $0.5096 \pm 0.0273$ | $0.8312 \pm 0.0129$ | $0.8298 \pm 0.0130$ | $0.1971 \pm 0.0030$ |
| SARAD | $0.8437 \pm 0.0213$ | $0.9320 \pm 0.0060$ | $0.9337 \pm 0.0060$ | $0.2425 \pm 0.0064$ |
| TimesNet | $0.2731 \pm 0.0672$ | $0.7194 \pm 0.0970$ | $0.7194 \pm 0.0970$ | $0.0330 \pm 0.0197$ |
| OFA | $0.2977 \pm 0.0254$ | $0.7897 \pm 0.0326$ | $0.7897 \pm 0.0326$ | $0.0425 \pm 0.0138$ |
| A.T. | $0.2557 \pm 0.1120$ | $0.6327 \pm 0.0959$ | $0.6326 \pm 0.0961$ | $0.0439 \pm 0.0884$ |
| FGANomaly | $0.9840 \pm 0.0012$ | $0.9957 \pm 0.0006$ | $0.9957 \pm 0.0006$ | $0.3473 \pm 0.0099$ |
| CAE-M | $0.0718 \pm 0.0002$ | $0.0203 \pm 0.0141$ | $0.0203 \pm 0.0141$ | $0.0000 \pm 0.0000$ |
| MTAD-GAT | $0.5121 \pm 0.2364$ | $0.8976 \pm 0.0501$ | $0.8976 \pm 0.0501$ | $0.0718 \pm 0.0659$ |
| OmniAnomaly | $0.9064 \pm 0.0272$ | $0.9111 \pm 0.0284$ | $0.9111 \pm 0.0284$ | $0.4186 \pm 0.1677$ |
| MSCRED | $0.1248 \pm 0.0207$ | $0.3914 \pm 0.1556$ | $0.3914 \pm 0.1556$ | $0.0000 \pm 0.0000$ |
| DAGMM | $0.1106 \pm 0.0124$ | $0.0597 \pm 0.0144$ | $0.0597 \pm 0.0144$ | $0.0225 \pm 0.0311$ |

| Model | PA-F1 | Event-based-F1 | R-based-F1 | Affiliation-F1 |
|---|---|---|---|---|
| CATCH | $1.0000 \pm 0.0000$ | $1.0000 \pm 0.0000$ | $0.3063 \pm 0.0027$ | $0.9485 \pm 0.0002$ |
| M2N2 | $0.0000 \pm 0.0000$ | $0.0000 \pm 0.0000$ | $0.0000 \pm 0.0000$ | $\text{NaN} \pm \text{NaN}$ |
| FITS | $0.9020 \pm 0.0015$ | $0.0988 \pm 0.0209$ | $0.1662 \pm 0.0007$ | $0.5542 \pm 0.0163$ |
| ModernTCN | $0.9986 \pm 0.0005$ | $0.9879 \pm 0.0040$ | $0.3322 \pm 0.0008$ | $0.9484 \pm 0.0005$ |
| Peri-midFormer | $0.9998 \pm 0.0002$ | $0.9984 \pm 0.0022$ | $0.3288 \pm 0.0011$ | $0.9498 \pm 0.0002$ |
| SARAD | $0.9989 \pm 0.0009$ | $0.9922 \pm 0.0063$ | $0.3435 \pm 0.0021$ | $0.9492 \pm 0.0009$ |
| TimesNet | $0.9183 \pm 0.0165$ | $0.1839 \pm 0.1134$ | $0.1682 \pm 0.0027$ | $0.5879 \pm 0.0432$ |
| OFA | $0.9151 \pm 0.0135$ | $0.2188 \pm 0.0699$ | $0.1690 \pm 0.0019$ | $0.6027 \pm 0.0358$ |
| A.T. | $0.7826 \pm 0.4380$ | $0.2615 \pm 0.3943$ | $0.1362 \pm 0.0768$ | $0.6389 \pm 0.1989$ |
| FGANomaly | $1.0000 \pm 0.0000$ | $1.0000 \pm 0.0000$ | $0.3724 \pm 0.0032$ | $0.9955 \pm 0.0001$ |
| CAE-M | $0.0000 \pm 0.0000$ | $0.0000 \pm 0.0000$ | $0.0000 \pm 0.0000$ | $\text{NaN} \pm \text{NaN}$ |
| MTAD-GAT | $0.9480 \pm 0.0293$ | $0.4098 \pm 0.3328$ | $0.1997 \pm 0.0676$ | $0.6742 \pm 0.1815$ |
| OmniAnomaly | $1.0000 \pm 0.0000$ | $1.0000 \pm 0.0000$ | $0.3893 \pm 0.0526$ | $0.9961 \pm 0.0011$ |
| MSCRED | $0.0000 \pm 0.0000$ | $0.0000 \pm 0.0000$ | $0.0000 \pm 0.0000$ | $0.3334 \pm 0.0001$ |
| DAGMM | $0.4000 \pm 0.5477$ | $0.4000 \pm 0.5477$ | $0.1096 \pm 0.1500$ | $0.9490 \pm 0.0000$ |

Table 13: More results on dataset SMAP without IGAD.

| Model | AUC-PR | AUC-ROC | VUS-ROC | Standard-F1 |
|---|---|---|---|---|
| CATCH | $0.2898 \pm 0.0013$ | $0.5913 \pm 0.0052$ | $0.5912 \pm 0.0052$ | $0.1410 \pm 0.0051$ |
| M2N2 | $0.1954 \pm 0.0056$ | $0.6345 \pm 0.0123$ | $0.6344 \pm 0.0123$ | $0.0909 \pm 0.0081$ |
| FITS | $0.2713 \pm 0.0115$ | $0.7675 \pm 0.0154$ | $0.7675 \pm 0.0154$ | $0.0203 \pm 0.0048$ |
| ModernTCN | $0.4594 \pm 0.0098$ | $0.8220 \pm 0.0051$ | $0.8220 \pm 0.0050$ | $0.1746 \pm 0.0053$ |
| Peri-midFormer | $0.5105 \pm 0.0288$ | $0.8304 \pm 0.0132$ | $0.8304 \pm 0.0133$ | $0.1977 \pm 0.0044$ |
| SARAD | $0.8490 \pm 0.0184$ | $0.9335 \pm 0.0053$ | $0.9335 \pm 0.0053$ | $0.2462 \pm 0.0076$ |
| TimesNet | $0.2677 \pm 0.0742$ | $0.7256 \pm 0.0993$ | $0.7256 \pm 0.0993$ | $0.0297 \pm 0.0168$ |
| OFA | $0.2959 \pm 0.0256$ | $0.7949 \pm 0.0254$ | $0.7949 \pm 0.0254$ | $0.0442 \pm 0.0148$ |
| A.T. | $0.2397 \pm 0.0828$ | $0.5897 \pm 0.0911$ | $0.5895 \pm 0.0913$ | $0.0810 \pm 0.1053$ |
| FGANomaly | $0.9208 \pm 0.0270$ | $0.9560 \pm 0.0060$ | $0.9560 \pm 0.0060$ | $0.0606 \pm 0.0012$ |
| CAE-M | $0.0719 \pm 0.0002$ | $0.0218 \pm 0.0157$ | $0.0218 \pm 0.0157$ | $0.0000 \pm 0.0000$ |
| MTAD-GAT | $0.2324 \pm 0.0271$ | $0.6605 \pm 0.0518$ | $0.6604 \pm 0.0518$ | $0.0340 \pm 0.0008$ |
| OmniAnomaly | $0.0764 \pm 0.0002$ | $0.1057 \pm 0.0002$ | $0.1057 \pm 0.0002$ | $0.0000 \pm 0.0000$ |
| MSCRED | $0.0936 \pm 0.0011$ | $0.1213 \pm 0.0001$ | $0.1213 \pm 0.0001$ | $0.0000 \pm 0.0000$ |
| DAGMM | $0.0746 \pm 0.0032$ | $0.0242 \pm 0.0042$ | $0.0242 \pm 0.0043$ | $0.0018 \pm 0.0025$ |

| Model | PA-F1 | Event-based-F1 | R-based-F1 | Affiliation-F1 |
|---|---|---|---|---|
| CATCH | $1.0000 \pm 0.0000$ | $1.0000 \pm 0.0000$ | $0.3093 \pm 0.0042$ | $0.9489 \pm 0.0002$ |
| M2N2 | $0.8854 \pm 0.0087$ | $0.3166 \pm 0.0229$ | $0.1884 \pm 0.0014$ | $0.7227 \pm 0.0062$ |
| FITS | $0.9022 \pm 0.0017$ | $0.1029 \pm 0.0215$ | $0.1664 \pm 0.0008$ | $0.5547 \pm 0.0177$ |
| ModernTCN | $1.0000 \pm 0.0000$ | $1.0000 \pm 0.0000$ | $0.3197 \pm 0.0041$ | $0.9500 \pm 0.0003$ |
| Peri-midFormer | $0.9998 \pm 0.0002$ | $0.9984 \pm 0.0022$ | $0.3291 \pm 0.0016$ | $0.9498 \pm 0.0002$ |
| SARAD | $0.9992 \pm 0.0006$ | $0.9943 \pm 0.0042$ | $0.3447 \pm 0.0025$ | $0.9495 \pm 0.0006$ |
| TimesNet | $0.9142 \pm 0.0143$ | $0.1597 \pm 0.0884$ | $0.1677 \pm 0.0024$ | $0.5771 \pm 0.0349$ |
| OFA | $0.9179 \pm 0.0129$ | $0.2311 \pm 0.0778$ | $0.1692 \pm 0.0021$ | $0.6041 \pm 0.0382$ |
| A.T. | $0.9865 \pm 0.0131$ | $0.6238 \pm 0.3904$ | $0.1939 \pm 0.0414$ | $0.7903 \pm 0.1868$ |
| FGANomaly | $1.0000 \pm 0.0000$ | $1.0000 \pm 0.0000$ | $0.2745 \pm 0.0004$ | $0.9487 \pm 0.0000$ |
| CAE-M | $0.0000 \pm 0.0000$ | $0.0000 \pm 0.0000$ | $0.0000 \pm 0.0000$ | $\text{NaN} \pm \text{NaN}$ |
| MTAD-GAT | $0.9059 \pm 0.0014$ | $0.1674 \pm 0.0051$ | $0.1685 \pm 0.0001$ | $0.5657 \pm 0.0018$ |
| OmniAnomaly | $0.0000 \pm 0.0000$ | $0.0000 \pm 0.0000$ | $0.0000 \pm 0.0000$ | $\text{NaN} \pm \text{NaN}$ |
| MSCRED | $0.0000 \pm 0.0000$ | $0.0000 \pm 0.0000$ | $0.0000 \pm 0.0000$ | $\text{NaN} \pm \text{NaN}$ |
| DAGMM | $0.4000 \pm 0.5477$ | $0.4000 \pm 0.5477$ | $0.0991 \pm 0.1358$ | $0.9553 \pm 0.0001$ |

## E.2 Efficiency Evaluation

In this section, we compare the Training Time per Epoch ($\mathcal{T}_{w/o}$, $\mathcal{T}_w$), GPU usage ($\mathcal{G}_{w/o}$, $\mathcal{G}_w$), and CPU usage ($\mathcal{C}_{w/o}$, $\mathcal{C}_w$) before and after applying IGAD for each dataset and each model from Tab.14 to Tab.17. From the defined formulas shown as (7) and (8), a model will perform four additional mappings after the application of IGAD. More concretely, before integration with IGAD, a model performs a mapping $f(x)$, while $f(x)$, $f(z)$, $f'(z)$, $f(f'(z))$, and $f'(f(z))$ are carried out sequentially under the effect of IGAD. **While more operations are involved, the application of IGAD does not introduce additional parameters that need to be trained, as IGAD leverages a frozen copy of the training model.** IGAD introduces additional time and resource consumption due to the mapping associated with the manifold constraints, but **these costs occur only in the training phases. During inference, each model can perform only one mapping for reconstruction, calculate anomaly scores, and detect abnormal time points.** This means that a model with IGAD can achieve the same efficiency as the same model without IGAD during inference. For a dataset with larger scale, the time is possible to last longer.

In the evaluation of *Training Time per Epoch*, we find that in most cases, the models with IGAD show an increase in time of less than 10 seconds and in many cases, even less than 3 seconds per training epoch. The increase generally has limited impact on the overall training process, especially when higher performance is desired. Meanwhile, it is also indicated that OFA [74] tends to show a higher increase during one epoch. This phenomenon can be theoretically concluded as that the utilize of a pre-trained language model is more source-sensitive to fine-tune it for time series tasks. An additional fact shows that the PSM dataset needs more training time than SMD, MSL, and SMAP. The reason is that the data scale of the PSM dataset is larger than others, and the cumulative effect of multiple iterations in the same epoch and multiple mappings results in a longer training time. In the evaluation of *Max GPU Allocation for Training* and *Max CPU Allocation for Training*, different degrees of increase are listed. During these, the maximum GPU occupancy is around 8535.80 MB (about 8.34 GB), and the maximum CPU occupancy is around 537.44 MB (about 0.52GB). All experiments are performed with a single NVIDIA RTX 3090 24GB GPU, and 0.52 GB is also acceptable for most hardware conditions for deep learning currently. These additional GPU and CPU memories are selected to save frozen parameters, gradient graphs, and data segments to update the training model.

Meanwhile, we have also envisioned some potential strategies to reduce these additional computational costs in our future implementations: (1) Accelerate calculation by data and model parallelism; (2) For large models such as OFA [74], we can perform parameter-efficient fine-tuning with advanced methods, including [33, 19, 55] based on LoRA [23] to reduce the number of trainable parameters; (3) A memory mechanism can be included to reduce the number of true reconstructions. Concretely, when a piece of reconstructed time series is needed, we can index the memory module to generate this in terms of association.

Table 14: Efficiency evaluation on dataset SMD, comparing training time per epoch ($\mathcal{T}_{w/o}$, $\mathcal{T}_w$), GPU usage ($\mathcal{G}_{w/o}$, $\mathcal{G}_w$), and CPU usage ($\mathcal{C}_{w/o}$, $\mathcal{C}_w$) before and after applying IGAD.

| Model | Training Time per Epoch (s) | | | Max GPU Allocation for Training (MB) | | | Max CPU Allocation for Training (MB) | | |
|---|---|---|---|---|---|---|---|---|---|
| | $\mathcal{T}_{w/o}$ | $\mathcal{T}_w$ | $\mathcal{T}_w - \mathcal{T}_{w/o}$ | $\mathcal{G}_{w/o}$ | $\mathcal{G}_w$ | $\mathcal{G}_w - \mathcal{G}_{w/o}$ | $\mathcal{C}_{w/o}$ | $\mathcal{C}_w$ | $\mathcal{C}_w - \mathcal{C}_{w/o}$ |
| CATCH | 6.0878 | 17.2277 | 11.1399 | 3038.8540 | 5317.9945 | 2279.1405 | 2252.5469 | 2362.8828 | 110.3359 |
| M2N2 | 0.8424 | 1.2809 | 0.4385 | 6.5044 | 10.9773 | 4.4729 | 2190.1797 | 2603.9336 | 413.7539 |
| FITS | 1.1934 | 1.9876 | 0.7942 | 19.0562 | 39.9126 | 20.8564 | 4719.1914 | 4756.3789 | 37.1875 |
| ModernTCN | 3.4464 | 9.9653 | 6.5188 | 1679.2700 | 6333.5298 | 4654.2598 | 4776.7734 | 4793.4648 | 16.6914 |
| Peri-midFormer | 1.1767 | 5.8734 | 4.6967 | 289.3027 | 8825.1006 | 8535.7979 | 4773.2188 | 4826.9961 | 53.7773 |
| SARAD | 1.4122 | 2.5430 | 1.1308 | 778.5454 | 2392.6172 | 1614.0718 | 4751.1641 | 4766.2148 | 15.0508 |
| TimesNet | 1.3733 | 2.6364 | 1.2632 | 299.8774 | 411.8228 | 111.9453 | 4790.5352 | 4832.9365 | 42.4013 |
| OFA | 3.6833 | 21.4194 | 17.7361 | 1918.6016 | 6223.6816 | 4305.0801 | 4863.4492 | 4910.8242 | 47.3750 |
| A.T. | 2.0369 | 3.9399 | 1.9030 | 1834.0044 | 4781.5825 | 2947.5781 | 4760.8867 | 4789.8633 | 28.9766 |
| FGANomaly | 1.6585 | 2.1942 | 0.5357 | 58.0522 | 523.9907 | 465.9385 | 4739.8789 | 4746.5241 | 6.6452 |
| CAE-M | 0.9261 | 1.2022 | 0.2761 | 115.1748 | 301.7012 | 186.5264 | 4732.7109 | 4749.4766 | 16.7656 |
| MTAD-GAT | 1.2675 | 2.2031 | 0.9356 | 599.8101 | 2052.2046 | 1452.3945 | 4775.2891 | 4783.0039 | 7.7148 |
| OmniAnomaly | 1.0875 | 1.8211 | 0.7336 | 45.4243 | 169.6577 | 124.2334 | 4748.5391 | 4760.3894 | 11.8503 |
| MSCRED | 2.8281 | 10.2687 | 7.4406 | 3314.2231 | 8588.2192 | 5273.9961 | 4742.3281 | 4743.5591 | 1.2309 |
| DAGMM | 0.9995 | 1.1030 | 0.1034 | 10.8193 | 21.5449 | 10.7256 | 4698.9609 | 4701.4883 | 2.5273 |

Table 15: Efficiency evaluation on dataset MSL, comparing training time per epoch ($\mathcal{T}_{w/o}$, $\mathcal{T}_w$), GPU usage ($\mathcal{G}_{w/o}$, $\mathcal{G}_w$), and CPU usage ($\mathcal{C}_{w/o}$, $\mathcal{C}_w$) before and after applying IGAD.

| Model | Training Time per Epoch (s) | | | Max GPU Allocation for Training (MB) | | | Max CPU Allocation for Training (MB) | | |
|---|---|---|---|---|---|---|---|---|---|
| | $\mathcal{T}_{w/o}$ | $\mathcal{T}_w$ | $\mathcal{T}_w - \mathcal{T}_{w/o}$ | $\mathcal{G}_{w/o}$ | $\mathcal{G}_w$ | $\mathcal{G}_w - \mathcal{G}_{w/o}$ | $\mathcal{C}_{w/o}$ | $\mathcal{C}_w$ | $\mathcal{C}_w - \mathcal{C}_{w/o}$ |
| CATCH | 1.5938 | 3.3659 | 1.7721 | 4230.5103 | 8037.9696 | 3807.4593 | 2312.8828 | 2560.7383 | 247.8555 |
| M2N2 | 0.5443 | 0.9091 | 0.3647 | 12.1436 | 20.9563 | 8.8127 | 2193.8672 | 2581.3945 | 387.5273 |
| FITS | 0.8837 | 1.0449 | 0.1612 | 28.5674 | 58.7539 | 30.1865 | 4731.3164 | 4768.3633 | 37.0469 |
| ModernTCN | 1.2722 | 2.3140 | 1.0418 | 2438.2866 | 9111.5781 | 6673.2915 | 4776.7539 | 4792.0039 | 15.2500 |
| Peri-midFormer | 0.8942 | 1.4531 | 0.5589 | 725.4805 | 4876.8081 | 4151.3276 | 4819.9063 | 4861.1347 | 41.2284 |
| SARAD | 0.9593 | 1.1728 | 0.2136 | 1113.1733 | 3489.0684 | 2375.8950 | 4747.0664 | 4777.9102 | 30.8438 |
| TimesNet | 0.8971 | 1.0556 | 0.1585 | 303.2749 | 419.8989 | 116.6240 | 4789.4727 | 4818.4922 | 29.0195 |
| OFA | 1.2976 | 3.4958 | 2.1982 | 1919.3901 | 6229.9653 | 4310.5752 | 4853.4688 | 5387.6055 | 534.1367 |
| A.T. | 0.9892 | 1.2458 | 0.2566 | 1253.3545 | 2974.2109 | 1720.8564 | 4768.0742 | 4797.3750 | 29.3008 |
| FGANomaly | 0.9215 | 1.0265 | 0.1051 | 59.7305 | 165.9316 | 106.2012 | 4728.4922 | 4733.8086 | 5.3164 |
| CAE-M | 0.8479 | 0.8915 | 0.0436 | 165.8027 | 435.1094 | 269.3066 | 4754.1836 | 4772.0156 | 17.8320 |
| MTAD-GAT | 0.8763 | 1.0014 | 0.1251 | 1006.0601 | 3351.1172 | 2345.0571 | 4759.7734 | 4762.5593 | 2.7859 |
| OmniAnomaly | 0.8426 | 0.9404 | 0.0978 | 49.0093 | 183.8770 | 134.8677 | 4740.9492 | 4768.8750 | 27.9258 |
| MSCRED | 1.2203 | 2.3347 | 1.1144 | 4697.5435 | 12323.8037 | 7626.2603 | 4767.2188 | 4767.6367 | 0.4180 |
| DAGMM | 0.8396 | 0.8602 | 0.0206 | 15.6182 | 32.5830 | 16.9648 | 4727.1992 | 4728.4023 | 1.2031 |

Table 16: Efficiency evaluation on dataset PSM, comparing training time per epoch ($\mathcal{T}_{w/o}$, $\mathcal{T}_w$), GPU usage ($\mathcal{G}_{w/o}$, $\mathcal{G}_w$), and CPU usage ($\mathcal{C}_{w/o}$, $\mathcal{C}_w$) before and after applying IGAD.

| Model | Training Time per Epoch (s) | | | Max GPU Allocation for Training (MB) | | | Max CPU Allocation for Training (MB) | | |
|---|---|---|---|---|---|---|---|---|---|
| | $\mathcal{T}_{w/o}$ | $\mathcal{T}_w$ | $\mathcal{T}_w - \mathcal{T}_{w/o}$ | $\mathcal{G}_{w/o}$ | $\mathcal{G}_w$ | $\mathcal{G}_w - \mathcal{G}_{w/o}$ | $\mathcal{C}_{w/o}$ | $\mathcal{C}_w$ | $\mathcal{C}_w - \mathcal{C}_{w/o}$ |
| CATCH | 44.4446 | 131.0026 | 86.5580 | 2228.2168 | 5258.5916 | 3030.3749 | 2285.8203 | 2389.8164 | 103.9961 |
| M2N2 | 2.5754 | 4.8112 | 2.2358 | 3.7534 | 8.4933 | 4.7399 | 2229.9570 | 2725.6836 | 495.7266 |
| FITS | 4.0280 | 8.0651 | 4.0370 | 12.5127 | 26.6787 | 14.1660 | 4739.9609 | 4760.7305 | 20.7695 |
| ModernTCN | 15.9907 | 54.3645 | 38.3737 | 1105.6284 | 4172.5288 | 3066.9004 | 4769.9844 | 4785.1836 | 15.1992 |
| Peri-midFormer | 4.2939 | 22.7822 | 18.4883 | 146.5879 | 583.9233 | 437.3354 | 4780.7969 | 4785.6680 | 4.8711 |
| SARAD | 4.9100 | 14.7800 | 9.8700 | 551.7329 | 1610.5713 | 1058.8384 | 4745.7266 | 4748.1641 | 2.4375 |
| TimesNet | 5.4077 | 16.3706 | 10.9629 | 288.7642 | 400.0938 | 111.3296 | 4781.2070 | 4791.6211 | 10.4141 |
| OFA | 24.8965 | 225.4139 | 200.5174 | 1917.9985 | 6221.9976 | 4303.9990 | 4872.3086 | 5395.2734 | 522.9648 |
| A.T. | 13.7400 | 34.3816 | 20.6416 | 1870.1704 | 4878.5933 | 3008.4229 | 4787.3672 | 4831.0295 | 43.6623 |
| FGANomaly | 9.6031 | 15.4628 | 5.8597 | 56.7183 | 5440.7212 | 5384.0029 | 4759.8750 | 4766.3008 | 6.4258 |
| CAE-M | 1.7830 | 3.0684 | 1.2854 | 76.6055 | 201.3789 | 124.7734 | 4756.0391 | 4762.5078 | 6.4688 |
| MTAD-GAT | 3.6808 | 10.4775 | 6.7966 | 374.0786 | 1268.0796 | 894.0010 | 4972.4648 | 5000.8047 | 28.3398 |
| OmniAnomaly | 4.0156 | 10.3294 | 6.3138 | 43.8564 | 159.6650 | 115.8086 | 4752.7734 | 4767.4421 | 14.6687 |
| MSCRED | 16.2451 | 58.4670 | 42.2220 | 2259.1724 | 5729.1802 | 3470.0078 | 4765.6328 | 4765.6641 | 0.0313 |
| DAGMM | 1.8711 | 3.0836 | 1.2125 | 7.1514 | 14.2622 | 7.1108 | 4740.0156 | 4758.8516 | 18.8359 |

Table 17: Efficiency evaluation on dataset SMAP, comparing training time per epoch ($\mathcal{T}_{w/o}$, $\mathcal{T}_w$), GPU usage ($\mathcal{G}_{w/o}$, $\mathcal{G}_w$), and CPU usage ($\mathcal{C}_{w/o}$, $\mathcal{C}_w$) before and after applying IGAD.

| Model | Training Time per Epoch (s) | | | Max GPU Allocation for Training (MB) | | | Max CPU Allocation for Training (MB) | | |
|---|---|---|---|---|---|---|---|---|---|
| | $\mathcal{T}_{w/o}$ | $\mathcal{T}_w$ | $\mathcal{T}_w - \mathcal{T}_{w/o}$ | $\mathcal{G}_{w/o}$ | $\mathcal{G}_w$ | $\mathcal{G}_w - \mathcal{G}_{w/o}$ | $\mathcal{C}_{w/o}$ | $\mathcal{C}_w$ | $\mathcal{C}_w - \mathcal{C}_{w/o}$ |
| CATCH | 2.5922 | 6.4548 | 3.8626 | 2228.2236 | 4565.1578 | 2336.9341 | 2193.1484 | 2210.7266 | 17.5781 |
| M2N2 | 0.6537 | 0.9807 | 0.3269 | 3.7534 | 7.4113 | 3.6579 | 2158.4492 | 2596.3633 | 437.9141 |
| FITS | 0.9403 | 1.2141 | 0.2737 | 13.1851 | 26.6787 | 13.4937 | 4717.2734 | 4723.3047 | 6.0313 |
| ModernTCN | 1.6361 | 3.5830 | 1.9468 | 1105.6284 | 4172.5288 | 3066.9004 | 4753.0820 | 4772.8477 | 19.7656 |
| Peri-midFormer | 1.0926 | 2.1401 | 1.0474 | 1263.2222 | 1407.9321 | 144.7100 | 4773.3984 | 4784.6780 | 11.2795 |
| SARAD | 1.0455 | 1.4257 | 0.3802 | 551.9321 | 1610.5713 | 1058.6392 | 4736.4648 | 4750.1250 | 13.6602 |
| TimesNet | 1.0640 | 1.6658 | 0.6017 | 306.8296 | 408.3438 | 101.5142 | 4765.6133 | 4771.3477 | 5.7344 |
| OFA | 2.2287 | 10.4260 | 8.1973 | 1917.9985 | 6221.9976 | 4304.0000 | 4855.6602 | 5393.3086 | 537.4414 |
| A.T. | 1.3751 | 2.2921 | 0.9170 | 1936.3228 | 5066.8916 | 3130.5688 | 4766.0039 | 4792.0977 | 26.0938 |
| FGANomaly | 1.1534 | 1.3887 | 0.2354 | 56.7046 | 228.3887 | 171.6841 | 4727.0039 | 4740.8906 | 13.8867 |
| CAE-M | 0.8584 | 0.9544 | 0.0960 | 76.6055 | 201.3789 | 124.7734 | 4736.6563 | 4746.7227 | 10.0664 |
| MTAD-GAT | 0.9592 | 1.3120 | 0.3528 | 374.0786 | 1268.0796 | 894.0010 | 4734.4883 | 4738.7656 | 4.2773 |
| OmniAnomaly | 0.9489 | 1.2228 | 0.2739 | 43.8564 | 159.6650 | 115.8086 | 4728.8945 | 4739.2734 | 10.3789 |
| MSCRED | 1.6051 | 3.7906 | 2.1856 | 2259.1724 | 5729.1802 | 3470.0078 | 4743.4219 | 4744.6059 | 1.1841 |
| DAGMM | 0.8615 | 0.9203 | 0.0588 | 7.1514 | 14.2622 | 7.1108 | 4711.6016 | 4737.3438 | 25.7422 |

### E.3 Hyperparameter Analysis

Here, we list the instructions to choose the optimal $\lambda_{\text{idem}}$, $\lambda_{\text{tight}}$ and $\alpha$ for a given dataset and select the model DAGMM [75] with dataset SMAP to show the analysis of hyperparameters. The results are displayed in Fig.8, Fig.9 and Fig.10.

First, in Fig.8, we calculate the frequency of $\lambda_{\text{idem}}$, $\lambda_{\text{tight}}$, and $\alpha$ for all the experiments conducted. The results indicate that, for most cases, the values of $\lambda_{\text{idem}}$ are located in the interval [0.1, 0.5], which means that a relatively smaller $\lambda_{\text{idem}}$ may be better for new datasets. For the values of $\lambda_{\text{tight}}$, they focus mainly on the two endpoint values and maintain a relatively uniform distribution throughout the other central parts. Finally, a larger $\alpha$ may be considered as a priority.

In the following part, the parameter sensitivity analysis conducted in the following also supports the instructions listed above. We show the results with mean and standard deviation in Fig.9, and 95% confidence intervals in Fig.10. Concretely, we fix two of $\lambda_{\text{idem}}$, $\lambda_{\text{tight}}$ and $\alpha$ as the optimal values shown in Tab.5. Then, we vary the remaining one from 0.1 to 2.0 with a step of 0.1. For $\lambda_{\text{idem}}$, the model achieves the best performance when $\lambda_{\text{idem}}$ is 0.1 (a smaller one). For $\lambda_{\text{tight}}$, with the tightness effect changing from loose to strict ($\lambda_{\text{tight}}$ changing from small to large), the performance of the model changes from up to down. We attribute it to over tightness, which even drops out normal instances. For $\alpha$, it is clearly shown that the performance improves with larger $\alpha$ and levels off when $\alpha$ is greater than 1.5.

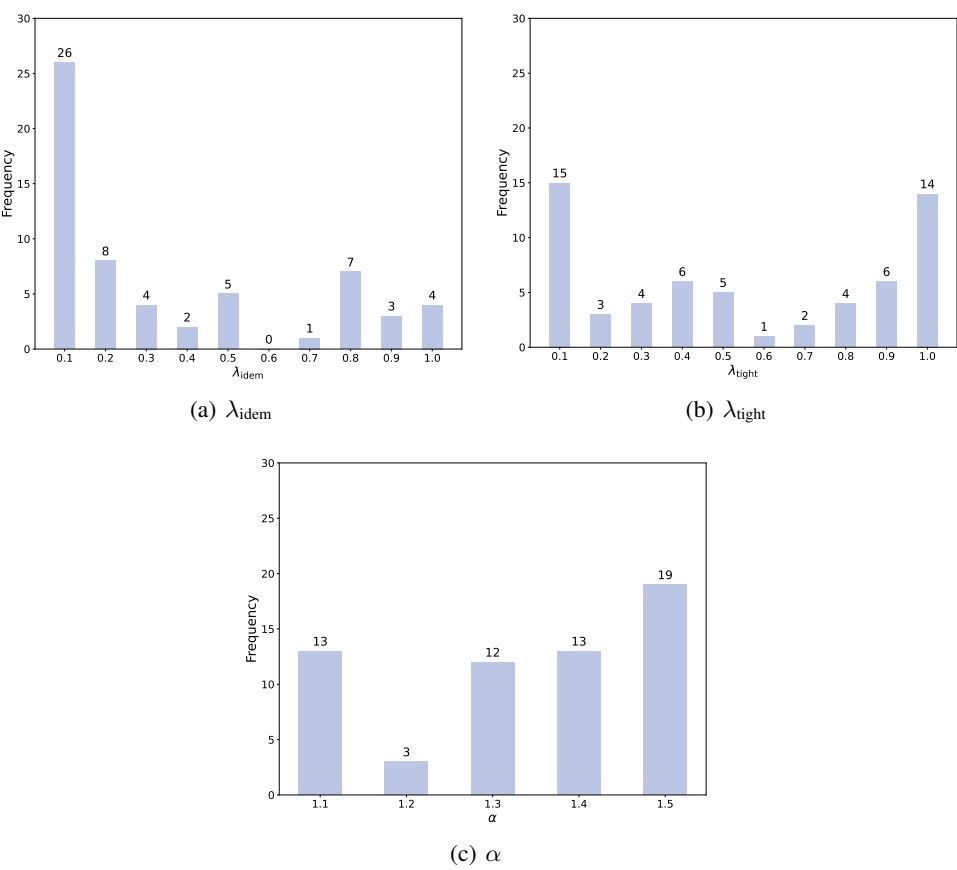

(a) $\lambda_{\text{idem}}$          (b) $\lambda_{\text{tight}}$

(c) $\alpha$

Figure 8: Parameter frequency records for $\lambda_{\text{idem}}$, $\lambda_{\text{tight}}$ and $\alpha$.

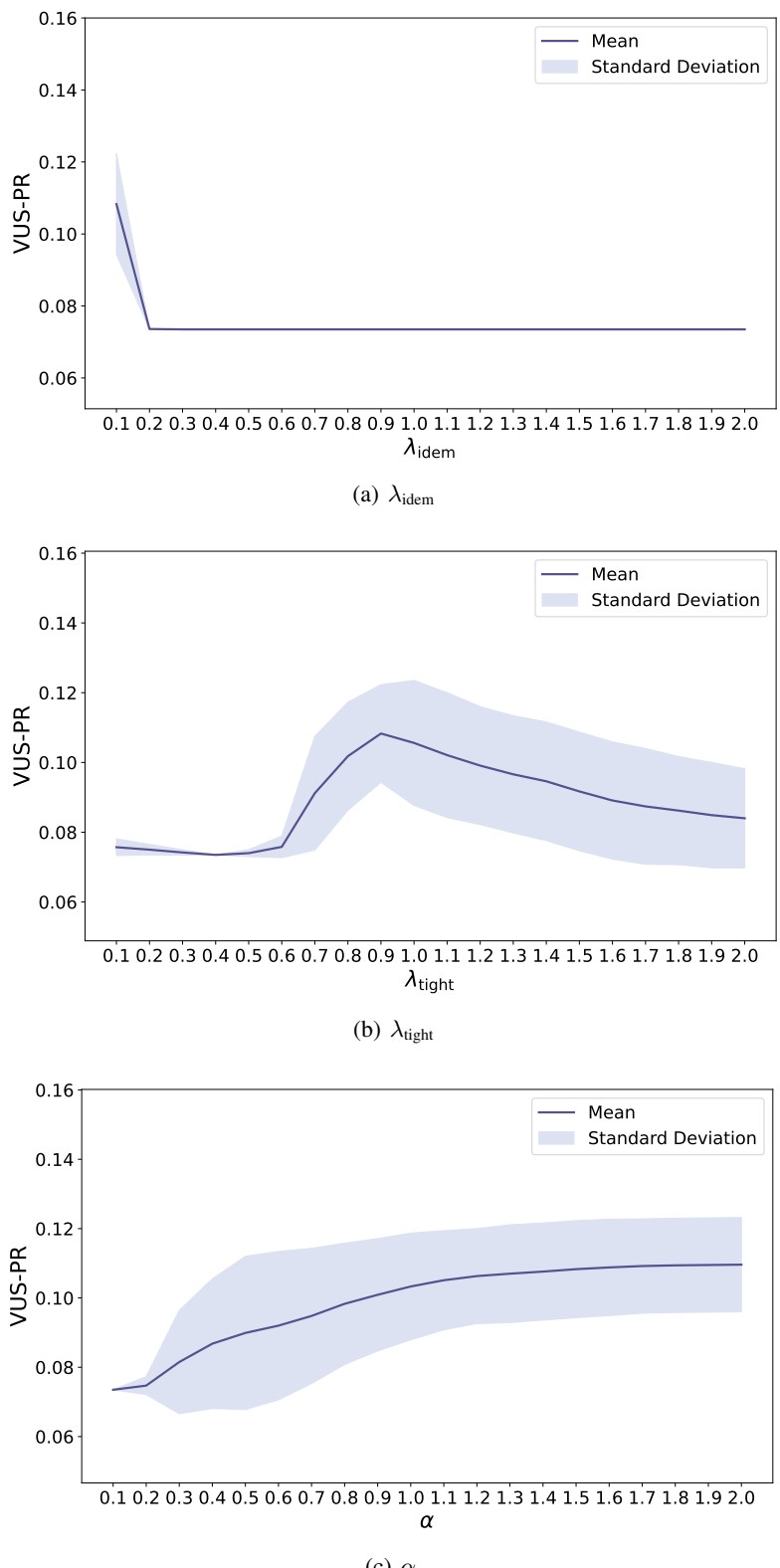

Figure 9: Parameter sensitivity analysis for $\lambda_{\text{idem}}$, $\lambda_{\text{tight}}$ and $\alpha$ with mean and standard deviation.

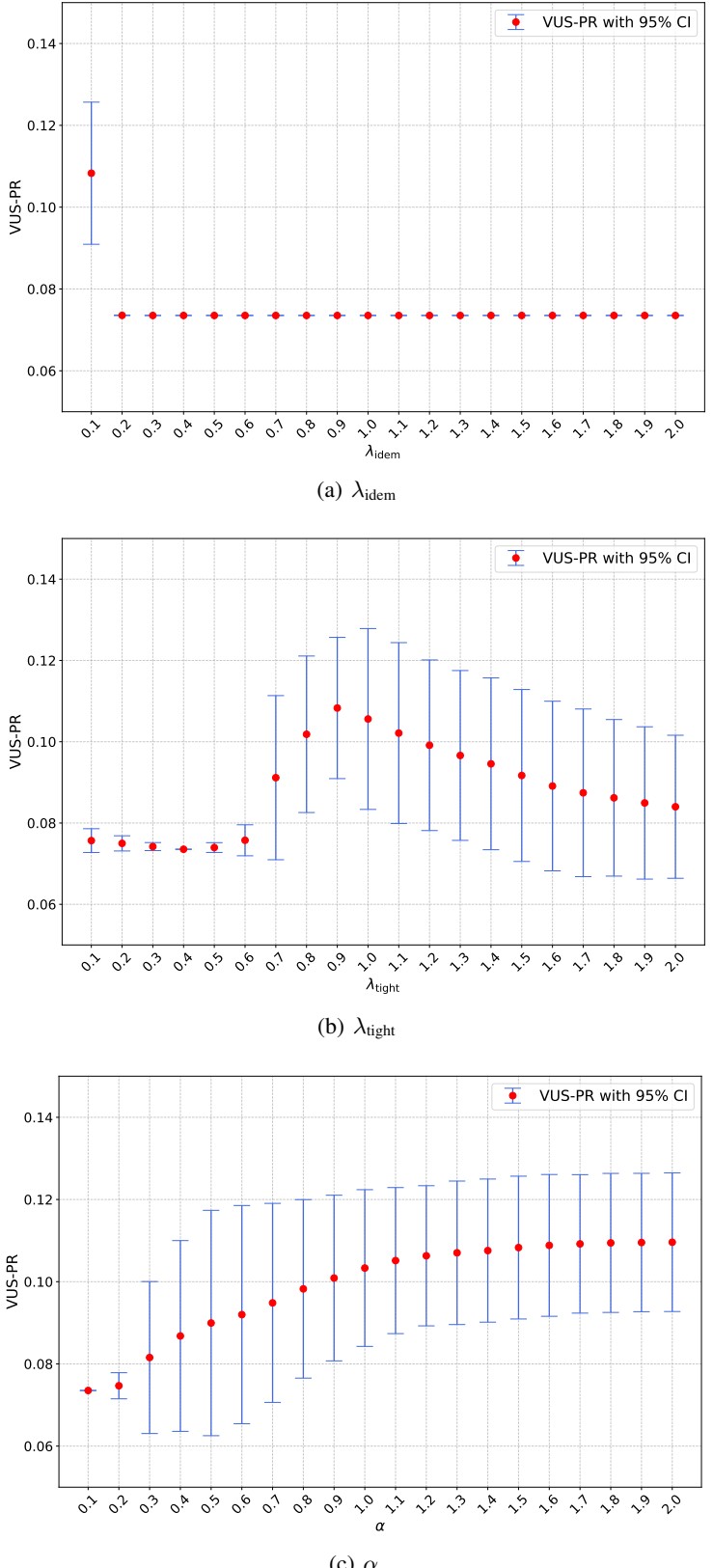

(a) $\lambda_{\text{idem}}$

(b) $\lambda_{\text{tight}}$

(c) $\alpha$

Figure 10: Parameter sensitivity analysis for $\lambda_{\text{idem}}$, $\lambda_{\text{tight}}$ and $\alpha$ with 95% confidence intervals.

## E.4 Visualization of Latent Space

To further verify the effectiveness of IGAD, we visualize the latent space of different models before and after applying IGAD in Fig.11. It can be observed that, under the effect of IGAD, the model gains a clearer boundary to distinguish normal instances from abnormal instances. This aligns with our design principles to modify and tighten the target manifold $\mathcal{M}_{\text{target}}$, with the aim of containing enough normal instances and drop out potential abnormal instances.

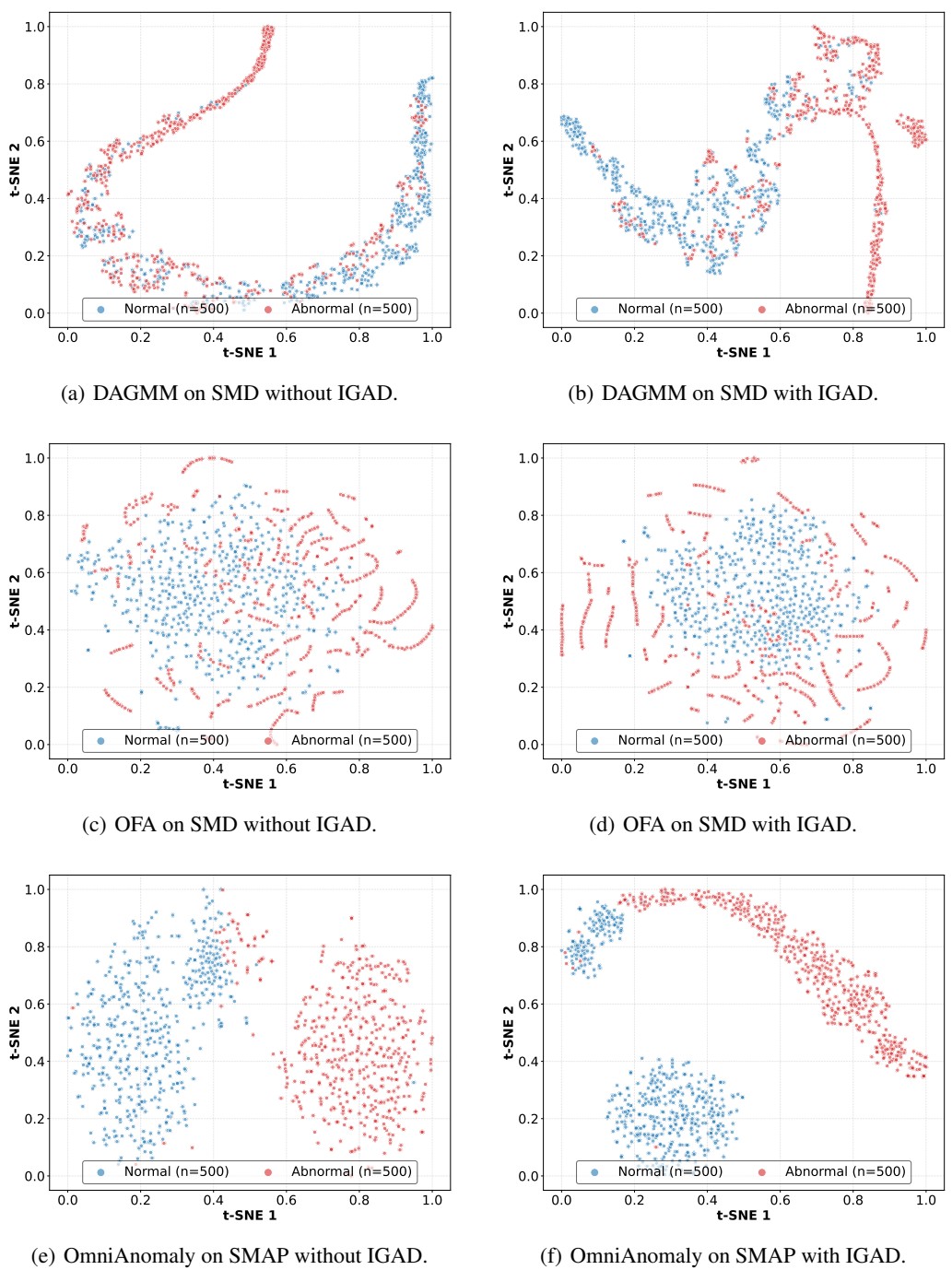

(a) DAGMM on SMD without IGAD.

(b) DAGMM on SMD with IGAD.

(c) OFA on SMD without IGAD.

(d) OFA on SMD with IGAD.

(e) OmniAnomaly on SMAP without IGAD.

(f) OmniAnomaly on SMAP with IGAD.

Figure 11: The visualization of latent space using t-SNE before and after applying IGAD.

## E.5 Maintain Data Patterns under Noise

In our experiments to demonstrate that IGAD can help balance robustness and sensitivity in Sect.4.2.3, we have incorporated a noise strategy into the testing data. In this part, we employ heatmaps to show these data with weighted noise from Fig.12 to Fig.15. We have found that noise-effected testing data display similar change patterns with the original data, which means that our noise strategy can verify their abilities to balance the robustness and sensitivity of different models while maintaining the necessary information.

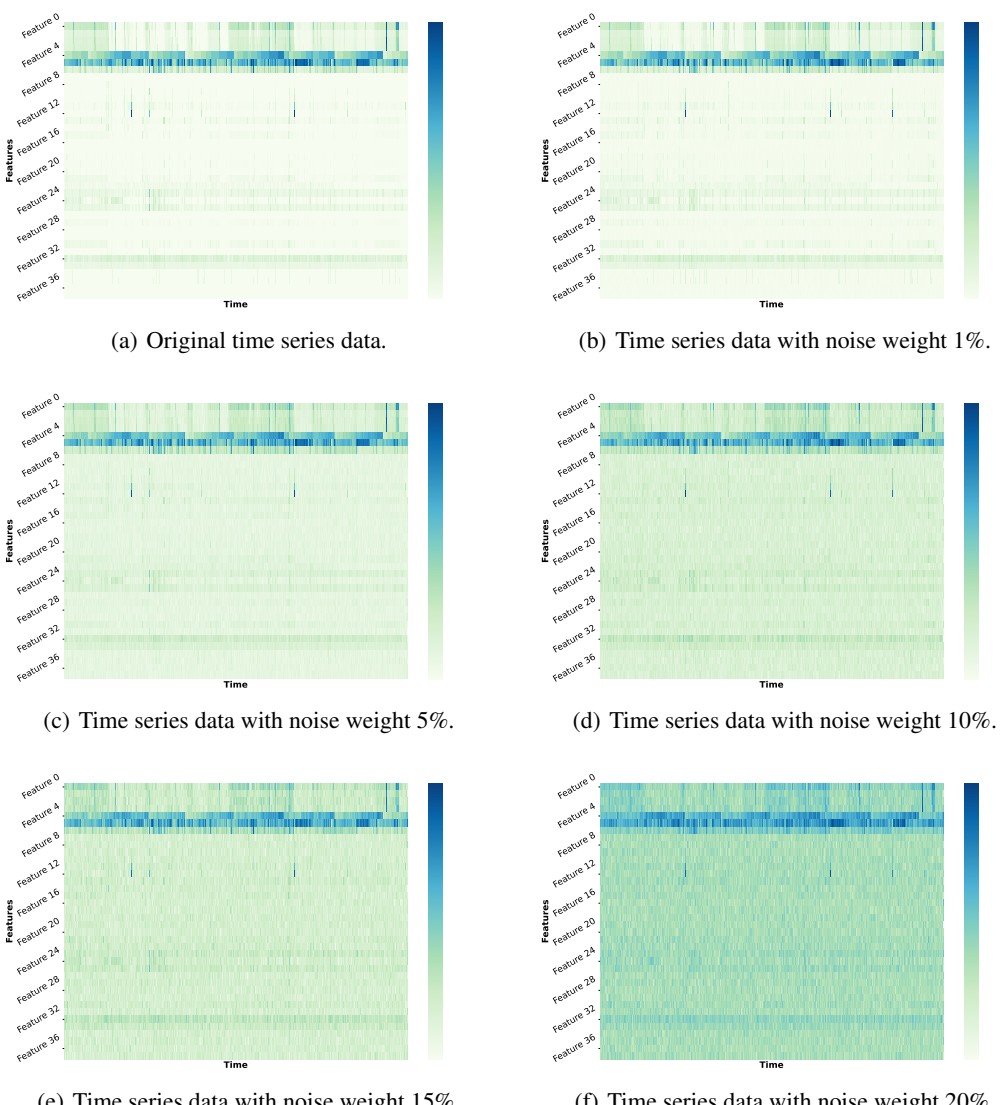

(a) Original time series data.

(b) Time series data with noise weight 1%.

(c) Time series data with noise weight 5%.

(d) Time series data with noise weight 10%.

(e) Time series data with noise weight 15%.

(f) Time series data with noise weight 20%.

Figure 12: Visualization for original data and noise-effect data on SMD.

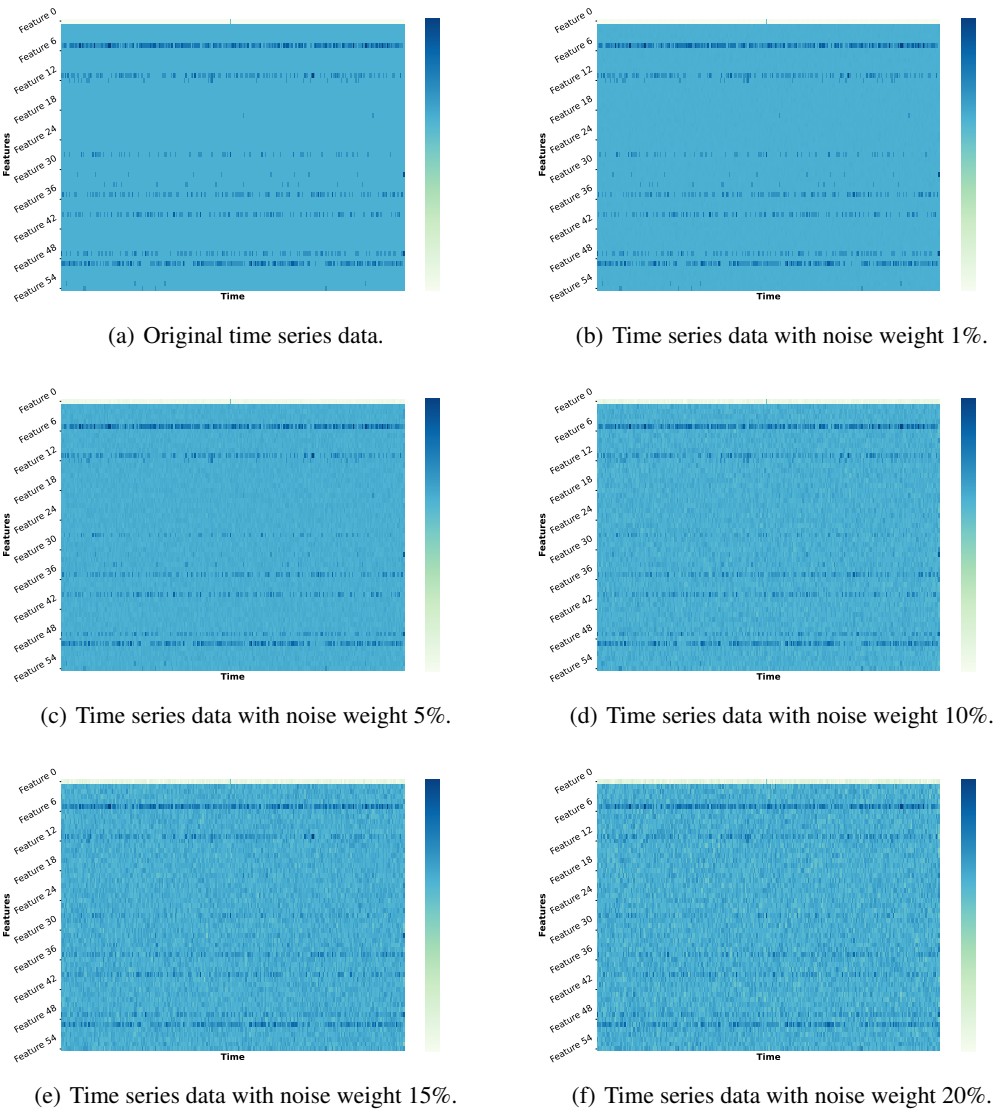

(a) Original time series data.

(b) Time series data with noise weight 1%.

(c) Time series data with noise weight 5%.

(d) Time series data with noise weight 10%.

(e) Time series data with noise weight 15%.

(f) Time series data with noise weight 20%.

Figure 13: Visualization for original data and noise-effect data on MSL.

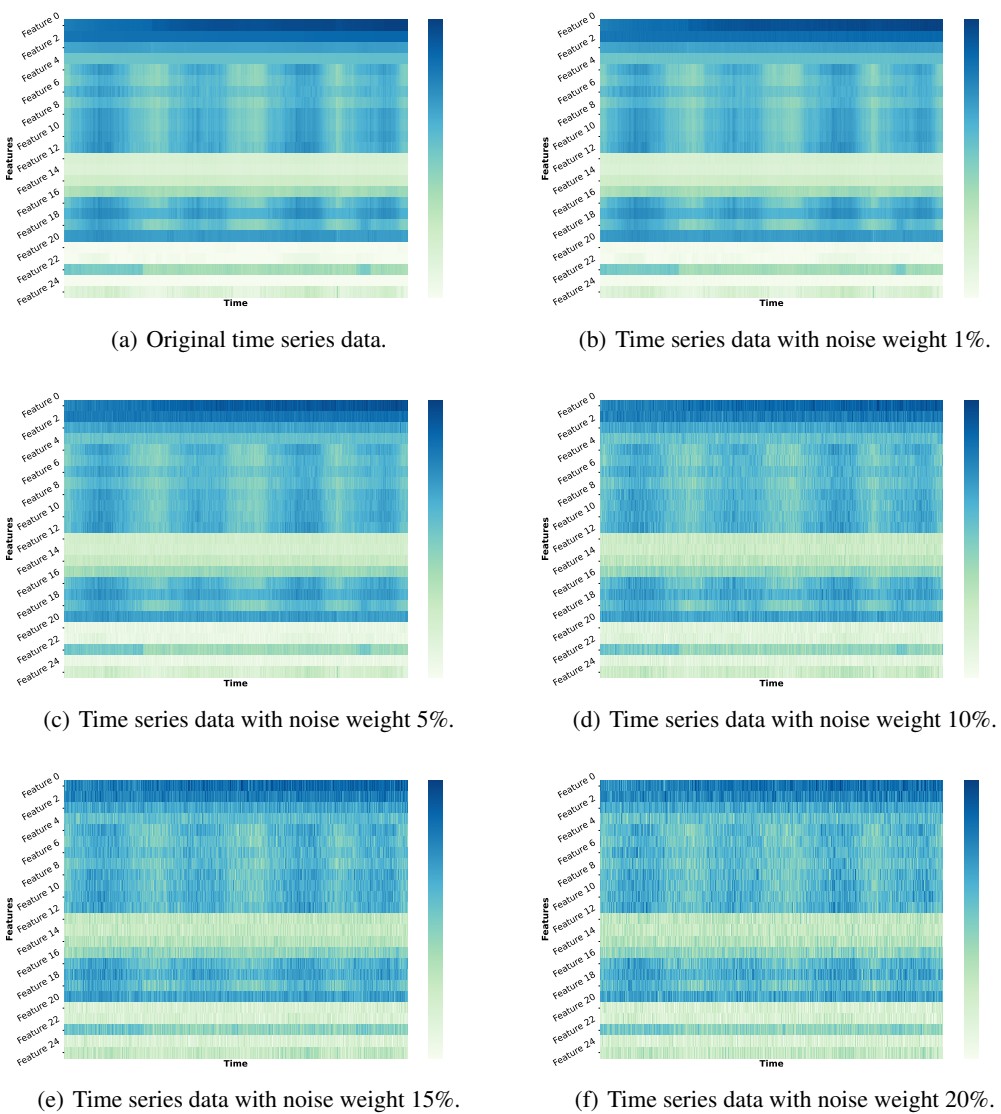

(a) Original time series data.

(b) Time series data with noise weight 1%.

(c) Time series data with noise weight 5%.

(d) Time series data with noise weight 10%.

(e) Time series data with noise weight 15%.

(f) Time series data with noise weight 20%.

Figure 14: Visualization for original data and noise-effect data on PSM.

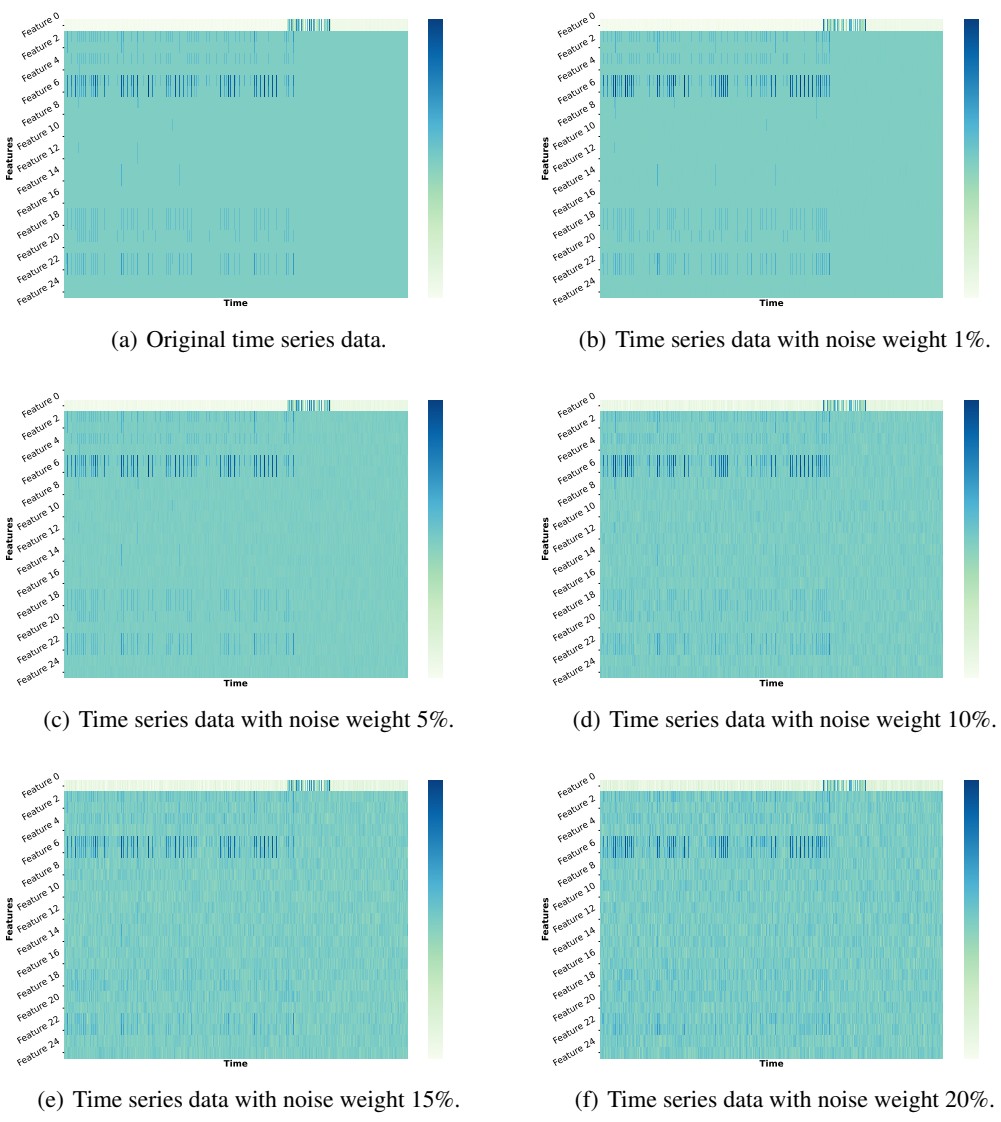

(a) Original time series data.

(b) Time series data with noise weight 1%.

(c) Time series data with noise weight 5%.

(d) Time series data with noise weight 10%.

(e) Time series data with noise weight 15%.

(f) Time series data with noise weight 20%.

Figure 15: Visualization for original data and noise-effect data on SMAP.

### E.6 Nyquist Criteria

We perturbed the time series data in the frequency domain to balance robustness and sensitivity in our experiments. This indicates that we should perform more analysis in the frequency domain to support this strategy. The **Nyquist Criterion** establishes a fundamental requirement for faithful reconstruction of a continuous-time signal from its discrete samples. This principle underpins modern digital signal processing systems, analog-to-digital conversion, and telecommunications. The theorem originated from Harry Nyquist's work in [37] and was later rigorously formalized by Claude Shannon in information theory [45]. It states that a band-limited signal with no frequency components exceeding $f_{max}$ (Hz) must be sampled at a rate $f_s \geq 2f_{max}$ to avoid aliasing and ensure complete signal recovery. We conclude the key concepts in this theory as follows:

- The minimum sampling rate $2f_{max}$ is termed the **Nyquist rate**.
- Half of the sampling frequency $\frac{f_s}{2}$ is referred to as the **Nyquist frequency**, $f_{Nyq}$.
- Spectral overlap, i.e., **aliasing**, may occur if $f_s < 2f_{max}$ or $f_{Nyq} < f_{max}$, causing irreversible distortion.

To verify compliance with the Nyquist-Shannon sampling theorem, the proposed procedure shown in Alg.1 first determines the sampling rate $f_s$ and the corresponding Nyquist frequency $f_{Nyq}$ from a user-provided sampling frequency descriptor for the input multivariate time series dataset $\mathbf{D}$. For each time series feature within $\mathbf{D}$, its frequency spectrum is obtained by the Fast Fourier Transform (FFT). The significant frequency components are then identified by comparing their normalized magnitudes against a relative threshold based on the peak magnitude in the spectrum. If the highest significant frequency detected in any feature exceeds $f_{Nyq}$, the dataset is flagged as non-compliant; otherwise, compliance is affirmed.

---

**Algorithm 1** Nyquist Criterion Compliance Verification

---

1: **Input:**
- Multivariate time series dataset $\mathbf{D} \in \mathbb{R}^{n \times k}$ ($n$ instances, $k$ variables)
- Sampling frequency descriptor $f_{desc}$ (e.g., "1 min")

2: **Output:** Boolean compliance status $\text{flag}_{Nyq}$

3: **procedure** CHECKNYQUIST($\mathbf{D}$, $f_{desc}$)

4:     Compute sampling rate: $f_s = 1/\Delta t$ according to $f_{desc}$            ▷ Unit: Hz

5:     Calculate Nyquist frequency: $f_{Nyq} = f_s/2$

6:     **for** each feature column $\mathbf{d}_i \in \mathbf{D}$ (where $i$ is the feature index) **do**

7:         Let $N_s = |\mathbf{d}_i|$ be the number of samples in the current feature column.

8:         Compute FFT: $\mathbf{Y}_i = \mathcal{F}(\mathbf{d}_i)$

9:         Generate frequency axis (positive frequencies): $\mathbf{f}_i = \text{fftfreq}(N_s, \Delta t)[0 : N_s/2]$

10:        Compute normalized magnitude: $\mathbf{A}_i = |\mathbf{Y}_i[0 : N_s/2]|/N_s$

11:        Detect significant frequencies:

$$\mathcal{F}_{sig} = \{f \in \mathbf{f}_i \mid A(f) > 0.1 \cdot \max(\mathbf{A}_i)\}$$

12:        **if** $\mathcal{F}_{sig} = \emptyset$ **then**

13:            **Continue**         ▷ No significant frequencies above threshold for this feature

14:        **if** $\max(\mathcal{F}_{sig}) > f_{Nyq}$ **then**

15:            **return False**    ▷ Violation: Max significant frequency exceeds Nyquist frequency

16:     **return True**                 ▷ All features comply with Nyquist criterion

---

From the study [50], we can get the information that the selected datasets in our experiments, SMD, MSL, PSM and SMAP, are sampled with a sampling frequency of 1 min, so we set $f_s = 60$ according to the unit in seconds. **After verification, all datasets in our experiments satisfy the Nyquist criterion.** Further, due to the space constraints on each page, we visualized the validation results for the first eight variables of each dataset from Fig.16 to Fig.19. The full validation codes can be found in our anonymized repository and run directly to carry out and check the results of full verification.

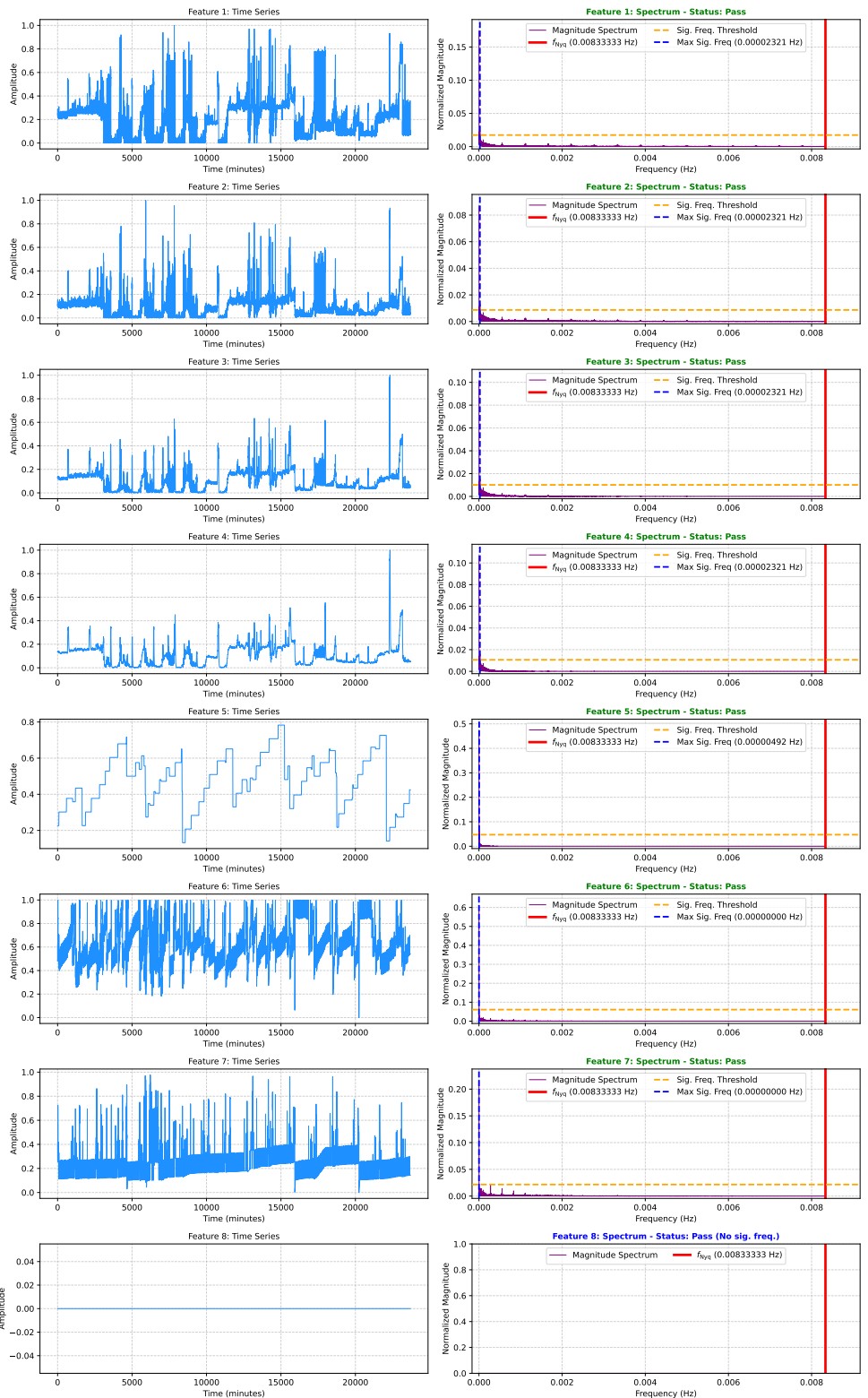

Figure 16: Nyquist criteria verification on SMD.

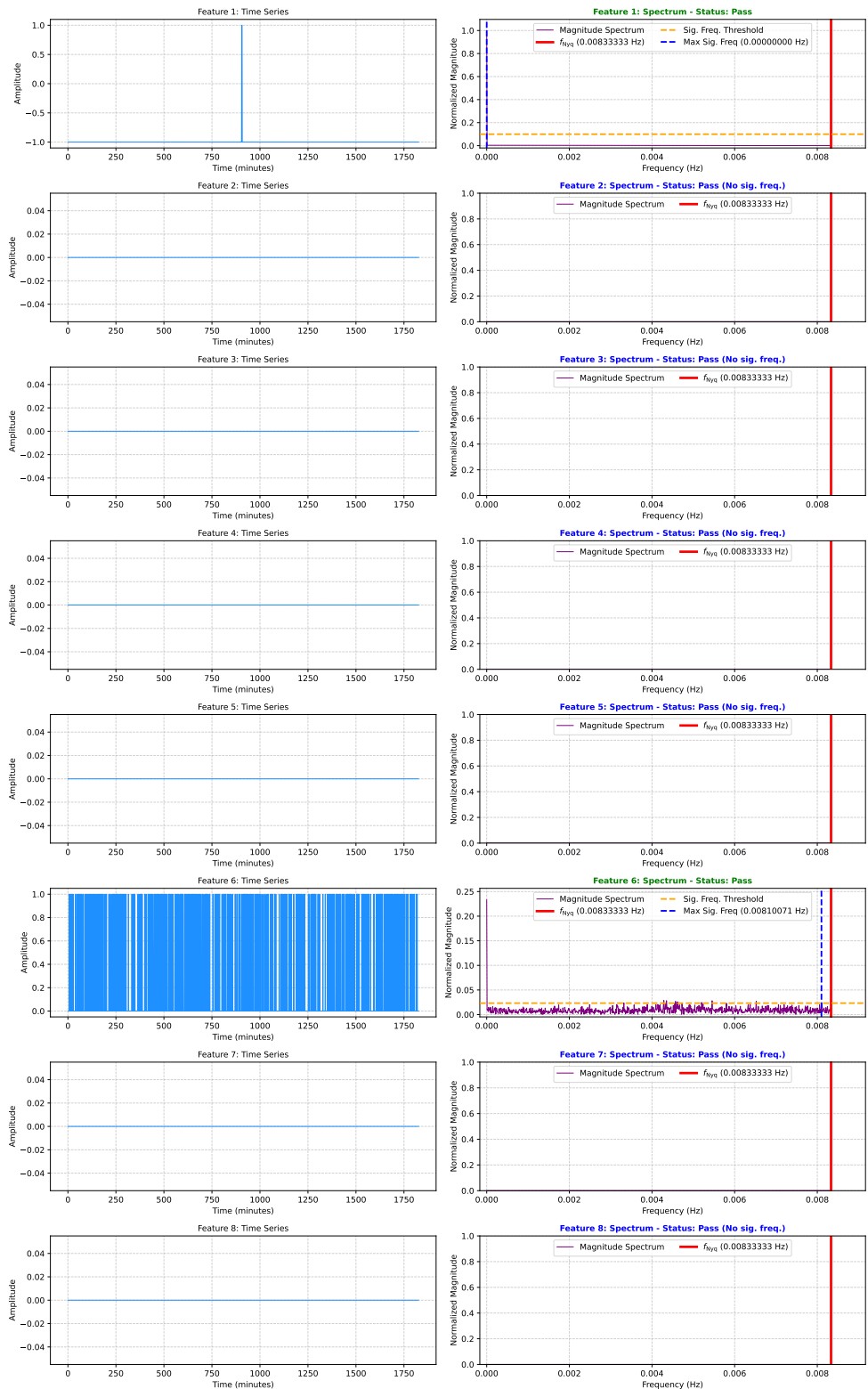

Figure 17: Nyquist criteria verification on MSL.

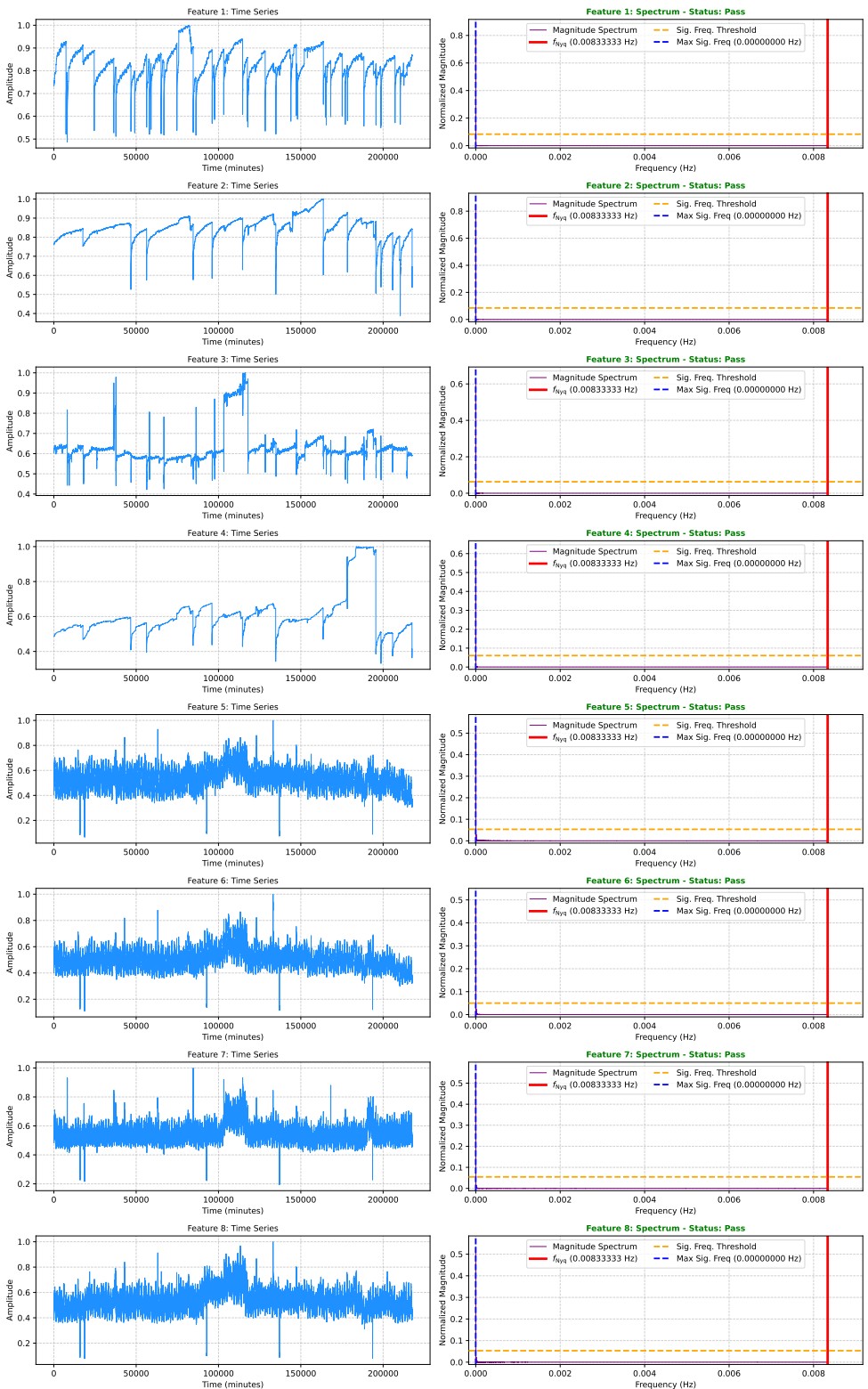

Figure 18: Nyquist criteria verification on PSM.

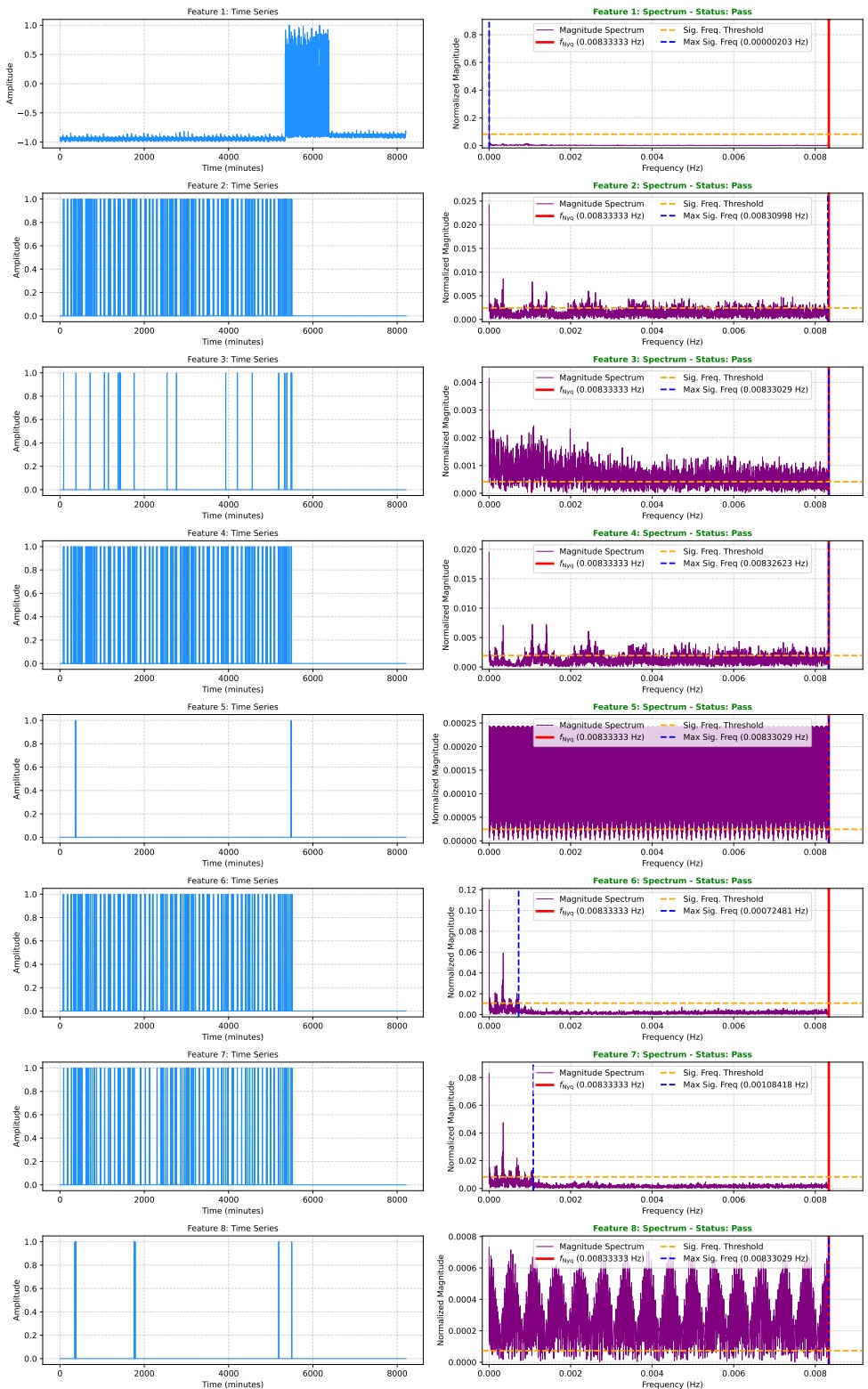

Figure 19: Nyquist criteria verification on SMAP.

### E.7 Comparison with Contrastive-based Models

As shown in Tab.1, noticeable improvements can be observed for different models. In this section, our aim is to explain that **reconstruction-based methods are also competitive candidates, so it is necessary to further optimize for better performance when conducting MTS AD.** Given these, we compare selected reconstruction-based methods with contrastive-based models, including DCdetector [66], TS-TCC [15] and CoST [57], which are also powerful tools in recent years. The results are listed in Tab.18. The results indicate that many reconstruction-based methods also outperform contrastive-based methods and become more competitive under the effect of IGAD.

Table 18: $\overline{\text{VUS-PR}}$ under five random seeds for contrastive learning model. † denotes the average value of all the mean values reported in Tab.1.

| Model | Dataset | | | |
|---|---|---|---|---|
| | **SMD** | **MSL** | **PSM** | **SMAP** |
| DCdetector | 0.0210 | 0.0096 | 0.1117 | 0.1508 |
| TS-TCC | 0.0221 | 0.0088 | 0.1064 | 0.1491 |
| CoST | 0.0217 | 0.0228 | 0.1118 | 0.1459 |

| Model | Dataset | | | |
|---|---|---|---|---|
| | **SMD** | **MSL** | **PSM** | **SMAP** |
| Mean of Contrastive-based Methods | 0.0216 | 0.0137 | 0.1010 | 0.1486 |
| Mean of Reconstruction-based Methods w/o IGAD | 0.1554† | 0.0416† | 0.1542† | 0.3223† |
| Mean of Reconstruction-based Methods w/ IGAD | **0.1752†** | **0.0920†** | **0.1592†** | **0.4056†** |

### E.8   Codes for IGAD

For better understanding of IGAD, we also provide a pseudo-code block to help better describe the IGAD working flow during the training process, as shown in Code.1.

```python
# First, we define:
# f: The training model initialized by f =
↪  Model(parameters...).to(device)
# f_copy (the defined f'): The frozen model initialized by
# f_copy = Model(parameters...).requires_grad(False).to(device)
def train_in_a_single_iteration(f, f_copy, data):

    # f_copy is the frozen of current training model
    f_copy.load_state_dict(f.state_dict())

    recon_data = f(data)

    z = get_augumented_data(data)  # Get z^i for x^i
    fz = f(z)  # f(z^i)
    f_z = fz.detach()  # f'(z^i)
    ff_z = f(f_z)  # f(f'(z^i))
    f_fz = f_copy(fz)  # f'(f(z^i))

    # Calculate losses
    loss_rec = (recon_data - data).pow(2)  # Reconstruction
    loss_idem = (f_fz - fz).pow(2)  # Idempotent
    loss_tight = -(ff_z - f_z).pow(2)  # Tightness
    #  loss_auxiliary if exists

    # Optimize for losses
    loss = lambda_rec * loss_rec + lambda_idem * loss_idem +
    ↪  lambda_tight * loss_tight  # loss_auxiliary if exists
    opt.zero_grad()
    loss.backward()
    opt.step()
```

Listing 1: Python implementation for IGAD.

## F   Limitation and Future Work

Although significant improvements have been observed, there remain unexplained performance drops in a limited number of experiments. This warrants further investigation to identify the underlying causes. Meanwhile, the slightly larger standard deviation observed in certain cases suggests the need to optimize the training process to achieve more stable convergence in our future work. The workflow of IGAD also inspires us to explore potential strategies to reduce computational complexity for large models, such as OFA [74].

## G   Impact Statements

This paper presents work focused on advancing the field of multivariate time series anomaly detection, with applications in healthcare, finance, and industrial monitoring. Although the ethical implications of anomaly detection are generally well-established, the misuse of such methods in sensitive areas could lead to privacy concerns and unintended biases in decision-making. We believe that this research contributes to improving anomaly detection techniques, improving system reliability, and early warning capabilities. Specific ethical issues are not identified beyond these general considerations.

