# OpenReview forum: "Multivariate Time Series Anomaly Detection with Idempotent Reconstruction"
_NeurIPS.cc/2025/Conference — NeurIPS 2025 poster_

### Official Review · Reviewer_TXdD · 2025-06-21

**Clarity:** 3
**Significance:** 4
**Originality:** 4
**Rating:** 5
**Confidence:** 4

**Summary:**

The authors propose a new framework to enhance general reconstruction-based time series anomaly detection methods. The main idea is to complement the training objective with an idempotent loss term and a tightness loss term. The former aims to avoid the model of being too affected by noisy normal data in the training phase, while the latter focus on tightening the learned manifold to mitigate the model capability of reconstructing abnormal instances. The proposal is evaluated with 15 models from the literature on 4 datasets. Most results indicate gains (sometimes by a large margin) when using the new strategy when compared to not using it.

**Questions:**

- Section 3.3.2: When explaining the idempotent objective, there is no obvious reason to pursue the frequency domain reconstruction path. Could, for instance, you elaborate on that? Is just a way to avoid the inclusion of trainable parameters? Could time-domain PCA be considered? This discussion could be a valuable inclusion to the text.

- Section 3.3.3: The "Inverse Effect" is the main idea behind the tightness objective. However, it is not straightforward to link the accident expansion of the learned manifold and this operation. This section can be enhanced with a more thorough discussion on this.

- Fig. 1: Can you make the font size slightly bigger?

- Some references indicate the arXiv version, instead of the published version: [19, 27, 57, 69].

**Ethical Concerns:**

["NO or VERY MINOR ethics concerns only"]

**Final Justification:**

The authors provided detailed responses and clarifications to my questions. I maintain my favorable score.

**Limitations:**

The authors state the limitations of their work.

**Paper Formatting Concerns:**

None.

**Quality:**

4

**Strengths And Weaknesses:**

The paper is well organized and written, with each component of the IGAD framework detailed both visually and textually. The experiments are comprehensive, covering several recent models from the literature and presenting strong results when the proposed approach is included. Moreover, the fact that IGAD can used to enhance very distinct reconstruction-based methods is very interesting. The robustness and sensitivity analysis further highlight the advantages of IGAD. The ablation section is very detailed and justifies the methodological choices. I also praise the authors for the provided code and supplementary material, with several additional results.

Despite not presenting serious weaknesses, it would be of interest to further explore the intuition behind the importance of the idempotent loss term. In practice, does it prevent the cases illustrated in Fig. 1? Perhaps some toy examples could be used to emphasize this point. Moreover, in the light of the work by Liu and Paparrizos, 2024 [31], which praises simpler models, it would be useful to verify how IDAG behaves with models such as standard AEs and VAEs.

The need of tuning the five hyperparameters in the IGAD objective could hinder its use in practice. Tab. 4 indicates that it would be difficult to fix default values across distinct datasets. Nevertheless, the authors argue that this choice may be reduced to three hyperparameters (the ones introduced by IGAD) and indicate good search intervals, which can be informed to standard optimization tools, such as Optuna.

---

> ### Author Rebuttal · Authors · 2025-07-29
>
> Thank you for your insightful and constructive suggestions, which inspires us to further improve our work. We have carefully considered your points and provide our detailed responses below.
> # Main Comments:
> > **W1: Further explore the intuition behind the importance of the idempotent loss term. In practice, does it prevent the cases illustrated in Fig.1?**
>
> **A-W1:** Thank you for this insightful suggestion. First, we wish to clarify the role of the idempotent loss term ($\mathcal{L}\_\text{idem}$) in addressing the challenges illustrated in Fig.1. In practice, the primary intuition behind the idempotent loss term is to enhance the ability to balance robustness against natural data noise with sensitivity to true anomalies, which directly corresponds to the scenario depicted in **Case 3 of Fig.1,** where normal data affected by noise becomes difficult to distinguish from subtle anomalies. Our method generates the augmented vector $z$ by applying a Fast Fourier Transform to a normal time window, sampling from the distributions of its frequency-domain features (real and imaginary parts), and then applying an Inverse FFT. This process ensures that $z$ retains the intrinsic patterns of the normal series while incorporating structured, naturalistic noise. By enforcing the idempotent constraint $f(f(z))=f(z)$, we expect the model to learn a stable mapping for these ***noisy-but-fundamentally-normal*** samples. In essence, it teaches the model to see through the noise, recognize the underlying normal structure within the perturbed signal, and map it to a consistent point on the target manifold. This enhances the model's ability to recognize core normal dynamics in the presence of real-world disturbances, thereby improving robustness without being overly sensitive to such fluctuations.
> The issue of over generalization, shown in **Case 1**, is handled by the tightness objective $\mathcal{L}\_\text{tight}$, which is divided from the original idempotent loss term. We further explore the effect and intuitive of this **in our response to Q2. We have also added more visualization examples in our work to show this.**
>
> > **W2: Verify how IDAG behaves with models such as standard AEs and VAEs.**
>
> **A-W2:** Thank you for your valuable suggestions and highlighting this important findings in [31]. We also agree that demonstrating the utility of IGAD for simpler base models is crucial. Here, we implement standard AE and VAE, recording their behaviours before and after applying IGAD to conduct further exploration. The experimental results are shown in the following table. We can observe that IGAD also shows positive effects on models with simpler structure, including AE and VAE here. Meanwhile, it can be noticed that:
> 1. Before applying IGAD, VAE shows a higher level than AE on four datasets;
> 2. After applying IGAD, AE with IGAD shows higher improvements than VAE with IGAD, and even has better performance than VAE with IGAD;
> 3. AE and VAE achieve comparable results compared to certain complex models, which is consistent with the mentioned findings in [31].
> | Model | Dataset |  |  |  |  |  |  |  |  |  |  |  |
> | :---: | :---: | :---: | :---: | :---: | :---: | :---: | :---: | :---: | :---: | :---: | :---: | :---: |
> |  | SMD |  |  | MSL |  |  | PSM |  |  | SMAP |  |  |
> |  | w / o IGAD | w / IGAD | $p$-value | w / o IGAD | w / IGAD | $p$-value | w / o IGAD | w / IGAD | $p$-value | w / o IGAD | w / IGAD | $p$-value |
> | AE | 0.3113±0.0029 | 0.3823±0.0053 | *** | 0.0083±0.0003 | 0.0370±0.0208 | *** | 0.1437±0.0001 | 0.1640±0.0098 | *** | 0.0969±0.0007 | 0.9149±0.0784 | *** |
> | VAE | 0.3556±0.0041 | 0.3556±0.0037 | *** | 0.0086±0.0004 | 0.0087±0.0002 | *** | 0.1464±0.0004 | 0.1465±0.0003 | *** | 0.0930±0.0014 | 0.0947±0.0014 | *** |
>
> We conclude this as that VAE itself has a regularization by KL divergence on the latent space, which requires the prior distribution of latent vector $z$, denoted as $p_\theta(z)$, is close to the standard normal distribution, which leaves non-regularization AE with more room for improvements.
>
> > **W3: Tab. 4 indicates that it would be difficult to fix default values across distinct datasets.**
>
> **A-W3:** Thank you for your concerns on choosing the optimal parameters for different datasets. Yes, we have provided a hyperparameter analysis in Appendix D and Appendix E.3 to help address this. For example, we have found that $\lambda_\text{idem}$ in the range [0.1, 0.5], not too strict $\lambda_\text{tight}$ and larger values for $\alpha$ can be given priority consideration in the most cases. We hope these analysis can be helpful for reducing practical difficulty and further applying IGAD.
>
> > **Q1: The reason for frequency reconstruction path and the effect of time-domain PCA?**
>
> **A-Q1:** Thank you for this constructive suggestion which inspires us to further explore our design in IGAD. In fact, we choose frequency domain reconstruction path considering several advantages of this, not limited to avoid additional trainable parameters. First, concretely, these advantages can be stated as follows:
> 1. Many multivariate time series are defined by their periodic dynamics. FFT decomposes a signal into a basis of functions, which is a powerful mathematically language for describing such patterns. Perturbing the signal in the frequency domain by sampling from the same distribution of real part and image part allows for direct manipulation of these crucial components, resulting in a complex, structured, and globally consistent form of noise in the time domain that is more realistic than simple additive noise, which helps us to generate the ***noisy-but-fundamentally-normal*** samples;
> 2. The basis functions used in FFT are fixed and independent of specific data. This makes the method more flexible, eliminating the need to learn basis functions from data, offering higher computational efficiency, and being easy to implement. This aligns with our design principle for IGAD to be a lightweight, flexible module that does not require additional training parameters.
>
> Second, we agree with you and believe that the time-domain PCA is a powerful candidates for this part. It also has its own advantages:
> 1. Unlike the fixed basis functions of FFT, the principal components of PCA are learned entirely from the data itself. This means that PCA can capture the important linear variation patterns in a specific dataset;
> 2. PCA can find the optimal linear low-dimensional representation of time series. By perturbing only a few of the most important principal components, it is possible to selectively modify the main structure of the signal.
>
> After comparison, we realized that the time-domain PCA may suffer the following issues:
> 1. In contrast, although PCA can identify the main directions of data changes, these directions are linear and may not effectively capture the complex dynamics in the signal;
> 2. The principal components of PCA represent the directions with the greatest variance in the data. Although perturbations to these components are effective, their impact may be more limited to the main linear patterns in the data and may not necessarily produce equally global and structural complex perturbations;
> 3. PCA requires an additional, data-driven step to learn its principal component basis from time series data;
> 4. The choice of the number of principle component $k$ can be viewed as an additional constrains. For example, if $k$ is too small, augmented $z$ may only reflect the most significant patterns of change in the data, resulting in simplified $z$ that might lack diversity; if $k$ is too large, it may introduce noise that already exists in the original data (reflected in principal components with small variance) into the perturbation process, and may not achieve the desired effect of retaining normal patterns and injecting randomness.
>
> > **Q2: More thorough discussion on the tightness objective and the Inverse Effect.**
>
> **A-Q2:** Thank you for pointing this out. The link between the "Inverse Effect" of our tightness objective and preventing the "accidental expansion" of the learned manifold is important for addressing the issue of over generalization. As described in Sect.3.3.3 and Fig.3 of the paper, a standard idempotent mapping causes the gradient to flow through both $f$ during optimization. **This produces a red path gradient that tends to “stretch” the manifold outward to include every $z$.** If the training data contains unlabeled outliers or the model has overly strong decoding capabilities, this expansion may erroneously include the outlier region within the normal manifold, leading to over generalization. To address this, our tightness objective, introduces an adversarial training dynamic by using a frozen copy of the model, $f'(\cdot)$. This objective explicitly rewards the current model $f(\cdot)$ for ***maximizing*** the distance between the reconstruction $f(f'(z))$ and the input $f'(z)$, where $f'(z)$ represents a point on the old manifold. **This push operation is the "Inverse Effect." It directly restrict the tendency of expansion by preventing the model from learning a lazy or overly-inclusive mapping, i.e., $f(z)=z$, by *maximizing* the distance between the first mapping $f'(z)$ and $f(f'(z))$.** The model is constantly required to refine the representations produced by its past self, forcing it to learn a more precise and compact definition of normality. We will further integrate and update into our manuscript, ensuring the connection is clear to the reader for better understanding.
>
> > **Q3 & Q4: Font size and reference version?**
>
> **A-Q3 & Q4:** Thank you for pointing these out. We will increase the font size in Fig.1 for better readability in our next version. Meanwhile, After searching, these four papers have been published on ICML 2024, ICLR 2014, ICLR 2025 and IJCAI 2024. We will recheck the references in the text and update them to the published versions.

---

> > ### Comment · Reviewer_TXdD · 2025-08-02
> > **Thanks for the clarifications**
> >
> > I thank the authors for the detailed responses and clarifications. I maintain my favorable score.

---

> > > ### Author Response · Authors · 2025-08-03
> > >
> > > We sincerely thank you for your further reply. Your constructive comments have been very helpful for improving the clarity and quality of our work. Thank you again for your time and insightful suggestions.

---

### Official Review · Reviewer_KChG · 2025-06-27

**Clarity:** 2
**Significance:** 2
**Originality:** 3
**Rating:** 3
**Confidence:** 5

**Summary:**

This paper introduces the use of a idempotent generative network framework for multivariate time series anomaly detection in order to address the over generalization problem of reconstruction based time series anomaly detection methods, i.e., reconstruction models that rely on the assumption that abnormal time series would be poorly reconstructed by a generative model that has only seen normal data, may not in practice be the case. Two hypotheses are provided for this behaviour: 1) there is contaminated data in the normal training samples that are impractical to remove, and 2) reconstruction models that have good regularization/generalization properties could generalize well to anomalies. The proposed solution, IGAD, adds an idempotent loss and a tightness loss to ensure a tight mapping of normal samples to the target manifold. Improvements shown over many base reconstruction models and ablation studies demonstrate the effectiveness of this approach.

**Questions:**

- Since the original IGN was tested on MNIST, can the proposed framework be applied to image data?
- What is the justification for doing reconstruction on x and IGAD on z, why not use z for reconstruction, or use both x and z for IGAD?
- How do we know that the tightness objective implicitly excludes anomalies from the manifold, when we don't have knowledge of the nature of the anomalies?
- Are f and f' two separate networks of different weights? By the definition of idempotent, shouldn't f and f' be the same function? Is it the same network but in different points of time? Please clarify. It would also clear up how Eq (9) wouldn't cancel out.
- The MAE over MSE experiment is interesting. Since 0.1684 VUS-PR is the performance of no IGAD with MSE, is it possible that the 0.2080 VUS-PR is attributed to the MAE loss instead of IGAD - this is potentially a weakness.

**Ethical Concerns:**

["NO or VERY MINOR ethics concerns only"]

**Final Justification:**

I maintain my score as was discussed in the rebuttal.

**Limitations:**

Some limitations have been discussed.

**Quality:**

3

**Strengths And Weaknesses:**

Strengths
- Novel solution to the problem of over-generalization problem in reconstruction-based MTS AD methods
- Theoretical analysis of how regularization leads to over-generalization and good reconstruction of anomalies (undesired)
- Framework is compatible with almost every existing reconstruction-based MTS AD models, with no additional trainable parameters required
- Consistent positive improvements of the framework over the original backbones.
- A larger-than-average number of baselines is compared
- Figure 5, Table 2, and Table 3 are impressive results
- Robust protocol that includes optimizing the hyperparameters of baselines

Weaknesses
- Framework is almost the same as [45], although the application in MTS AD is novel
- Proof of convergence is the same as [45]
- Does not mention other work in the contaminated data in MTS AD, e.g., [Ref 1], and there are more in the general context of Anomaly detection. Note that the original IGN was tested on MNIST, so it is expected to cover contaminated AD in the non-TS field as well.
- Does not achieve state-of-the-art results compared to [63]
- Limited number of datasets (only 4, and they are small datasets). It would be more convincing to follow [63] and include SWaT, NIPS-TS, and UCR. More datasets would be favoured over a large number of baselines.
- The delta_model percentage should be w/IGAD minus w/oIGAD as opposed to w/IGAD divided by w/oIGAD
- Justify why VUS-PR is favoured over the other evaluation metrics. Can also report reconstruction error on abnormal data to be meaningful.
- Figure 4b) is not very impressive looking compared to Figure 4a)

[Ref 1] Contaminated Multivariate Time-Series Anomaly Detection with Spatio-Temporal Graph Conditional Diffusion Models


Minor weaknesses:
- Inconsistent use of the terms "over generalization" and "over generation"
- The distinction between x and z is not clear in the introduction, but becomes clearer later.
- Please clarify if a sliding window is used

---

> ### Author Rebuttal · Authors · 2025-07-30
>
> Thank you for your insightful suggestions. We provide our detailed responses below.
> # Main Comments:
> > **W1 & W2: The framework and proof of convergence are same as [45].**
>
> **A-W1 & W2:** Thank you for this concern. IGAD differs from [45] in its design concept, objectives, and its function as a task-specific plugin. Please refer to our response to **Reviewer vmjz on W3** for details due to page limits. Meanwhile, we redefine math objects in our convergence proof to validate that our target manifold learns the distribution of normal data only.
>
> > **W3: More works should be mentioned.**
>
> **A-W3:** Thank you for your suggestions that more comprehensive references should be included to further improve the integrity. After searching from 2023, we have found the following works as well as you kindly mentioned:
> 1. Image, Tabular or Video: BiGAN, IAD, SAE-CRIT, LOE.
> 2. TS: NegCo, TSAD-C, COUNTA, NormFAAE.
>
> These outstanding works primarily process data contamination through methods such as sample weighting, designing robust loss functions, or explicitly separating clean and contaminated data for modeling. However, IGAD, from a manifold geometry perspective, uses adversarial objectives to tighten the normal data manifold, implicitly excluding contaminated and anomalous data.
>
> > **W4: Does not achieve state-of-the-art results compared to [63].**
>
> **A-W4:** The primary contribution of our paper is not a new, end-to-end SOTA model, but a novel, plug-and-play module, IGAD, without introducing additional trainable parameters. To demonstrate this , we included a comparison with contrastive-based models like [63] in **Tab.17 (Page 48)**. The results show that the average VUS-PR of reconstruction-based models with IGAD is higher than the average of contrastive-based methods.
>
> > **W5: Limited datasets.**
>
> **A-W5:** Thank you for this suggestion. SWaT and UCR datasets are added to further verify IGAD. We would greately appreciate it if you could refer to **Reviewer vmjz on W1.**
>
> > **W6: The calculation method for $\Delta_\text{model}$.**
>
> **A-W6:** Thank you for this suggestion. In the current version, $\Delta_\text{model}$ is reported as the improvement ratio, calculated by:
>
> $$
> \frac{\text{VUS-PR}\_\text{w / IGAD}-\text{VUS-PR}\_\text{w / o IGAD}}{\text{VUS-PR}\_\text{w / o IGAD}} * 100.
> $$
>
> We chose a relative measure to better reflect the impact of improvements, especially for weak baselines. However, we agree that absolute results provide a direct reflection of gains and will add them to our manuscript for a more complete evaluation.
>
> > **W7: Why VUS-PR is favoured and report reconstruction errors on abnormal points.**
>
> **A-W7:** This finding is highlighted in [31]. Due to the constrains on uploading images and space limit, we would greatly appreciate it if you could refer to the **Fig.5 in [31].** The other metrics mainly suffer three issues: bias, indiscrimination and lack of adaptability. VUS-PR can effectively handle these issues, emerging as the most robust metric. We guarantee that we will add a section to further explain this. We also report the differences on average errors for abnormal points in the following table. Most cases show that model with IGAD tend to generate larger errors for anomalies. For readability, these values have been multied by $10^3$.
>
> | Model | Errors (w/ - w/o)|  |  |  |
> |---|---|---|---|---|
> |  | SMD | MSL | PSM | SMAP |
> | 1 | 2.0  | 0.3  | 123531.2  | 4.0  |
> | 2 | 11.5  | 12.2  | 34.0  | 103.2  |
> | 3 | 2.2  | 6.4  | 0.3  | -0.2  |
> | 4 | 2.6  | 1.2  | 0.1  | -1.2  |
> | 5 | 0.9  | -0.2  | 0.1  | 1.1  |
> | 6 | 4.7  | 98.8  | 15.3  | 7.3  |
> | 7 | 3.5  | 4.0  | 1.1  | 6.1  |
> | 8 | 438.4  | 14.4  | 5.6  | 3.2  |
> | 9 | 7.1  | 0.1  | 0.1  | 14.4  |
> | 10 | 485.2  | 395.4  | 358.6  | 95046.0  |
> | 11 | -64.9  | 68.7  | 12.4  | -78.2  |
> | 12 | 2463.1  | 231.8  | 59.4  | 2008.7  |
> | 13 | -0.1  | 4.4  | 0.1  | 332.2  |
> | 14 | -25.6  | 0.1  | -2.7  | 147.8  |
> | 15 | 110.6  | 134.4  | 10.8  | 103.6 |
>
> > **W8: Fig.4(b) is not very impressive looking compared to Fig.4(a).**
>
> **A-W8:** Thank you for your careful observation. We think that this is another reflect of the effect of IGAD and conclude this as the trade-off between over-fit to clean data and the ability for high-level noise. We would greatly appreciate it if you could refer to our response to **Reviewer vmjz on Q2** for more details.
>
> > **W9 & W10: Inconsistent term and the distinction on $x$ and $z$.**
>
> **A-W9 & W10:** Thank you pointing these out. We have located the terms and fixed them. Meanwhile, to improve readability, we will add a clarifying sentence in the introduction for final $x$ and $z$ to provide readers with a clearer initial understanding.
>
> > **W11: If a sliding window is used.**
>
> **A-W11:** Yes, as we stated in Sect.3.2, the whole dataset is split into different windows with the size of $w$, and the stride is 1 according to the global and unified settings in [31].
>
> > **Q1: Can the proposed framework be applied to image data?**
>
> **A-Q1:** Thank you for this constructive question. We think that IGAD can be applied into image AD with appropriate adaption:
> 1. Image data also suffer the problem of contaminated data due to anomaly leakage or ambiguous boundaries, shown as works in W3, and over generalization from the constrains on reconstruction. These are consistent with the issues IGAD can address.
> 2. While our FFT-based augmentation is technically applicable to images via 2D-FFT, it is designed for the global and periodic feature of time series, and have more physical meaning. A direct application may be suboptimal, as its global disturbance might influence the model to detect highly localized anomalies often found in images.
>
> In conclusion, IGAD framework is transferable and its strategy for generating $z$ would need to be re-designed for images.
>
> > **Q2: What is the justification for doing reconstruction on x and IGAD on z?**
>
> **A-Q2:** We use reconstruction on $x$ and IGAD on $z$ because they are responsible for different goals. Concretely:
> 1. Reconstruction on $x$: This is the fundamental objective. It forces the network to learn to accurately reconstruct the original normal time series data and define the core of the normal data manifold.
> 2. IGAD on $z$: The vector $z$ is a perturbed version of $x$, representing a point nearly the ***boundary*** of the normal manifold. The idempotent objective then teaches the model that these slightly perturbed, yet still normal, points should be mapped stably onto the manifold. This ensures the learned manifold is robust to noise. Simultaneously, the tightness objective acts as a crucial adversarial regularizer. It prevents the manifold from expanding to achieve stability for addressing over generalization.
> 3. If we were to use $z$ for reconstruction, i.e., $f(z)=z$, we would be directly teaching the model to reproduce noise, which is contradictory. Conversely, applying IGAD to $x$ would be less effective, as $x$ is already assumed to be an ideal point on the manifold, and the objective could be directly satisfied.
>
> > **Q3: How do we know tightness objective excludes anomalies from the manifold?**
>
> **A-Q3:** We think, in fact, we do not need prior knowledge about anomalies:
> 1. Our principle is that the space of normal data is typically compact and has a specific, learnable structure, while the space of abnormal data is vast, diverse, and essentially unbounded. Therefore, we do not learn to identify every possible anomaly, but learn an tight and precise model of normality. Points that fail to conform to this strict model are flagged as anomalies.
> 2. IGAD is designed to enforce through the manifold geometry. The tightness objective works adversarially to prevent the manifold from loosening its boundaries to accommodate points that are not central to the normal data distribution, penalizing the manifold expansion. Consequently, the diverse and unseen anomalies, which by definition do not share this compact structure, will fall far from this tightened manifold.
> 3.  Meanwhile, more visualization results that support this can be found in **Fig.10 (Page 38)**. After applying IGAD, normal points (blue) are closer and abnormal points (red) are pushed further from the center of normal area. This is due to the effect of tightness objective that the manifold are tighten so that more anomalies are dropped out from this area.
>
> > **Q4: Relation between $f$ and $f'?$**
>
> **A-Q4:** $f'$ is the frozen and non-trainable copy of $f$, so they are two separate models with the same weights at all the time and share the same network architecture. $f'$ facilitates more efficient training rather than frequently opening or closing the gradient of $f$. Meanwhile, at the beginning of each training iteration, their weights are identical as $f'$ will copy the weights of $f$. **As indicated in the code example in Page 49,** it can also be verified $f'$ is the frozen copy of $f$, their weights are the same at all the time. This is also why the objectives in Eq.9 do not cancel out: Since $f'$ is fixed and frozen during gradient calculation, $\mathcal{L}\_\text{idem}$ and $\mathcal{L}\_\text{tight}$ create adversarial computational graphs to update $f$.
>
> > **Q5: Loss function or IGAD?**
>
> **A-Q5:** Thank you for this insightful question. We appreciate the opportunity to clarify this. We agree with you that IGAD with MSE loss shows the same results as the baseline (0.1684, Tab.2 vs. Tab.3). We interpret this as a key finding: IGAD is destabilized by the noise-sensitive MSE loss but is enabled by the robust MAE loss. The evidence for IGAD's contribution is in Tab.3, which show its effect within the optimal MAE setting. When our core IGAD objectives are removed, the average performance drops noticeably. This performance decrease, even while using a robust loss, demonstrates that the IGAD module is the primary driver of the improvement, with the MAE loss acting as the necessary stabilizing factor for it to function effectively.

---

> > ### Comment · Reviewer_KChG · 2025-08-08
> >
> > Thanks for your clarification.  However, I maintain the overall score as the rebuttal has not convinced me about the main weaknesses (i.e., weak performance and almost identical model compared to [45]).

---

> ### Author Response · Authors · 2025-08-06
>
> Dear Reviewer KChG,
>
> We sincerely thank you for your insightful and constructive suggestions on our manuscript.
>
> We have provided additional information in response to your feedback, which we hope addresses your concerns. Could you please kindly let us know if our clarifications have been helpful or if you have any further questions? Your further feedback would be highly valuable for us to continue improving the clarity and quality of our work.
>
> Thank you again for your time and suggestions.
>
> Best regards,
>
> The Authors of Submission 16078

---

> ### Author Response · Authors · 2025-08-08
>
> Dear Reviewer KChG,
>
> Thank you for your valuable time and insightful feedback.
>
> Following your suggestions and questions, we have submitted a supplementary response with additional experiments and clarifications, hoping to address your concerns about IGAD.
>
> Your further feedback would be highly valuable for us to continue improving the clarity and quality of our work. We would be very grateful if you had a moment to consider these additional clarifications before the discussion period ends.
>
> Thank you once again for your time and consideration to the review process.
>
> Best regards,
>
> The Authors of Submission 16078

---

> ### Author Response · Authors · 2025-08-09
>
> **Thank you for your further comments and questions.**
>
> > **W1: Weak performance.**
>
> **A-W1:** We think this question relates to ***Weakness 4: Does not achieve state-of-the-art results compared to [63].***
>
> We wish to clarify that the primary contribution of our paper is not a new, end-to-end SOTA model, but a novel, plug-and-play module, IGAD, which is designed to enhance existing reconstruction-based models without introducing additional trainable parameters. **To demonstrate this, we included a comparison with contrastive-based models like DCdetector [63] in Tab.17 in the Appendix (Page 48).** The results show that on all four datasets, the average VUS-PR of reconstruction-based models with IGAD is higher than the average of contrastive-based methods. For instance, on the SMD dataset, our enhanced models with IGAD achieve an average VUS-PR of 0.1752, outperforming the 0.0216 from contrastive methods. In addition, if we compare with Tab.1 and Tab.17, models with IGAD achieve better performance than contrastive-based methods including [63] in most cases. Each model has its own design principle, and we hope to improve the overall performance of reconstruction-based methods through IGAD enhancements. For the additional two larger datasets included, i.e., SWaT and UCR, the effects of IGAD can also be observed.
>
> > **W2: Almost identical model compared to [45].**
>
> **A-W2:** In addition to the clarifications in our answer for ***Weakness 1***, we wish to further clarify this from the following aspects:
>
> **1. Different Goals for IGAD and IGN:** The primary goals of the two methods are different. The original IGN is a generative model designed to produce novel samples within a specific domain, such as new images. In contrast, our IGAD module is not used for generation. It is designed to address two specific challenges in MTS AD: over generalization and the trade-off between robustness and sensitivity. Our work adapts the concept of idempotency and performs task-specific modification, using it as a tool for regularization and manifold definition in the context of anomaly detection.
>
> **2. Different Physical Meaning for Augmented $z$:** The original idempotent generative network samples $z$ mainly for generating new images. However, we apply a FFT to a window of normal time series, introduce randomness through resampling in the frequency domain, and then use an IFFT to produce the final augmented vector $z$. This design injects the intrinsic patterns and spectral features of real normal samples into $z$ while simulating the natural noise and fluctuations of real-world data acquisition. This approach grounds the latent vector in the properties of the input data.
>
> **3. Adapted Optimization Objectives:** The roles of the $\mathcal{L}\_\text{idem}$ and $\mathcal{L}\_\text{tight}$ in our framework are defined for important issues in the field of MTS AD, rather than for ensuring generation stability. Concretely, $\mathcal{L}\_\text{idem}$ is used to ensure the model learns to recognize and reconstruct the augmented vectors $z$ that carry the intrinsic features of normal data, thus making the target manifold $\mathcal{M}\_\text{target}$ more robust in covering diverse normal instances. $\mathcal{L}\_\text{tight}$ is also repurposed for dropping out potential abnormal instances from $\mathcal{M}\_\text{target}$, which prevents over generalization.
>
> **4. A Flexible and Applicable Module:** IGAD is designed as a plug-and-play module instead of an end-to-end model, which can be combined with reconstruction-based methods without introducing additional trainable parameters. This makes it flexible, efficient, and broadly compatible with different reconstruction-based AD models.
>
> Thank you again for your valuable time and suggestions. We hope these clarifications could help further address your concerns. We would also be willing to provide further clarifications in the remaining rebuttal period.

---

### Official Review · Reviewer_vmjz · 2025-07-01

**Clarity:** 3
**Significance:** 2
**Originality:** 2
**Rating:** 4
**Confidence:** 3

**Summary:**

This paper proposes a novel module named IGAD to address the risk of over generalization in reconstruction-based methods for multivariate time series anomaly detection. IGAD draws on the ideas of idempotent generative networks and makes reasonable improvements for the task. And this paper presents sufficient theoretical analyses and experimental results.

**Questions:**

Q1: In some datasets and for certain models, the use of IGAD has instead led to performance degradation. Considering that these are results after hyperparameter search, is this related to the compatibility between IGAD and some methods? For example, CAE-M shows limited improvement on the PSM and SMAP datasets and even performance degradation on the SMD dataset; ModernTCN exhibits significant performance drops on both the PSM and SMAP datasets.
Q2: Why does the method with IGAD perform worse when the noise weight is smaller in Figure 4(b)?
Q3: The models marked with * are concentrated on the MSL dataset. Is this related to the characteristics of the dataset?

**Ethical Concerns:**

["NO or VERY MINOR ethics concerns only"]

**Final Justification:**

I recommend Borderline accept. My concerns are addressed after author rebuttal.

**Limitations:**

yes

**Quality:**

3

**Strengths And Weaknesses:**

Strengths:
S1: The paper has reasonable motivation.
S2: The paper is easy to follow and well-written.
S3: The paper has sufficient and detailed theoretical analysis.

Weaknesses:
W1: The number of datasets is slightly small.
W2: The increased training time caused by IGAD limits its practicality, especially the more complex the base methods and datasets used with IGAD, the greater the additional time overhead introduced by IGAD.
W3: The method in the paper is based on idempotent generative networks, which limits its innovativeness.

---

> ### Author Rebuttal · Authors · 2025-07-29
>
> Thank you for your insightful and valuable suggestions. We provide our feedbacks as follows.
> # Main Comments:
>
> > **W1: The number of datasets is slightly small.**
>
> **A-W1:** Thank you for your suggestions that more datasets should be taken into consideration to further demonstrate the effectiveness of IGAD. We have added two larger datasets, SWaT and UCR, and shown the results in the following table.
>
> | **Model** | **Dataset** |  |  |  |  |  |
> |:---:|:---:|:---:|:---:|:---:|:---:|:---:|
> |  | UCR |  |  | SWaT |  |  |
> |  | w / o IGAD | w / IGAD | $p$-value | w / o IGAD | w / IGAD | $p$-value |
> | CATCH | 0.0017  | 0.0019 ↑ | *** | 0.1472  | 0.1475 ↑ | *** |
> | M2N2 | 0.0037  | 0.0038 ↑ | *** | 0.4267  | 0.4442 ↑ | *** |
> | FITS | 0.0071  | 0.0203 ↑ | *** | 0.1448  | 0.1501 ↑ | * |
> | ModernTCN | 0.0028  | 0.0022 ↓ | *** | 0.1518  | 0.1523 ↑ | *** |
> | Peri-midFormer | 0.0018  | 0.0019 ↑ | *** | 0.1449  | 0.1445 ↓ | *** |
> | SARAD | 0.0026  | 0.0070 ↑ | *** | 0.1769  | 0.1780 ↑ | *** |
> | TimesNet | 0.0122  | 0.0124 ↑ | ** | 0.1302  | 0.1303 ↑ | *** |
> | OFA | 0.0157  | 0.0161 ↑ | *** | 0.1221  | 0.1222 ↑ | *** |
> | A.T. | 0.0046  | 0.0056 ↑ | *** | 0.1413  | 0.1622 ↑ | *** |
> | FGANomaly | 0.0020  | 0.3818 ↑ | *** | 0.1808  | 0.1809 ↑ | *** |
> | CAE-M | 0.6174  | 0.6311 ↑ | *** | 0.1804  | 0.2011 ↑ | *** |
> | MTAD-GAT | 0.0015  | 0.0021 ↑ | *** | 0.1540  | 0.1543 ↑ | *** |
> | OmniAnomaly | 0.0016  | 0.0309 ↑ | *** | 0.1768  | 0.1831 ↑ | *** |
> | MSCRED | 0.0021  | 0.0027 ↑ | *** | 0.1804  | 0.1802 ↓ | ** |
> | DAGMM | 0.5715  | 0.6481 ↑ | *** | 0.1804  | 0.2001 ↑ | *** |
>
> It can be found that 27 out of 30 additional experiments show improvements, which helps further demonstrate the effectiveness of IGAD.
>
> > **W2: The increased training time limits its practicality**
>
> **A-W2:** Thank you for your concerns about the efficiency. First, we agree with you that it will cost more time for complex methods or datasets during training. Second, we wish to clarify that:
>
> 1. **The additional computational overhead from IGAD only occurs during the training phase.** During inference, IGAD introduces no extra computational steps, meaning the inference efficiency is identical to that of the original base model. Given that anomaly detection systems often follow a "train once, deploy long-term" paradigm, in common cases, a moderate increase in training time is a reasonable trade-off for improved detection performance.
>
> 2. **An advantage of IGAD is that it introduces no new trainable parameters.** It operates by leveraging a frozen copy of the training model. This ensures that model storage and deployment costs do not increase.
>
> 3. **Our experiments show that the time overhead is controllable.** For most models, the training time increases by less than 10 seconds per epoch, and often by less than 3 seconds. We acknowledge that for larger models like OFA, the overhead is more significant, and we discuss potential optimizations for this in the future.
>
> > **W3: The method in the paper is based on idempotent generative networks, which limits its innovativeness.**
>
> **A-W3:** We admit that the idempotent generative network inspires us to conduct deeper exploration in the field of AD. However, we wish to respectfully clarify that transitioning this concept from novel sample generation to multivariate time series anomaly detection involved not only a simple application. To address the challenges in AD, such as over generalization and the balance between robustness and sensitivity, we have incorporated several key, task-specific modifications that distinguish our framework.
> 1. **First, the fundamental goal and the definition of the manifold are different.** The original work aims to learn a manifold to generate novel samples belonging to a specific domain. In contrast, our objective is to improve existing reconstruction-based models for more precise anomaly identification. Consequently, we strictly redefine the learned target manifold $\mathcal{M}_\text{target}$ as a distribution consisting only of normal time instances, which is a conceptual departure from the original work, as our framework is optimized to learn the boundary of this "normal-only" manifold.
>
> 2. **Second, the meaning for generating the latent vector $z$ is different.** The original idempotent generative network samples $z$ mainly for generating new images. However, we apply a FFT to a window of normal time series $x$, introduce randomness through resampling in the frequency domain, and then use an IFFT to produce the final augmented vector $z$. This design injects the intrinsic patterns and spectral features of real normal samples into $z$ while simulating the natural noise and fluctuations of real-world data acquisition.
>
> 3. **Third, we have repurposed the optimization objectives.** In our framework, the idempotent objective $f(f(z))=f(z)$ is not for generating new samples. Instead, its role is shifted to ensure the model learns to recognize and reconstruct the augmented vectors $z$ that carry the intrinsic features of normal data, thus making the target manifold $\mathcal{M}\_\text{target}$ more robust in covering diverse normal instances. In addition, the tightness objective $f(f(z)) \neq f(z)$ is also repurposed for dropping out potential abnormal instances from $\mathcal{M}\_\text{target}$.
>
> 4. **Finally, our IGAD is designed as a plug-and-play module that introduces no additional trainable parameters.** This makes our approach flexible and compatible with different reconstruction-based AD models.
>
> We hope these can help further clarify the innovation of IGAD and the differences from the traditional idempotent generative network.
>
> > **Q1: In some datasets and for certain models, the use of IGAD has instead led to performance degradation. Considering that these are results after hyperparameter search, is this related to the compatibility between IGAD and some methods?**
>
> **A-Q1:** Thank you for your insightful question. We agree that this is related to the compatibility between IGAD and the internal mechanisms of certain models. ModernTCN is a specialized tool, relying on a fine-tuned convolutional structure to extract features. While the manipulation of IGAD is generally beneficial in most cases, it might inadvertently alter the data representations in a way that is incompatible with the rigid learning structure of ModernTCN. The features that IGAD helps to shape may not be optimally suited for processing by the specific convolutional pathway, thus causing a drop in performance. In addition, CAE-M incorporates a Maximum Mean Discrepancy regularization term to push its feature distribution toward a Gaussian one. The objectives of IGAD, which also modify the data distribution, could conflict with the native regularization strategy of CAE-M, leading to a suboptimal result for complex multi-objective optimization problems. Finally, these cases are relatively rare, and IGAD leads to performance improvements in the majority of our experiments (52 out of 60), with statistically significant gains in most of those cases (49 out of 52), demonstrating its effectiveness and applicability.
>
> > **Q2: Why does the method with IGAD perform worse when the noise weight is smaller in Figure 4(b)?**
>
> **A-Q2:** In Fig.4(a), we show the ideal case that the model enhanced with IGAD get better performance under different noise levels. In Fig.4(b), we want to show another effect results of IGAD. The performance of IGAD is below the baseline at low noise levels (1% and 5%) but surpasses it at higher noise levels (15% and 20%). We think that this behavior validates one primary goal: to balance robustness and sensitivity. In a low-noise setting, the original A.T. model is well-tuned for the clean data patterns. IGAD, which involves generating a perturbed $z$ and optimizing for robustness, effectively acts as a form of regularization. This regularization, which is critical for noisy data, can slightly hinder the model from perfectly fitting the clean patterns, resulting in a marginal performance decrease. However, the true value of IGAD is revealed as noise increases. While the performance of baseline model degrades from 0.0307 to 0.0214, the model with IGAD maintains a highly stable performance with the trend of increase. This means that IGAD prevents the model from overfitting to clean data and equips it with a more robust understanding of the underlying data patterns in more challenging and realistic noisy scenarios.
>
> > **Q3: The models marked with * are concentrated on the MSL dataset. Is this related to the characteristics of the dataset?**
>
> **A-Q3:** Thank you for your further consideration on this. First, we agree that this is related to the dataset characteristics. The data in MSL dataset originated from spacecraft telemetry, and the normal operational patterns here are complex, variable, and noisy. In such cases, reconstruction-based models can generate a high baseline of reconstruction errors even for normal data points. These make MSL a challenging task and the error separability is already limited to some extent. Consequently, in certain cases, even though IGAD improves the VUS-PR score by the effect of regularization, the magnitude of its improvement on the score distribution is not substantial enough to be deemed statistically significant. Second, this phenomenon can also be amplified by factors such as the inherent sparsity of anomalies in the MSL dataset (MSL dataset has smaller average number and length of abnormal points than others), which affects the statistical power of the test. Third, VUS-PR can be highly sensitive to improvements at critical decision thresholds. It is valid that IGAD makes small but crucial adjustments to the anomaly scores right around the critical decision boundaries. This could significantly boost the VUS-PR value without causing a statistically significant shift across the entire distribution rank ordering, thus leading to a VUS-PR improvement with a p-value > 0.05.

---

> > ### Comment · Reviewer_vmjz · 2025-08-03
> >
> > Thanks for the detailed response from the authors. My main concerns are addressed.

---

> > > ### Author Response · Authors · 2025-08-03
> > >
> > > Thank you for your further reply. Your insightful comments are very helpful for us to improve the clarity and quality of our work. Thank you again for your time and valuable suggestions.

---

### Official Review · Reviewer_3SEA · 2025-07-03

**Clarity:** 3
**Significance:** 2
**Originality:** 2
**Rating:** 5
**Confidence:** 3

**Summary:**

This paper proposes a plug-and-play module named Idempotent Generation for Anomaly Detection (IGAD), which can be flexibly combined with a reconstruction-based method without introducing additional trainable parameters.  This method helps to release over-generalization and balance between robustness and sensitivity from a manifold perspective. Experiments on four datasets demonstrate its effectiveness.

**Questions:**

1. Are all the experimental results listed in Table 1reimplemented or directly from the original paper? Clear marks should be added.

**Ethical Concerns:**

["NO or VERY MINOR ethics concerns only"]

**Final Justification:**

My concerns have been addressed during the rebuttal, thus I have decided to raise my score to 5.

**Limitations:**

Not mentioned.

**Paper Formatting Concerns:**

No major formatting issues.

**Quality:**

3

**Strengths And Weaknesses:**

Strengths:
1. Figure 1 clearly illustrates the motivation of this paper.
2. In the experiments, 15 reconstruction-based multivariate time series anomaly detection methods are utilized as backbone models. Experimental results are sufficient.
3. Representation is clear and fluent.

Weakness:
1. The lower part of Figure 2 should be improved with clear information flow, as well as its caption.
2. There are lots of hyperparameters of losses in Eq.11. There is not sufficient analysis about how these hyperparameters influence the model performance. Do those hyperparameters need to be carefully fine-tuned?
3. Is the method implemented on the pretrained backbone models or trained from initialization?
4. The proposed model performs as an additional module based on the backbone model. Thus, it is important to analyze the increased complexity and inference time, compared to the backbones.

---

> ### Author Rebuttal · Authors · 2025-07-29
>
> Thank you for your constructive comments. We provide our feedbacks as follows.
> # Main Comments:
> > **W1: The lower part of Figure 2 should be improved with clear information flow, as well as its caption.**
>
> **A-W1:** Thank you for pointing this out. We first further explain the information flow in Fig.2, then we will update and integrate these information into this figure and add more descriptions to improve it.
>
> 1. **The first part of Fig.2 lies in the top half,** which means the conventional reconstruction workflow. A time window is fed into the backbone architecture for MTS AD, and the optimization objective is to minimize the deviation between the original time window $x^i$ and the reconstructed time window $x^i_\text{recon}$. These operations are employed to calculate $\mathcal{L}_\text{recon}$:
> \begin{equation}
> \mathcal{D}\_{\mathrm{recon}}(x,f(x))=\frac{1}{m}\sum\_{i=1}^{m}\|\|x^{i}-x\_{\mathrm{recon}}^{i}\|\|\_{2}^{2}, \mathcal{L}\_{\mathrm{recon}}(x,f(x))=\mathbb{E}\_{x}[\mathcal{D}\_{\mathrm{recon}}(x,f(x))].
> \end{equation}
>
> 2. **The second part of Fig.2 lies in the bottom half,** which contains the workflow of a given model integrated with our proposed IGAD plugin. Concretely, **in the left side of this part**, when we have a time window $x^i$, we first conduct FFT on $x^i$ to get the frequency-domain signal. This signal can be expressed with a complex form. Here, according to Eq.3, we use $\mathbf{Re}(\cdot)$ and $\mathbf{Im}(\cdot)$ to extract the real part and image part. After we get the real part and image part, we can calculate the mean and standard deviation of them, which can be denoted as $\mu_{\text{real}}^i$, $\sigma_{\text{real}}^i$, $\mu_{\text{image}}^i$ and $\sigma_{\text{image}}^i$, respectively. Then, we randomly sample vectors from $\mathcal{N}(\mu_{\text{real}}^i, \sigma_{\text{real}}^i)$ and $\mathcal{N}(\mu_{\text{real}}^i, \sigma_{\text{real}}^i)$ to get the sampled-effected vector $z^i$ through IFFT for the following calculation. **As for the right side of this part**, it is the information flow of IGAD. The red arrow and the blue arrow in the right manifold stand for the training and frozen model, i.e., $f(\cdot)$ and $f'(\cdot)$, then $z^i$ is passed through multiple mappings $f(f'(z^i))$ and $f'(f(z^i))$ for the calculation of $\mathcal{L}\_\text{idem}$ and $\mathcal{L}_\text{tight}$, respectively:
> \begin{equation}
> \mathcal{D}\_{i\mathrm{dem}}(f(z),f^{\prime}(f(z)))=\frac{1}{m}\sum\_{i=1}^m\left|f(z^i)-f^{\prime}(f(z^i))\right|,\mathcal{L}\_{i\mathrm{dem}}(f(z),f^{\prime}(z))=\mathbb{E}\_z[\mathcal{D}\_{i\mathrm{dem}}(f(z),f^{\prime}(f(z)))],
> \end{equation}
> \begin{equation}
> \mathcal{D}\_{\mathrm{dight}}(f^{\prime}(z),f(f^{\prime}(z)))=\frac{1}{m}\sum\_{i=1}^{m}\left|f^{\prime}(z^{i})-f(f^{\prime}(z^{i}))\right|,\mathcal{L}\_{\mathrm{dight}}(f(z),f^{\prime}(z))=\mathbb{E}\_{z}[-\mathcal{D}\_{\mathrm{dight}}(f^{\prime}(z),f(f^{\prime}(z)))].
> \end{equation}
>
> We hope these descriptions can help readers better unserstand the information flow of IGAD, and we will further improve this figure and caption in our manuscript.
>
> > **W2: There is not sufficient analysis about how these hyperparameters influence the model performance. Do those hyperparameters need to be carefully fine-tuned?**
>
> **A-W2:** Thank you for pointing out the importance of analyzing the hyperparameters introduced by IGAD. There are five relative hyperpatameters in Eq.11, including the weights of different loss items ($\lambda_{\text{recon}}$, $\lambda_{\text{idem}}$, $\lambda_{\text{tight}}$ and $\lambda_{\text{aux}}$) and the control parameter $\alpha$. As we mentioned in Sect.4.1.2, the latest work on time series anomaly detection benchmark [1] has integrated the most of the models we have selected in our experiments and searched for the optimal hyperparameters in optimizer, learning rate, and weights of existing loss functions for them. For some of our selected base models, which are not temporarily imported, we set their hyperparameters in their original papers or repositories as the optimal ones. **These mean that $\lambda_{\text{recon}}$ and $\lambda_{\text{aux}}$ have been fixed as the optimal ones, and we need to explore the effects of $\lambda_{\text{idem}}$, $\lambda_{\text{tight}}$ and $\alpha$ in IGAD.** The search interval for $\lambda_{\text{idem}}$ and $\lambda_{\text{tight}}$ is set as [0.1, 1.0] with a step of 0.1, and [1.1, 1.5] with a step of 0.1 for $\alpha$. During this process, we select OPTUNA [2] to help search. **We summarize the results of hyperparameter influence in Appendix D (Page 23) and Appendix E.3 (Page 35 to Page 37), including Tab.4, Fig.7, Fig.8 and Fig.9.** We also list potential selection strategies for a new dataset. It can be concluded as that $\lambda_{\text{idem}}$ in [0.1, 0.5], not too strict $\lambda_{\text{tight}}$ and a larger $\alpha$ tend to achieve better performance.
>
> [1] The Elephant in the Room: Towards A Reliable Time-Series Anomaly Detection Benchmark, NeurIPS DB Track, 2024.
>
> [2] OPTUNA: A Next-generation Hyperparameter Optimization Framework, KDD 2019.
>
> > **W3: Is the method implemented on the pretrained backbone models or trained from initialization?**
>
> **A-W3:** In our selected models, OFA is an anomaly detection model based on GPT-2, so it is trained after loaded the pre-trained weights from huggingface. For the other models, they are trained from initialization according to the random seeds, following the standard training principle.
>
> > **W4:It is important to analyze the increased complexity and inference time, compared to the backbones.**
>
> **A-W4:** Thank you for your suggestions on this key point. **We conduct effectiveness analysis and report the results in Appendix E.2 (Page 33).** In this part, we calculate the training time, max GPU allocation and max CPU allocation. We admit that additional mappings from IGAD will increase the time and resource allocation during training, **but these costs occur only in the training stage. During inference, each model can perform only one traditional mapping for reconstruction, calculate anomaly scores, and detect abnormal time points.** We find that in the most cases, the models with IGAD show an increase in time of less than 10 seconds. The increase generally has relatively limited impact on the overall training process, especially when higher performance is desired. Meanwhile, model OFA tends to show a higher increase. This is because that OFA is a pre-trained language model, which means that it has more dense parameters than traditional models for AD. To reduce these potential cost, we list three possible future strategies in this section (Page 33):
> 1. Data and model parallelism;
>
> 2. Fint-tune large models with parameter-efficient methods;
>
> 3. Build memory mechanism to save necessary information, reducing the number of full reconstruction.
>
> > **Q1: Are all the experimental results listed in Table 1 reimplemented or directly from the original paper? Clear marks should be added.**
>
> **A-Q1:** All results listed in Tab.1 are **reimplemented.** We follow the experimental setting in Sect.4.1.2 and rerun models to report their results with five random seeds. We have added the corresponding information in Sect.4.1.2 for better clarification.

---

> > ### Comment · Reviewer_3SEA · 2025-08-06
> >
> > Thanks for the detailed response. Most of my questions have been solved. In addition, the detailed reference time comparison should be listed.

---

> ### Author Response · Authors · 2025-08-06
>
> **Thank you for your further reply and valuable suggestions. We provide more information on the time comparison before and after applying IGAD.**
>
> **1. Training Stage**
>
> First, we would like to clarify that IGAD only effects the training process due to the additional mappings, $f(f'(z))$ and $f'(f(z))$, for calculating $\mathcal{L}\_\text{idem}$ and $\mathcal{L}\_\text{tight}$. To analyze these, we use **Appendix E.2, including Tab.13, Tab.14, Tab.15 and Tab.16 (Page 33)** to show that these costs can be controllable in the most cases, and we also provide potential strategies for optimization and larger models. This section can help explain the influence of IGAD during the training phase, and we also report the results in seconds per epoch here.
>
> |**Model**|**Training Time (s)**||||||||
> |:---:|:---:|:---:|:---:|:---:|:---:|:---:|:---:|:---:|
> ||SMD||MSL||PSM||SMAP||
> ||w/oIGAD|w/IGAD|w/oIGAD|w/IGAD|w/oIGAD|w/IGAD|w/oIGAD|w/IGAD|
> |CATCH|6.0878|17.2277|1.5938|3.3659|44.4446|131.0026|2.5922|6.4548|
> |M2N2|0.8424|1.2809|0.5443|0.9091|2.5754|4.8112|0.6537|0.9807|
> |FITS|1.1934|1.9876|0.8837|1.0449|4.0280|8.0651|0.9403|1.2141|
> |ModernTCN|3.4464|9.9653|1.2722|2.3140|15.9907|54.3645|1.6361|3.5830|
> |Peri-midFormer|1.1767|5.8734|0.8942|1.4531|4.2939|22.7822|1.0926|2.1401|
> |SARAD|1.4122|2.5430|0.9593|1.1728|4.9100|14.7800|1.0455|1.4257|
> |TimesNet|1.3733|2.6364|0.8971|1.0556|5.4077|16.3706|1.0640|1.6658|
> |OFA|3.6833|21.4194|1.2976|3.4958|24.8965|225.4139|2.2287|10.4260|
> |AnomalyTransformer|2.0369|3.9399|0.9892|1.2458|13.7400|34.3816|1.3751|2.2921|
> |FGANomaly|1.6585|2.1942|0.9215|1.0265|9.6031|15.4628|1.1534|1.3887|
> |CAE-M|0.9261|1.2022|0.8479|0.8915|1.7830|3.0684|0.8584|0.9544|
> |MTAD-GAT|1.2675|2.2031|0.8763|1.0014|3.6808|10.4775|0.9592|1.3120|
> |OmniAnomaly|1.0875|1.8211|0.8426|0.9404|4.0156|10.3294|0.9489|1.2228|
> |MSCRED|2.8281|10.2687|1.2203|2.3347|16.2451|58.4670|1.6051|3.7906|
> |DAGMM|0.9995|1.1030|0.8396|0.8602|1.8711|3.0836|0.8615|0.9203|
>
> **2. Inference Stage**
> Second, as for the inference, theoretically, the time costs of the models with IGAD should be identical to the models without IGAD. This is because that during this stage, **each model performs only one traditional mappping to get the reconstructed $x^i_\text{recon}$ from $x^i$ to detect anomalies, which is unified for models with IGAD and without IGAD.** To verify this, we further include the results on the time costs during the inference stage in the following table. The inference phase can be viewed as a complete process in which we feed the time points into the trained model to classify whether each time point is normal or abnormal. Then, we record the complete inference time with the average values of five repeats for base models with different datasets. It can be found that the time cost for inference stage are closely identical and the differences mainly come from system runtime deviation.
>
> |**Model**|**Inference Time (s)**||||||||
> |:---:|:---:|:---:|:---:|:---:|:---:|:---:|:---:|:---:|
> ||SMD||MSL||PSM||SMAP||
> ||w/oIGAD|w/IGAD|w/oIGAD|w/IGAD|w/oIGAD|w/IGAD|w/oIGAD|w/IGAD|
> |CATCH|24.4450|24.4437|2.5908|2.5845|151.0697|151.0619|5.5776|5.5728|
> |M2N2|1.2005|1.2014|0.1043|0.1078|9.7503|9.7439|0.3980|0.4057|
> |FITS|0.9924|0.9960|0.1036|0.1067|7.5676|7.5704|0.3056|0.3113|
> |ModernTCN|3.9831|3.9790|0.4439|0.4361|23.8767|23.8792|0.9142|0.9176|
> |Peri-midFormer|3.4294|3.4286|0.3596|0.3562|21.2394|21.2437|1.6134|1.6176|
> |SARAD|1.7923|1.7928|0.2421|0.2453|12.8560|12.8524|0.5089|0.5129|
> |TimesNet|1.9518|1.9575|0.1621|0.1561|17.3870|17.3911|0.6388|0.6430|
> |OFA|8.5801|8.5803|0.6578|0.6531|77.4751|77.4740|2.9019|2.8999|
> |AnomalyTransformer|2.3277|2.3318|0.2001|0.2035|20.8671|20.8701|0.8329|0.8394|
> |FGANomaly|17.1768|17.1747|1.3507|1.3567|104.3454|104.3412|3.8336|3.8333|
> |CAE-M|0.7101|0.7167|0.0945|0.0876|5.9587|5.9569|0.2073|0.2037|
> |MTAD-GAT|2.8098|2.8112|0.3249|0.3226|18.4588|18.4621|0.7124|0.7186|
> |OmniAnomaly|0.5758|0.5756|0.0927|0.0988|5.2297|5.2220|0.2068|0.2084|
> |MSCRED|5.9769|5.9710|0.7198|0.7260|40.8488|40.8556|1.6010|1.6005|
> |DAGMM|0.4497|0.4511|0.0764|0.0715|4.1214|4.1271|0.1801|0.1739|
>
> We hope this additional information can provide further clarification on both the training and inference time comparison. Thank you again for your time and suggestions. If any points remain unclear, we are willing to discuss them further.

---

### Comment · Area_Chair_9tdj · 2025-08-08
**Diverging opinions in Review**

Dear Reviewers,

thx for engaging in discussions with the authors. Your opinions are diverging, so it would help my work if you could give a qualified statement during the AC-reviewer discussion period.

If you need further information from the authors, can you try to use the remaining time for clarifying questions or answering their comment?

Thx, AC

---

### Decision · Program_Chairs · 2025-09-17

**Decision:**

Accept (poster)

**Comment:**

This paper proposes IGAD, a plug-and-play module for reconstruction-based multivariate time series anomaly detection, inspired by Idempotent Generative Networks but adapted with redefined objectives, FFT/IFFT-based perturbations, and a tightness loss to mitigate over-generalization. IGAD requires no additional trainable parameters and can be integrated with diverse backbone models.

The method is evaluated on 15 models over 4 datasets, later expanded to 6 in rebuttal, showing consistent VUS-PR improvements in most cases, with detailed ablations and efficiency analysis. The inference cost is unchanged and training cost increases only moderately.

The reviewers are split: two recommend accept (strong empirical results, broad applicability), one gives borderline accept (concerns on dataset scope and overhead largely addressed), and one gives borderline reject (perceived high similarity to prior work [45] and no consistent SOTA performance over contrastive methods). The rebuttal added datasets, clarifications, and extra experiments, resolving most concerns except for novelty relative to IGN.

After looking at this borderline paper myself, I can recommend acceptence. While some novelty overlap is a valid concern, the adaptation to MTS AD, no-parameter-overhead design, and broad applicability make IGAD a practical contribution, which is well-validated by experiments.

For the camera ready version, I suggest the authors include all their promised changes.